# Temporary Feature Collapse Phenomenon in Early Learning of MLPs

## Abstract

In this paper, we focus on a typical two-phase phenomenon in the learning of multi-layer perceptrons (MLPs). We discover and explain the reason for the feature collapse phenomenon in the first phase, *i.e.*, the diversity of features over different samples keeps decreasing in the first phase, until samples of different categories share almost the same feature, which hurts the optimization of MLPs. We find that such phenomena usually occur when MLPs are difficult to be trained. We explain such a phenomenon in terms of the learning dynamics of MLPs. Furthermore, we theoretically analyze the reason why four typical operations can alleviate the feature collapse. The code has been attached with the submission.

## 1 Introduction

It has been widely observed that in initialized neural networks, especially when the network is deep, the loss decrease is likely to have two phases during early epochs of learning, *e.g.*, phenomena observed in (Saxe et al., 2013; Simsekli et al., 2019; Stevens et al., 2020). As Figure 1(a) shows, the first phase is usually relatively short, in which the training loss does not decrease or decreases very slowly. Then, in the second phase, the training loss suddenly begins to decrease fast.

In particular, as Figure 1(b) shows, the length of the first phase increases along with the network complexity. In some extreme cases when deep neural networks (DNNs) are very deep, the loss minimization gets stuck, which can be considered as a strong first phase with an infinite length, namely, a ***learning-sticking problem***. In fact, the learning-sticking problem is quite common in practice. Jepkoech et al. (2021) and Stevens et al. (2020) empirically observed the learning-sticking problem without any theoretical analysis. People usually owed the learning-sticking problem to the over-parameterized settings of DNNs or the optimization ability of DNNs.

However, we discover and attempt to further theoretically explain a new, quite common, yet counter-intuitive phenomenon in the first phase (the learning-sticking phase). That is, as Figure 1(b) shows, *features of different categories become increasingly similar to each other.* In some cases, **the feature diversity keeps decreasing even until all samples of different categories share almost the same feature in the first phase.** We can consider this as the ***temporary feature collapse*** (TFC). This TFC happens in various DNNs, including multi-layer perceptrons (MLPs), convolutional neural networks, and recurrent neural networks (see both Figure 2 and Appendix B). DNNs trained with different loss functions and different learning rates may all exhibit TFC phenomena. The TFC phenomenon usually happens in the early epochs of the training process, especially when the DNN is difficult to optimize. According to our analysis, when the DNN is very deep, when the task is difficult, when the variance of initial weights is small, and when the DNN is trained without momentum or batch normalization layers, the DNN is more likely to exhibit the TFC phenomenon.

Based on our theoretical analysis, we discover a set of conditions that strengthen the TFC phenomenon. Then, we can easily control such conditions by applying typical operations *i.e.*, batch normalization, momentum, $L_2$ regularization, and network initialization. Specifically, we investigate the learning dynamics of the MLP. Moreover, we theoretically explain that these conditions make the training of DNNs more likely to perform like a ***"self-enhanced system"*** towards the TFC phenomenon in early iterations. In comparison, (Glorot & Bengio, 2010; Saxe et al., 2013) investigated the influence of initialization methods on the learning-sticking problem. To this end, we find that the effectiveness of initialization methods is probably owing to the large variance of initial weights, which avoids the TFC phenomenon during the learning-sticking phase.

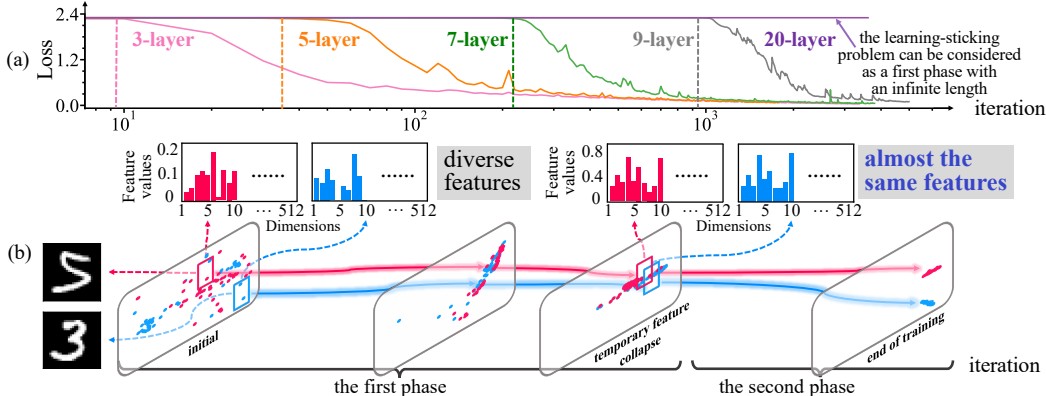

Figure 1: (a) The first phase (learning iterations before the dotted line) gets an increasing length and finally becomes the learning-sticking problem (purple curve), when the DNN has more layers. (b) Samples of different categories share almost the same features at the end of the first phase. We can consider this as a TFC phenomenon. We visualize the learning dynamics of an intermediate-layer feature in a 9-layer MLP. We select 10 salient dimensions to illustrate the feature similarity.

Fortunately, we discover that, when we use four typical operations to alleviate the TFC phenomenon, the learning-sticking problem can also be solved. Although previous studies have provided insightful analysis for these well-known operations, *e.g.,* batch normalization and network initialization, we are the first to establish the relationship between the TFC phenomenon and these typical operations. This provides theoretical guidance for the design of DNNs.

More crucially, the TFC phenomenon with the MLP is counter-intuitive, and has been neglected for a long time. The investigation of the learning dynamics of the TFC phenomenon would be useful for explaining complex optimization behaviors of DNNs and is of considerable value.

Contributions of this study can be summarized as follows. (1) We discover the common TFC phenomenon in early learning of the MLP, which has been ignored for a long time. (2) We explain this phenomenon from the perspective of learning dynamics. (3) We explain why four types of operations can alleviate the TFC phenomenon.

## 2 DISCOVERING THE TFC PHENOMENON

It has been widely observed that the loss decrease of DNNs is likely to have two phases (Saxe et al., 2013; Simsekli et al., 2019; Stevens et al., 2020). As Figure 1(b) shows, the training loss does not decrease significantly in the first phase, and the training loss suddenly begins to decrease in the second phase. In this paper, **we discover a new and counter-intuitive phenomenon in the first phase that both the diversity of intermediate-layer features over different samples and the diversity of feature gradients keep decreasing, until samples of different categories share almost the same feature in the first phase.** We consider this as a TFC phenomenon.

We consider an MLP $f$ with $L$ concatenated linear layers, each being followed by a ReLU layer. Only the last linear layer is followed by a softmax operation. Let $W_t^{(l)} \in \mathbb{R}^{h \times d}$ denote the weight matrix of the $l$-th linear layer with $h$ neurons ($1 \leqslant l \leqslant L$), and $W_t^{(l)}$ has been learned for $t$ iterations. Given an input sample $x$, the layer-wise forward propagation in the $l$-th layer is represented as

$$F_t^{(l)} = \text{ReLU}(W_t^{(l)} F_t^{(l-1)}) = D_t^{(l)} W_t^{(l)} F_t^{(l-1)}, \tag{1}$$

where $F_t^{(l)} \in \mathbb{R}^h$ denotes the output feature of the $l$-th layer after the $t$-th iteration. $D_t^{(l)}$ denotes a diagonal matrix, which represents gating states in the ReLU layer, and $D_{t,(i,i)}^{(l)} \in \{0, 1\}$.

Thus, the TFC phenomenon is shown as follows. Given two input samples $x_1$ and $x_2$, the cosine similarity of features $\cos(F_t^{(l)}|_{x_1}, F_t^{(l)}|_{x_2})$, and the cosine similarity of gradients $\cos(\dot{F}_t^{(l)}|_{x_1}, \dot{F}_t^{(l)}|_{x_2})$ keep increasing, which demonstrates the phenomenon. Here, $\dot{F}_t^{(l)}$ denotes the gradient of the loss *w.r.t.* the feature $F_t^{(l)}$. Besides, the increasing trend of feature similarity only exists in the first phase.

The TFC phenomenon is widely shared by different DNNs learned for different tasks. In early epochs (or iterations) of the training process, we observed such TFC phenomena on MLPs, VGG-11

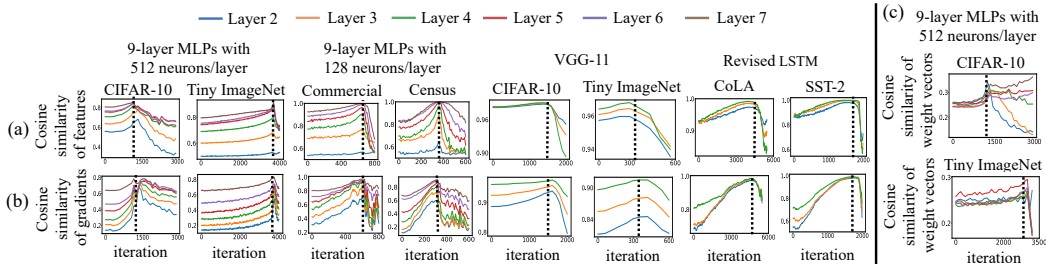

Figure 2: The TFC phenomenon. (a) Cosine similarity of features between samples in different categories $\mathbb{E}_{x,x' \in X}[\cos(F_t^{(l)}|_x, F_t^{(l)}|_{x'})]$ keeps increasing in the first phase (left to the dotted line), until the second phase. The low cosine similarity indicates the high diversity. (b) Cosine similarity of feature gradients between different samples of a category $\mathbb{E}_{x,x' \in X_c}[\cos(\dot{F}_t^{(l)}|_x, \dot{F}_t^{(l)}|_{x'})]$ keeps increasing in the first phase until the second phase, where $X_c$ denotes samples of the category $c$. (c) Cosine similarity of weight changes between weight vectors in a layer $\mathbb{E}_{x \in X} \cos(\Delta w_{t,i}^{(l)}|_x, \Delta w_{t,j}^{(l)}|_x)$ keeps increasing in the first phase. *Please see Appendix B for results on more DNNs.*

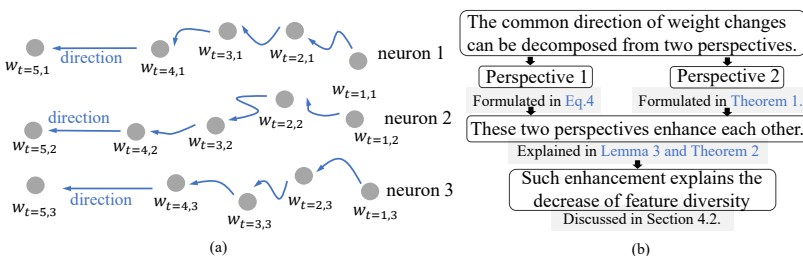

Figure 3: (a) Weights of neurons are changed towards a common direction. (b) The logic of explaining the TFC phenomenon.

(Simonyan & Zisserman, 2014), and the revised long short-term memory (LSTM) on different types of data, including image data (MNIST (LeCun et al., 1998), CIFAR-10 (Krizhevsky et al., 2009), and the Tiny ImageNet dataset (Le & Yang, 2015)), tabular data (two UCI datasets of census income and TV news (Asuncion & Newman, 2007)), and natural language data (CoLA (Warstadt et al., 2019), SST-2 (Socher et al., 2013), and AGNews (Del Corso et al., 2005)). We also tested MLPs with different loss functions, with Leaky ReLU layers (Maas et al., 2013), with different learning rates, and with different batch sizes. Figure 2(a,b) shows TFC phenomena on these DNNs, and please Appendix B for results on more DNNs.

**Besides, the learning-sticking problem can be considered as an extreme long first phase.** As Figure 1(a) shows, the length of the first phase increases along with the network complexity (depth). In extreme cases when DNNs are very deep or the task is difficult, the first phase reaches an infinite length, and the learning gets stuck (please see Appendix C for more discussions).

## 3 EXPLAINING THE DYNAMICS OF THE TFC PHENOMENON

In this section, we aim to investigate dynamics of network parameters in early epochs, so as to explain the condition that may boost the likelihood of the TFC phenomenon. In Section 3.1, we find that the decreasing diversity of feature gradients over different samples is owing to the phenomenon that different neurons in a layer are optimized towards a common direction in the first phase. Therefore, we propose two perspectives to illustrate the significance of the common direction. Then, in Section 3.2, we compare these two perspectives to analyze learning dynamics, and we find that the significance of the common direction may be enhanced, just like a "self-enhanced system." Finally, the self-enhanced common direction can explain the TFC phenomenon. The overall logic of the explanation is illustrated in Figure 3(b). In Section 3.3, we explain the reason why four types of operations can alleviate the TFC phenomenon based on our analysis.

### 3.1 TWO PERSPECTIVES TO ANALYZE THE COMMON DIRECTION OF LEARNING EFFECTS

In the beginning, let us first focus on the conjecture that **the decreasing diversity of feature gradients over different samples can be explained by** the phenomenon that different neurons in a layer

are optimized towards a common direction in the first phase. For example, as Figure 3(a) shows, at the beginning of the learning, different neurons are originally optimized towards different directions, but then gradients of different neurons gradually change to a similar direction. Let $\dot{F}_t^{(l)}$ denote the gradient of the loss *w.r.t.* the feature $F_t^{(l)}$ at the $l$-th layer. Then, according to Eq. (1), the back propagation of feature gradients $\dot{F}_t^{(l)} \in \mathbb{R}^h$ at the $l$-th layer can be written as

$$\dot{F}_t^{(l-1)} = W_t^{(l)^\top} D_t^{(l)} \dot{F}_t^{(l)}. \tag{2}$$

The emergence of a common direction of weight changes means that gradients of the $d$ **weight vectors** in $W_t^{(l)^\top} = [w_{t,1}^{(l)}, w_{t,2}^{(l)}, \cdots, w_{t,d}^{(l)}]^\top \in \mathbb{R}^{d \times h}$, *i.e.*, $\partial Loss / \partial w_{t,i}^{(l)}$, gradually become approximately collinear. According to Remark 1, we can explain why the enhancement of such a common direction decreases the diversity of feature gradients.

**Remark 1.** *Let us assume that different weight vectors $[w_{t,1}^{(l)}, w_{t,2}^{(l)}, \cdots, w_{t,d}^{(l)}]^\top$ have a dominating common direction $C^{(l)} \in \mathbb{R}^h$. Then, we can represent $w_{t,i}^{(l)} = \beta_i C^{(l)} + \epsilon_i$, where $\beta_i \in \mathbb{R}$; $\epsilon_i \in \mathbb{R}^h$ denotes a small residual; $\boldsymbol{\beta} = [\beta_1, \beta_2, \cdots, \beta_d]$, and $\boldsymbol{\epsilon} = [\epsilon_1, \epsilon_2, \cdots, \epsilon_d]^\top$. Then, we have*

$$\dot{F}_t^{(l-1)} = (C^{(l)^\top} D_t^{(l)} \dot{F}_t^{(l)}) \cdot \boldsymbol{\beta} + \boldsymbol{\epsilon} D_t^{(l)} \dot{F}_t^{(l)}. \tag{3}$$

Remark 1 well explains the rationale for the above conjecture. That is, if $\partial Loss / \partial w_{t,i}^{(l)}$ on different samples are roughly collinear to each other, then such a collinearity would make feature gradients $\dot{F}_t^{(l-1)}$ of different samples similar to each other. Specifically, during the learning process, if the DNN keeps optimizing $W_t^{(l)^\top}$ along the common direction $C^{(l)}$ for a long time, which keeps strengthening the value $C^{(l)^\top} D_t^{(l)} \dot{F}_t^{(l)} \in \mathbb{R}$, then feature gradients $\dot{F}_t^{(l-1)}$ of different samples are gradually pushed towards the same direction $\boldsymbol{\beta}$. In other words, as long as different weight vectors are optimized towards the same dominating direction, then feature gradients $\dot{F}_t^{(l-1)}$ are pushed in the same direction $\boldsymbol{\beta}$.

**Therefore, the first core task of proving the decreasing diversity of feature gradients is to explain the existence of the common optimization direction shared by different weight vectors.** Thus, we propose two perspectives to illustrate how different weight vectors $w_{t,i}^{(l)}$ are changed $\Delta W_t$ along a common direction during the learning process. By comparing these two perspectives, we can further explain the reason why the significance of the common direction will be further boosted, just like a "self-enhanced system." Such a "self-enhanced system" will be proven in Section 3.2.

**Perspective 1.** This perspective focuses on the influence of the common direction $C^{(l)}$ of the weight change in $l$-th layer. For clarity, we omit the superscript $(l)$ to simplify the notation in the following paragraphs in Section 3.1, *i.e.*, $\Delta w_{t,i}^{(l)}$, $\Delta W_t^{(l)}$, and $C^{(l)}$ can be simplified by $\Delta w_{t,i}$, $\Delta W_t$, and $C$, respectively. Let $\Delta W_t^\top = [\Delta w_{t,1}, \Delta w_{t,2}, \cdots, \Delta w_{t,d}]^\top$ denote weight changes of $d$ weight vectors in the $l$-th layer. We decompose $\Delta W_t^\top$ into the component along a common direction $C$ and a component along other directions as follows.

$$\Delta W_t^\top = \Delta V_t C^\top + \Delta \varepsilon_t, \tag{4}$$

where $\Delta V_t = [\Delta v_{t,1}, \Delta v_{t,2}, \cdots, \Delta v_{t,d}] \in \mathbb{R}^d$ denotes the coefficient vector for weight changes of different weight vectors along the common direction $C$. Specifically, $\Delta \varepsilon_t$ is relatively small "noise" term, which is orthogonal to $C$, *i.e.*, $\Delta \varepsilon_t C = \mathbf{0}$.

**Lemma 1.** *(Proof in Appendix E) For the decomposition $\Delta W_t^\top = \Delta V_t C^\top + \Delta \varepsilon_t$, given weight changes over different samples $\Delta W_t^\top$, we can compute the common direction $C$ by minimizing the fitting error $\Delta \epsilon_t$ when we use $\Delta v_{t,i} C^\top$ to approximate $\Delta w_{t,i}^\top$ over different samples across different iterations. I.e., $\min_{C, \Delta V_t |_x} \left( \mathbb{E}_{t \in [T_{start}, T_{end}]} \mathbb{E}_{x \in X} \|\Delta \varepsilon_t|_x\|_F^2 \right)$, s.t. $\Delta \varepsilon_t|_x = \Delta W_t^\top|_x - \Delta V_t|_x C^\top$. Thus, we obtain $\Delta V_t = \frac{\Delta W_t^\top C}{C^\top C}$ and $\Delta \varepsilon_t = \Delta W_t^\top - \Delta W_t^\top \frac{C C^\top}{C^\top C}$, s.t. $\Delta \varepsilon_t C = \mathbf{0}$. Such settings minimize $\|\Delta \varepsilon_t\|_F$.*

**Lemma 2.** *(**We can also decompose the weight $W_t^{(l)}$ into the component along the common direction $C$ and the component $\varepsilon_t$ in other directions**. Proof is in Appendix F.) Given the weight $W_t^\top$ and the common direction $C$, the decomposition $W_t^\top = V_t C^\top + \varepsilon_t$ can be conducted as $V_t = \frac{W_t^\top C}{C^\top C}$ and $\varepsilon_t = W_t^\top - W_t^\top \frac{C C^\top}{C^\top C}$ s.t. $\varepsilon_t C = \mathbf{0}$. Such settings minimize $\|\varepsilon_t\|_F$.*

We conduct experiments to verify the strength of the primary common direction $C$. To this end, let us focus on the average weight change over different samples $\Delta \overline{W}_t = \mathbb{E}_{x \in X} \Delta W_t|_x$. Then, we

Figure 4: The strength of different common directions in the CIFAR-10 dataset. We trained 9-layer MLPs, where each layer of the MLP had 512 neurons. We illustrated results on the two categories with the highest training accuracies. $s_i = \|C_i \Delta \overline{V}_i^\top\|_F$ measures the strength of weight changes along the $i$-th common direction, where $\Delta \overline{V}_i = \mathbb{E}_t[\Delta \overline{V}_{i,t}]$. The strength of the primary direction was much greater than the strength of other directions. Please see Appendix D for more results on the MNIST dataset and the Tiny ImageNet dataset.

Table 1: Strength of components of weight changes along the common direction and other directions. We trained 9-layer MLPs on the CIFAR-10 dataset and the Tiny ImageNet dataset, respectively. Each layer of the MLP had 512 neurons. The strength of the primary common direction was much greater than those of other directions. Appendix D provides results on the MNIST dataset and Appendix H.2 explains the phenomenon that $S_1^{(l)}$, $S_2^{(l)}$, and $S_3^{(l)}$ do not decrease monotonically.

| | Category | | | Cat | | | | | Truck | | |
|---|---|---|---|---|---|---|---|---|---|---|---|
| | $S\ (\times 10^{-3})$ | Layer 2 | Layer 3 | Layer 4 | Layer 5 | Layer 6 | Layer 2 | Layer 3 | Layer 4 | Layer 5 | Layer 6 |
| CIFAR-10 | $S_{\text{primary}}^{(l)}$ | $154.0_{\pm 17.1}$ | $176.5_{\pm 16.8}$ | $201.6_{\pm 18.7}$ | $253.6_{\pm 24.6}$ | $277.4_{\pm 25.6}$ | $169.9_{\pm 20.8}$ | $208.1_{\pm 21.5}$ | $223.6_{\pm 20.1}$ | $248.4_{\pm 19.2}$ | $281.5_{\pm 20.4}$ |
| | $S_1^{(l)}$ | $11.5_{\pm 1.5}$ | $13.0_{\pm 0.9}$ | $11.6_{\pm 1.7}$ | $16.1_{\pm 1.8}$ | $9.0_{\pm 0.8}$ | $15.6_{\pm 2.1}$ | $14.0_{\pm 1.8}$ | $14.3_{\pm 1.1}$ | $14.3_{\pm 1.7}$ | $10.0_{\pm 1.1}$ |
| | $S_2^{(l)}$ | $12.7_{\pm 1.7}$ | $11.9_{\pm 1.3}$ | $10.9_{\pm 1.3}$ | $11.9_{\pm 0.8}$ | $8.8_{\pm 1.1}$ | $14.4_{\pm 1.4}$ | $15.1_{\pm 2.0}$ | $11.3_{\pm 1.4}$ | $12.3_{\pm 0.9}$ | $12.9_{\pm 1.2}$ |
| | $S_3^{(l)}$ | $11.0_{\pm 1.1}$ | $14.4_{\pm 1.7}$ | $12.5_{\pm 2.2}$ | $13.9_{\pm 1.7}$ | $8.6_{\pm 1.1}$ | $14.3_{\pm 2.2}$ | $12.4_{\pm 1.9}$ | $12.8_{\pm 1.6}$ | $13.1_{\pm 1.2}$ | $9.7_{\pm 1.0}$ |
| | Category | | | Flagpole | | | | | Bottle | | |
| | $S\ (\times 10^{-3})$ | Layer 2 | Layer 3 | Layer 4 | Layer 5 | Layer 6 | Layer 2 | Layer 3 | Layer 4 | Layer 5 | Layer 6 |
| Tiny ImageNet | $S_{\text{primary}}^{(l)}$ | $97.8_{\pm 3.7}$ | $143.9_{\pm 5.6}$ | $198.9_{\pm 8.1}$ | $259.8_{\pm 10.1}$ | $322.8_{\pm 12.7}$ | $202.3_{\pm 12.2}$ | $234.4_{\pm 13.1}$ | $276.8_{\pm 13.9}$ | $345.2_{\pm 16.6}$ | $440.2_{\pm 22.2}$ |
| | $S_1^{(l)}$ | $10.6_{\pm 0.9}$ | $9.5_{\pm 0.8}$ | $14.4_{\pm 1.4}$ | $24.9_{\pm 1.3}$ | $8.8_{\pm 1.0}$ | $10.3_{\pm 1.4}$ | $11.2_{\pm 1.6}$ | $12.2_{\pm 1.3}$ | $11.9_{\pm 1.1}$ | $13.2_{\pm 1.6}$ |
| | $S_2^{(l)}$ | $7.5_{\pm 0.9}$ | $7.9_{\pm 1.2}$ | $9.7_{\pm 1.2}$ | $9.2_{\pm 1.2}$ | $8.3_{\pm 0.6}$ | $10.4_{\pm 1.1}$ | $11.6_{\pm 1.0}$ | $13.8_{\pm 1.3}$ | $10.0_{\pm 0.8}$ | $13.6_{\pm 1.2}$ |
| | $S_3^{(l)}$ | $7.1_{\pm 0.8}$ | $9.1_{\pm 1.1}$ | $11.3_{\pm 1.0}$ | $17.9_{\pm 2.2}$ | $16.6_{\pm 1.5}$ | $11.6_{\pm 1.4}$ | $15.7_{\pm 1.4}$ | $10.7_{\pm 1.1}$ | $10.8_{\pm 1.2}$ | $19.8_{\pm 1.6}$ |

decompose $\Delta \overline{W}_t$ into components along five common directions as $\Delta \overline{W}_t = C_1 \Delta \overline{V}_{1,t}^\top + C_2 \Delta \overline{V}_{2,t}^\top + \cdots + C_5 \Delta \overline{V}_{5,t}^\top + \Delta \overline{\varepsilon}_{5,t}^\top$, where $C_1 = C$ is termed the *primary common direction*. $C_2, C_3, C_4$ and $C_5$ represent the second, third, forth, and fifth common directions, respectively. $C_1, C_2, C_3, C_4$, and $C_5$ are orthogonal to each other. $C_i$ and $\Delta \overline{V}_{i,t}$ are computed based on Lemma 1 when we remove the first $(i-1)$ components along the direction $C, \cdots, C_{i-1}$ from the $\Delta \overline{W}_t$. Figure 4 shows that the strength of the primary common component $C_1 \Delta \overline{V}_1^\top$ is approximately ten times greater than the strength of the secondary common component $C_2 \Delta \overline{V}_2^\top$. Please see Appendix G for more discussions.

**Perspective 2 based on feature gradients $\dot{F}_t^{(l+1)}$.** We decompose the weight change by considering the influence of the common direction of the upper layer $C^{(l+1)}$. In order to distinguish variables belonging to different layers, we add the superscript $(l)$ back to $\Delta W_t^{(l)}$, $\Delta V_t^{(l)}$, and $\Delta \varepsilon_t^{(l)}$ to denote the layer in the following paragraphs.

**Theorem 1.** *(Proof in Appendix H.1) The weight change made by a sample can be decomposed into $(h+1)$ terms after the $t$-th iteration as follows.*

$$\Delta W_t^{(l)} = \Delta W_{\text{primary},t}^{(l)} + \sum_{k=1}^{h} \Delta W_{\text{noise},t}^{(l,k)} = \Gamma_t^{(l)} F_t^{(l-1)\top} + \kappa_t^{(l)\top}, \qquad (5)$$

*where $\Delta W_{\text{primary},t}^{(l)} = D_t^{(l)} V_t^{(l+1)} C^{(l+1)\top} C^{(l+1)} \Delta V_t^{(l+1)\top} F_t^{(l)} F_t^{(l-1)\top} / \|F_t^{(l)}\|_2^2$ denotes the component along the primary common direction, and $\Delta W_{\text{noise},t}^{(l,k)} = D_t^{(l)} \varepsilon_t^{(l+1,k)} \Delta \varepsilon_t^{(l+1)\top} F_t^{(l)} F_t^{(l-1)\top} / \|F_t^{(l)}\|_2^2$ denotes the component along the $k$-th common direction in the noise term. $\varepsilon_t^{(l+1,k)} = \Sigma_{kk} \mathcal{U}_k \mathcal{V}_k^\top$, where the SVD of $\varepsilon_t^{(l+1)} \in \mathbb{R}^{h \times h'}$ is given as $\varepsilon_t^{(l+1)} = \mathcal{U} \Sigma \mathcal{V}^\top$ $(h \leq h')$, and $\Sigma_{kk}$ denotes the $k$-th singular value $\in \mathbb{R}$. $\varepsilon_t^{(l+1)} = \sum_k \varepsilon_t^{(l+1,k)}$. $\mathcal{U}_k$ and $\mathcal{V}_k$ denote the $k$-th column of the matrix $\mathcal{U}$ and $\mathcal{V}$, respectively. Besides, we have $\forall k \in \{1,2,\ldots,h\}$, $\mathcal{U}_k^\top C^{(l+1)} = 0$. Consequently, we have $\Gamma_t^{(l)} = D_t^{(l)} V_t^{(l+1)} C^{(l+1)\top} C^{(l+1)} \Delta V_t^{(l+1)\top} F_t^{(l)} / \|F_t^{(l)}\|_2^2 \in \mathbb{R}^h$, and $\kappa_t^{(l)\top} = D_t^{(l)} \varepsilon_t^{(l+1)} \Delta \varepsilon_t^{(l+1)\top} F_t^{(l)} F_t^{(l-1)\top} / \|F_t^{(l)}\|_2^2 \in \mathbb{R}^{h \times d}$.*

Given weight changes $\Delta W_t^{(l)}$ made by a sample $x$, the primary term $\Delta W_{\text{primary},t}^{(l)}$ represents the component of weight changes along the common direction $C^{(l+1)}$. The $k$-th noise term $\Delta W_{\text{noise},t}^{(l,k)}$ represents the component along the $k$-th direction $\mathcal{U}_k$, which is orthogonal to $C^{(l+1)}$.

We conduct experiments to verify the significant strength of the component along the common direction $C^{(l+1)}$. We compute the average strength of the component along $C^{(l+1)}$ over all samples in $X$ as $S_{\text{primary}}^{(l)} = \mathbb{E}_{t \in [T_{\text{start}}, T_{\text{end}}]} \mathbb{E}_{x \in X}[\|\Delta W_{\text{primary},t}^{(l)}|x\|_F]$. Similarly, the strength of the component along the $k$-th noise direction is computed as $S_k^{(l)} = \mathbb{E}_{t \in [T_{\text{start}}, T_{\text{end}}]} \mathbb{E}_{x \in X}[\|\Delta W_{\text{noise},t}^{(l,k)}|x\|_F]$. Table 1 illustrates that the strength of the primary component $S_{\text{primary}}^{(l)}$ is more than ten times greater than the strength of components along noise directions $S_1^{(l)}, S_2^{(l)}$, and $S_3^{(l)}$.

*Discussion about comparing with the sum of all other directions' significance.* According to Table 1, it seems that the sum of strengths of components along other directions is also large. However, different directions decomposed by the above method are orthogonal to each other. Therefore, weight changes along different directions are independent, and their strengths cannot be summed up. Thus, we can directly compare the strength of the component of weight changes along each direction to verify the significant strength of the primary direction.

## 3.2 EXPLAINING THE ENHANCEMENT OF THE SIGNIFICANCE OF THE COMMON DIRECTION

The previous subsection owes the decreasing diversity of feature gradients to the typical common optimization direction shared by different weight vectors. In the current subsection, we explain that the common optimization direction phenomenon is very likely to be further enhanced, just like a "self-enhanced system." The self-enhancement of the optimization direction will explain the decreasing diversity. Specifically, the overall logic of this subsection has three steps. In Step 1, we explain the phenomenon that the significance of the common direction can be enhanced by training samples in a certain category in very early epochs. In Step 2, we extend the analysis of the enhancement of the common direction to a more generic case, *i.e.*, explaining the enhancement caused by training samples from different categories. In Step 3, we further explain that the self-enhancement of the common direction decreases the diversity of features and feature gradients, *i.e.*, explaining the TFC phenomenon.

Before explaining the enhancement of the significance of the common direction, let us first clarify assumptions in the proof. (1) The direct proof of the emergence of a "self-enhanced system" from the very beginning of training an initialized MLP is difficult. Instead, we explain that the self-enhancement of the common direction probably started under the background assumption that features of different samples have been pushed a little bit towards a specific common direction. (2) The MLP usually first learns a few categories, instead of simultaneously learning all categories. Experimental results in Figure 7 and Appendix O have verified the trustworthiness of this assumption.

According to Eq. (4) and Eq. (5), weight changes made by the sample $x$ can be given as

$$\text{Perspective 1: } \Delta W_t^{(l)} = C^{(l)} \Delta V_t^{(l)^\top} + \Delta \varepsilon_t^{(l)^\top} \quad \text{Perspective 2: } \Delta W_t^{(l)} = \Gamma_t^{(l)} F_t^{(l-1)^\top} + \kappa_t^{(l)^\top} \quad (6)$$

By comparing the above two perspectives, we discover an interesting potential that the common direction $C^{(l)}$ is similar to $\pm\Gamma_t^{(l)}$, and the feature $F_t^{(l-1)}$ is similar to the vector $\pm\Delta V_t^{(l)}$.

Inspired by this, **we aim to prove the self-enhancement of the significance of the common direction, by explaining the intuition that the feature $F_t^{(l-1)}$ and the vector $V_t^{(l)}$ become more and more similar to each other in the first phase.** As the first step of the proof, Theorem 2 shows that if we only consider training samples $x \in X_c$ in the same category $c$, then features $F_t^{(l-1)}$ of samples in this category would become increasingly similar to each other. On the other hand, such training samples have similar training effects, *i.e.*, pushing weights of different neurons $V_t^{(l)}$ all towards the average feature $\alpha_c \mathbb{E}_{x \in X_c}[F_t^{(l-1)}|x]$.

**Step 1: Explaining the significance of the common direction is enhanced by all training samples in a certain category.** Specifically, let us first consider the aforementioned background assumption that features $F_t^{(l-1)}$ of different samples have been pushed a little bit towards a specific common direction. We can obtain that there exists at least one learning iteration in the first phase, in which $\Delta F_t^{(l-1)}$ and $F_t^{(l-1)}$ of most samples have similar directions, and $\Delta V_t^{(l)}$ and $V_t^{(l)}$ have similar directions (see Appendix J for more discussions). Note that the assumed initial common direction is quite vague, and it is far from the TFC phenomenon. However, if we take the vague common direction as the starting point, we can further prove the significant self-enhancement of the common direction, which is responsible for the TFC phenomenon.

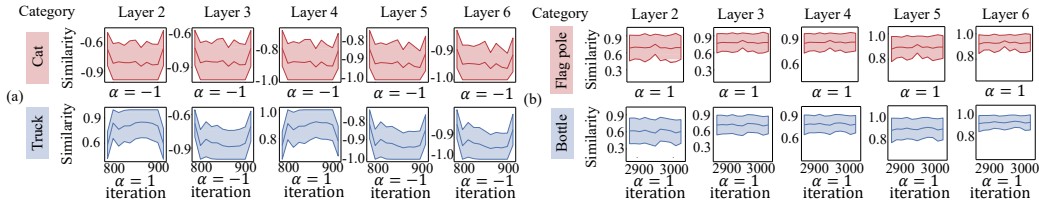

Figure 5: The average cosine similarity between the feature $F_t^{(l-1)}$ and the vector $\Delta V_t^{(l)}$ over different samples in the first phase. We conducted experiments on 9-layer MLPs trained on the (a) CIFAR-10 dataset, and the (b) Tiny ImageNet dataset. The shade in each subfigure represents the standard deviation of the cosine similarity over different samples.

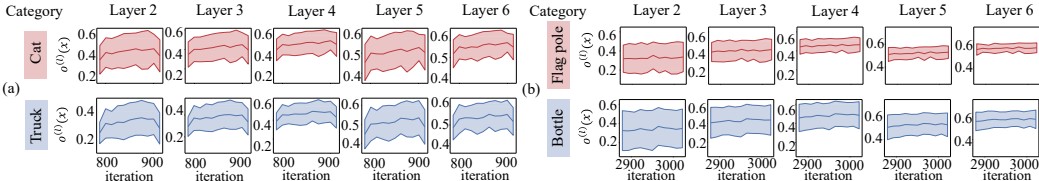

Figure 6: The change of $o^{(l)}$ in the first phase. We trained 9-layer MLPs on the (a) CIFAR-10 and the (b) Tiny ImageNet. Each layer of the MLP had 512 neurons. The Appendix D provides results on the MNIST. The shade represents the standard deviation over different samples.

Thus, Theorem 2 *explains how the significance of the common direction is enhanced by all training samples in the category c, i.e., $F_t^{(l-1)}$ and $\alpha_c V_t^{(l)}$ become increasingly similar.* We can consider $\cos(\alpha_c V_t^{(l)}, \Delta F_t^{(l-1)}|_x) \geq 0$ in Theorem 2 means that features of training samples in the same category $c$ are all pushed towards a common direction $\alpha_c V_t^{(l)}$, and make $\Delta F_t^{(l-1)}|_x$ highly similar to $\alpha_c V_t^{(l)}$, i.e., *making sample features $F_t^{(l-1)}|_x$ in the category c become increasingly similar to each other.* On the other hand, $\cos(\alpha_c \Delta V_t^{(l)}|_x, F_t^{(l-1)}|_x) \geq 0$ in Theorem 2 means that training samples in the category $c$ all push $V_t^{(l)}$ towards $\alpha_c \mathbb{E}_{x \in X_c}[F_t^{(l-1)}|_x]$, and make $\Delta V_t^{(l)}$ roughly parallel to $\alpha_c \mathbb{E}_{x \in X_c}[F_t^{(l-1)}|_x]$, i.e., *pushing weights of different neurons $V_t^{(l)}$ towards the average feature.* This phenomenon is verified in Figure 5, where $\cos(\alpha_c \Delta V_t^{(l)}, F_t^{(l-1)})$ is always positive over different samples of the same category. The above analysis also well explains the dynamics behind $\cos(\Delta V_t^{(l)}, F_t^{(l-1)}) \cdot \cos(V_t^{(l)}, \Delta F_t^{(l-1)}) \geq 0$ in Lemma 3.

**Lemma 3.** *(Proof in Appendix K) Given an input sample $x \in X$ and a common direction $C^{(l)}$ after the $t$-th iteration, if the noise term $\varepsilon_t^{(l)}$ is small enough to satisfy $|\Delta V_t^{(l)\top} F_t^{(l-1)} V_t^{(l)\top} V_t^{(l)} C^{(l)\top} C^{(l)} \Delta V_t^{(l)\top} F_t^{(l-1)}| \gg |\Delta V_t^{(l)\top} F_t^{(l-1)} V_t^{(l)\top} \varepsilon_t^{(l)} \Delta \varepsilon_t^{(l)\top} F_t^{(l-1)}|$, we can obtain $\cos(\Delta V_t^{(l)}, F_t^{(l-1)}) \cdot \cos(V_t^{(l)}, \Delta F_t^{(l-1)}) \geq 0$, where $\Delta V_t^{(l)} = \frac{\Delta W_t^{(l)\top} C^{(l)}}{C^{(l)\top} C^{(l)}}$, and $V_t^{(l)} = \frac{W_t^{(l)\top} C^{(l)}}{C^{(l)\top} C^{(l)}}$. $\Delta F_t^{(l-1)}$ denotes the change of features $\Delta F_t^{(l-1)} = F_{t+1}^{(l-1)} - F_t^{(l-1)}$ made by the training sample $x$ after the $t$-th iteration. To this end, we approximately consider the change of features $\Delta F_t^{(l-1)}$ after the $t$-th iteration negatively parallel to feature gradients $\dot{F}_t^{(l-1)}$, although strictly speaking, the change of features is not exactly equal to the feature gradients.*

**Theorem 2.** *(Proof in Appendix L) Under the aforementioned background assumption, for any training samples $x, x' \in X_c$ in the category c, if $[C^{(l)\top} D_t^{(l)}|_x \dot{F}_t^{(l)}|_x] \cdot [C^{(l)\top} D_t^{(l)}|_{x'} \dot{F}_t^{(l)}|_{x'}] > 0$ (i.e., $F_t^{(l)}|_x$ and $F_t^{(l)}|_{x'}$ have kinds of similarity in very early iterations), then $\cos(\alpha_c \Delta V_t^{(l)}|_x, F_t^{(l-1)}|_x) \geq 0$, and $\cos(\alpha_c V_t^{(l)}, \Delta F_t^{(l-1)}|_x) \geq 0$, where $\alpha_c \in \{-1, +1\}$ is a constant shared by all samples in category c.*

We conduct experiments to verify the relationship between the feature $F_t^{(l-1)}$ and the vector $V_t^{(l)}$. To this end, we measure the change of the value $o^{(l)} = \cos(\Delta V_t^{(l)}, F_t^{(l-1)}) \cdot \cos(V_t^{(l)}, \Delta F_t^{(l-1)})$. Figure 6 reports the average $o^{(l)}$ value over different samples at each iteration. For each sample $x$, $o^{(l)}$ is always positive and usually increases over iterations, which verifies Lemma 3. Besides, the assumption for a tiny $\varepsilon_t^{(l)}$ in Lemma 3 is verified by experimental results in Appendix K.

In sum, Step 1 focuses on the significance of the common direction enhanced by training samples in a certain category. In order to further analyze the learning effect of training samples from different categories, we propose Assumption 1 according to extensive experimental observations.

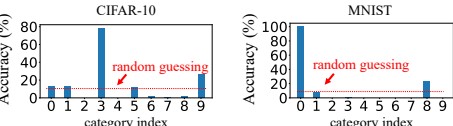

Figure 7: The training accuracy of MLPs on different categories at the end of the first phase. The MLP only learned features of a single or two categories in the first phase.

**Assumption 1.** *We assume that the MLP encodes features of very few (a single or two) categories in the first phase, instead of simultaneously learning all or most categories in this phase.*

Assumption 1 indicates that MLPs first learn a single or two categories in the first phase. Figure 7 verifies that only a single or two categories exhibit much higher accuracies than the random guessing at the end of the first phase. This means that the learning of the MLP is dominated by training samples of a single or two categories in very early iterations. Please see Appendix O for more results on different DNNs.

**Step 2: Extending the enhancement of the significance of the common direction to a more general case that considers all training samples from different categories**, *i.e.*, $F_t^{(l-1)}$ and $\alpha_{\hat{c}} V_t^{(l)}$ become increasingly similar. The overall learning dynamics in the first phase can be roughly described, by combining Theorem 2 and Assumption 1 as follows. Assumption 1 indicates that MLPs encode features of very few (a single or two) categories in early epochs. In other words, the overall learning effects of all training samples are dominated by very few categories $\hat{c}$. Based on this, Theorem 2 indicates two effects. First, features $F_t^{(l-1)}$ of different samples are all pushed towards the vector $\alpha_{\hat{c}} V_t^{(l)}$, where $\alpha_{\hat{c}}$ is determined by the dominating category/categories $\hat{c}$. Second, $V_t^{(l)}$ is pushed towards $\alpha_{\hat{c}} \mathbb{E}_{x \in X_{\hat{c}}}[F_t^{(l-1)}|_x]$. Therefore, features $F_t^{(l-1)}$ of different samples and $\alpha_{\hat{c}} V_t^{(l)}$ enhance each other, just like a "self-enhanced system." The "self-enhanced system" starts from from the assumed state that $\Delta F_t^{(l-1)}$ and $F_t^{(l-1)}$ of most samples have similar directions, and $\Delta V_t^{(l)}$ and $V_t^{(l)}$ have similar directions. In other words, the component along the common direction $C^{(l)} \Delta V_t^{(l)^\top}$ in $\Delta W_t^{(l)} \stackrel{.}{=} C^{(l)} \Delta V_t^{(l)^\top} + \Delta \varepsilon_t^{(l)^\top}$ will be further enhanced.

**Step 3: Explaining the increasing feature similarity and the increasing gradient similarity. *i.e.*, explaining the TFC phenomenon.** As aforementioned, features $F_t^{(l-1)}$ of different samples are consistently pushed towards the same vector $\alpha_{\hat{c}} V_t^{(l)}$. It increases the similarity between features of different samples $\mathbb{E}_{x,x' \in X}[\cos(F_t^{(l-1)}|_x, F_t^{(l-1)}|_{x'})]$ in the first phase. On the other hand, the increasing similarity between feature gradients can be also explained from two views. (1) The increasing feature similarity over different samples makes different training samples generate similar gating states $D_t^{(l)}$ in each ReLU layer. The increasing similarity of ReLU layers' gating states between different samples also increases the similarity of feature gradients between different samples in the same category $\mathbb{E}_{x,x' \in X_c}[\cos(\dot{F}_t^{(l-1)}|_x, \dot{F}_t^{(l-1)}|_{x'})]$. (2) Another view is that the component along the common direction $C^{(l)} V_t^{(l)^\top}$ in $W_t^{(l)}$ is enhanced in the first phase. Because $C^{(l)}$ denotes the principle weight direction of the $i$-th column $w_{t,i}^{(l)}$ of $W_t^{(l)}$, *each weight vector $w_{t,i}^{(l)}$ is optimized towards the common direction $C^{(l)}$. Eq. (3) shows that the increasing cosine similarity between $w_{t,i}^{(l)}$ and $C^{(l)}$ for all weight vectors will boost the similarity between feature gradients of different samples.*

**Vanishing gradients on correctly classified samples destroy the "self-enhanced system."** All our explanation focuses on the early epochs of training, when only a few training samples of one or two dominating categories can be confidently classified. However, when the optimization of a single or two dominating categories in the first phase soon saturates at the end of the first phase, gradients on the correctly classified samples of the dominating categories vanish. Then, gradients from training samples of other categories weaken the dominating role of a single or two categories in the learning of the MLP. Thus, the "self-enhanced system" is destroyed, and the learning of the MLP enters the second phase.

### 3.3 THEORETICALLY ALLEVIATING THE TFC PHENOMENON

In previous sections, we have discovered and explained a fundamental yet counter-intuitive TFC phenomenon with the MLP. This is the distinctive contribution of this study, which has not been theoretically explained for a long time. Besides, we find that we can use the above findings to explain that four typical operations can usually alleviate or strengthen the TFC phenomenon, *i.e.*, normalization, momentum, initialization, and $L_2$ regularization. Although these operations have

Figure 8: Effects of (a) normalization and (b) initialization. We trained $L$-layer MLPs, where each layer had 512 neurons. A shorter first phase indicates that the TFC phenomenon is more alleviated. Effects of momentum and $L_2$ regularization are shown in Appendix M.2.

been widely used, previous studies failed to theoretically explain their effectiveness. To this end, our analysis can explain a high likelihood for such operations to affect the TFC phenomenon, although it is not a proof of a strict sufficient condition or a necessary condition for the TFC phenomenon.

**Centering operations for normalization.** Based on theoretical analysis, we explain that the centering operation in normalization operations (*e.g.*, that in batch normalization (BN)) can alleviate the TFC phenomenon in the first phase. Specifically, according to Theorem 2, the "self-enhanced system" of decreasing feature diversity requires features $F_t^{(l)}$ of any two training samples $x$ and $x'$ in the same category to be similar to each other. However, the centering operation prevents features $F_t^{(l)}$ of different samples from being similar to each other, because it subtracts the mean feature $\bar{F}_t^{(l)} = \mathbb{E}_{x \in X}[F_t^{(l)}|_x]$ from features of all samples, *i.e.*, $F_t^{'(l)}|_x = F_t^{(l)}|_x - \bar{F}_t^{(l)}$. Therefore, the dissimilarity between features of different samples breaks the "self-enhanced system." Please see Appendix M.1 for more discussions.

We conducted experiments to verify the above analysis. We compared MLPs trained with and without BN layers. Specifically, we added a BN layer after each linear layer to construct MLPs. Figure 8(a) shows that the feature similarity in MLPs with BN layers kept decreasing. This verified that BN layers alleviated the TFC phenomenon.

**Momentum.** Our theorems explain that momentum in gradient descent can alleviate the TFC phenomenon. Based on Lemma 3, the "self-enhanced system" of the decreasing of feature diversity requires weights along other directions $\varepsilon_t^{(l)}$ to be small enough. However, because the momentum operation strengthens influences of the initialized noisy weights $W_{t=0}^{(l)}$, it strengthens singular values of $\varepsilon_t^{(l)}$, to some extent, thereby alleviating the TFC phenomenon. Specifically, a larger momentum coefficient usually more alleviates the TFC phenomenon. To this end, we trained MLPs with different momentum coefficients, and experimental results in Appendix M.2 verified the above analysis.

**Initialization.** We explain that the initialization of MLPs affects the TFC phenomenon. According to Lemma 3, the "self-enhanced system" requires very small weights along noise directions $\varepsilon_t^{(l)}$. However, increasing the variance of the initialized weights $W_{t=0}^{(l)}$ can boost singular values of $\varepsilon_t^{(l)}$, which alleviates the TFC phenomenon. Please see Appendix M.3 for more discussions.

To verify the above claim, we conducted experiments by comparing MLPs trained using different initializations with different variances. We used $\gamma$ to control the variance of the initialization, *i.e.*, $W_{t=0}^{(l)} \sim \mathcal{N}(\mathbf{0}, \gamma \sigma_{\text{var}}^2 I)$, where $\sigma_{\text{var}}$ is a constant computed following (Glorot & Bengio, 2010). Figure 8(b) verifies that the initialization with a large variance alleviated the TFC phenomenon.

$L_2$ **regularization (ridge loss).** We also explain that the $L_2$ regularization (the ridge loss) can strengthen the TFC phenomenon. The total loss is given as $\mathcal{L}(W_t) = \mathcal{L}^{CE}(W_t) + \lambda \|W_t\|_2^2$, where $\mathcal{L}^{CE}(W_t)$ represents the cross entropy loss, and $\lambda \|W_t\|_2^2$ denotes the ridge loss. As aforementioned, the TFC phenomenon requires singular values of $\varepsilon_t^{(l)}$ to be small enough. However, because the loss of $\|W_t\|_2^2$ penalizes singular values of $\varepsilon_t^{(l)}$, it strengthens the TFC phenomenon. The experimental verification is provided in Appendix M.4.

## 4  CONCLUSION

In this paper, we find that in the early stage of the training process, the MLP exhibits a fundamental yet counter-intuitive TFC phenomenon, *i.e.*, the feature diversity keeps decreasing in the first phase. We explain this phenomenon by analyzing the learning dynamics of the MLP. Furthermore, we explain the reason why four typical operations can alleviate the TFC phenomenon.

ETHIC STATEMENT

we focus on a typical two-phase phenomenon in the learning of multi-layer perceptrons (MLPs). We discover and explain the reason for the feature collapse phenomenon in the first phase, *i.e.*, the diversity of features over different samples keeps decreasing in the first phase, until samples of different categories share almost the same feature, which hurts the optimization of MLPs. There are no ethic issues with this paper.

REPRODUCIBILITY STATEMENT

We have provided the proof for all theoretical results in Appendix E, Appendix F, Appendix H.1, Appendix K, and Appendix L. We have also provided experimental details in Appendix B. The code has been attached with the submission.

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

# A    RELATED WORK

Understanding the optimization and the representation capacity of DNNs is an important direction to explain DNNs. The information bottleneck theory (Wolchover, 2017; Shwartz-Ziv & Tishby, 2017) quantitatively explained the information encoded by features in intermediate layers of DNNs. Xu & Raginsky (2017), Achille & Soatto (2018), and Cheng et al. (2018) used the information bottleneck theory to evaluate and improve the DNN's representation capacity. Arpit et al. (2017) analyzed the representation capacity of DNNs with real training data and noises. In addition, several metrics were proposed to measure the generalization capacity or robustness of DNNs, including the stiffness (Fort et al., 2019), the sensitivity metrics (Novak et al., 2018), the Fourier analysis (Xu, 2018), and the CLEVER score (Weng et al., 2018). In comparison, we explain the MLP from the perspective of the learning dynamics, *i.e.*, we explain the TFC phenomenon in early iterations of the MLP.

Analyzing the learning dynamics is another perspective to understand DNNs. Many studies analyzed the local minima in the optimization landscape of linear networks (Baldi & Hornik, 1989; Saxe et al., 2013; Hardt & Ma, 2016; Daniely et al., 2016) and nonlinear networks (Choromanska et al., 2015; Kawaguchi, 2016; Safran & Shamir, 2018). Some studies discussed the convergence rate of gradient descent on separable data (Soudry et al., 2018; Xu et al., 2018; Nacson et al., 2019). Hoffer et al. (2017) and Jastrzębski et al. (2017) have investigated the effects of the batch size and the learning rate on SGD dynamics. In addition, some studies analyzed the dynamics of gradient descent in the overparameterization regime (Arora et al., 2018; Jacot et al., 2018; Lee et al., 2018; Du et al., 2018). Besides, (Papyan et al., 2020; Han et al., 2021) explored the neural collapse phenomenon, which was observed at the end of the training stage. Unlike previous studies, we analyze the learning dynamics of features and weights of the MLP, in order to explain the TFC phenomenon in the early training process of the MLP.

# B COMMON PHENOMENON SHARED BY DIFFERENT DNNS FOR DIFFERENT TASKS.

In this section, we aim to demonstrate an interesting phenomenon of the decrease of the feature diversity when we train an MLP in early iterations. Specifically, the training process of the MLP can usually be divided into the following two phases according to the training loss. In the first phase, the training loss does not decrease significantly, and the training loss suddenly begins to decrease in the second phase.

The two-phase phenomenon of the training loss is well-known, because many previous studies (Simsekli et al., 2019; Saxe et al., 2013; Vogl, 2018; Nguyen et al., 2018; Arab et al., 2020; Jepkoech et al., 2021; Stevens et al., 2020) have shown this phenomenon during the training process in their papers. However, previous studies did not theoretically explain the emergence of such a phenomenon. Instead, they usually understood this phenomenon in an intuitive manner, *i.e.*, initialized DNNs failed to find a clear optimization direction, and thus these DNNs usually spent a long time searching for a reliable optimization direction. In this way, the training loss did not decrease significantly in very early epochs of training.

More crucially, the feature diversity decreases in the first phase. This phenomenon is widely shared by different DNNs with different architectures for different tasks. As Figure 1, Figure 2, and Figure 3 show, the feature diversity keeps decreasing (*i.e.*, the cosine similarity between features of different samples keeps increasing) until samples of different categories share almost the same feature in the first phase. We can consider this as the temporary feature collapse (TFC). This TFC happens in various DNNs, including multi-layer perceptrons (MLPs), convolutional neural networks, and recurrent neural networks. DNNs trained with different loss functions and different learning rates may all exhibit TFC phenomenon. Specifically, we calculated the feature cosine similarity between fifty samples from ten categories on the CIFAR-10 dataset, the MNIST dataset, and the Tiny ImageNet dataset. The abscissa and ordinate of each heatmap represent the sample index. For each grid, color indicates the cosine similarity of that sample pair. Note that all the features are extracted after the ReLU layer. Thus, the cosine similarity is always greater than zero.

Besides, as Figure 1 in the main paper shows, samples from different categories share diverse features in the beginning of the training, but share almost the same feature at the end of the training. Specifically, we used t-SNE for visualization (initialized by PCA).

Let us take the 9-layer MLP trained on the CIFAR-10 dataset for an example, where each layer of the MLP had 512 neurons. As Figure 4(e)(f) shows, before the 1300-th iteration (the first phase), both the feature diversity and the gradient diversity kept decreasing, *i.e.*, both the cosine similarity between features over different samples and the cosine similarity between gradients kept increasing. After the 1300-th iteration (the second phase), the feature diversity and the gradient diversity suddenly began to increase, *i.e.* their similarities began to decrease. Therefore, the MLP had the lowest feature diversity and the lowest gradient diversity at around the 1300-th iteration. Specifically, the training loss was evaluated on the whole training set.

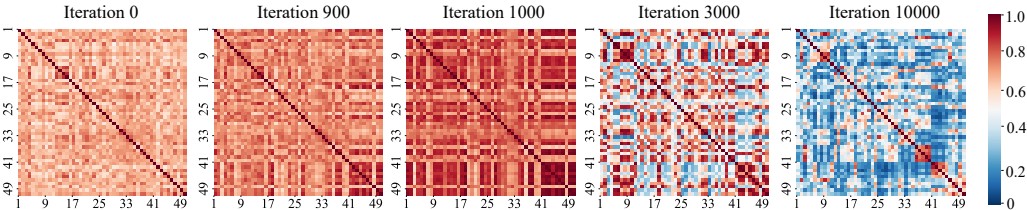

Figure 1: Cosine similarity between features of different samples on the CIFAR-10 dataset. We trained a 9-layer MLP, where each layer had 512 neurons. The cosine similarity between features of different samples kept increasing until samples of different categories share almost the same feature in the first phase. The features were used in the fourth linear layer of the MLP. The TFC phenemonon happens in the 1000-th iteration. The abscissa and ordinate of each heatmap represent the sample index. For each grid, color indicates the cosine similarity of that sample pair.

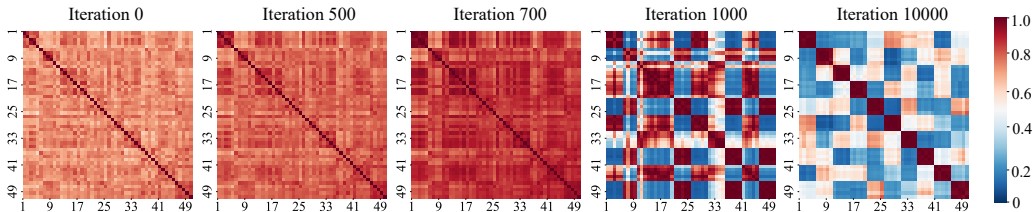

Figure 2: Cosine similarity between features of different samples on the MNIST dataset. We trained a 9-layer MLP, where each layer had 512 neurons. The cosine similarity between features of different samples kept increasing until samples of different categories share almost the same feature in the first phase. The features were used in the fourth linear layer of the MLP. The TFC phenemonon happens in the 700-th iteration. The abscissa and ordinate of each heatmap represent the sample index. For each grid, color indicates the cosine similarity of that sample pair.

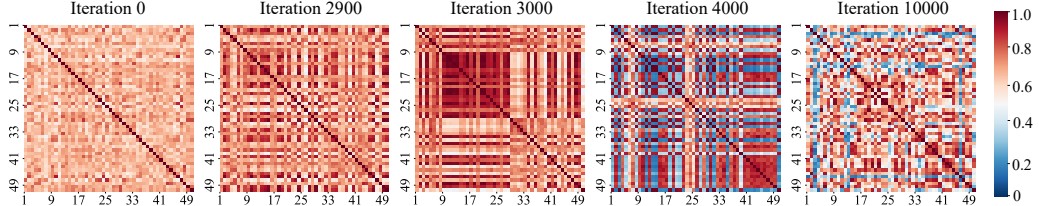

Figure 3: Cosine similarity between features of different samples on the Tiny ImageNet dataset. We trained a 9-layer MLP, where each layer had 512 neurons. The cosine similarity between features of different samples kept increasing until samples of different categories share almost the same feature in the first phase. The features were used in the fourth linear layer of the MLP. The TFC phenemonon happens in the 3000-th iteration. The abscissa and ordinate of each heatmap represent the sample index. For each grid, color indicates the cosine similarity of that sample pair.

### B.1 ON THE CIFAR-10 DATASET

In this subsection, we demonstrated that the two-phase phenomenon was shared by different MLPs on the CIFAR-10 dataset (Krizhevsky et al., 2009). For different MLPs, we adopted the learning rate $\eta = 0.1$, the batch size $bs = 100$, the SGD optimizer, and the ReLU activation function. Besides, we used two data augmentation methods, including random cropping and random horizontal flipping. The training loss, the testing loss, the training accuracy, the testing accuracy, the cosine similarity of features, and the cosine similarity of feature gradients of MLPs trained on the CIFAR-10 dataset are shown in Figure 4.

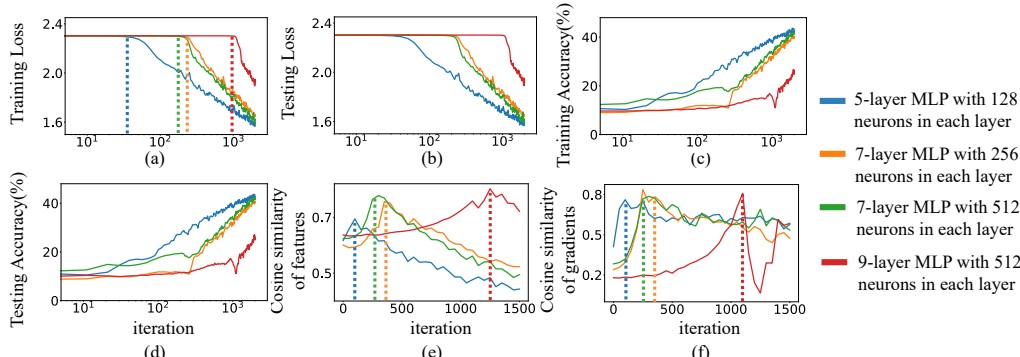

Figure 4: (a) The training loss of four MLPs trained on the CIFAR-10 dataset. (b) The testing loss of four MLPs. (c) Training accuracies of four MLPs. (d) Testing accuracies of four MLPs. (e) Cosine similarity between features of different categories. (f) Cosine similarity between gradients of different samples in a category. The feature and the feature gradient were used in the third linear layer of MLPs.

## B.2    ON THE MNIST DATASET

In this subsection, we demonstrated that the two-phase phenomenon was shared by different MLPs on the MNIST dataset (LeCun et al., 1998). For different MLPs, we adopted the learning rate $\eta = 0.01$, the batch size $bs = 100$, the SGD optimizer, and the ReLU activation function. The training loss, the testing loss, the training accuracy, the testing accuracy, the cosine similarity of features, and the cosine similarity of feature gradients of MLPs trained on the MNIST are shown in Figure 5.

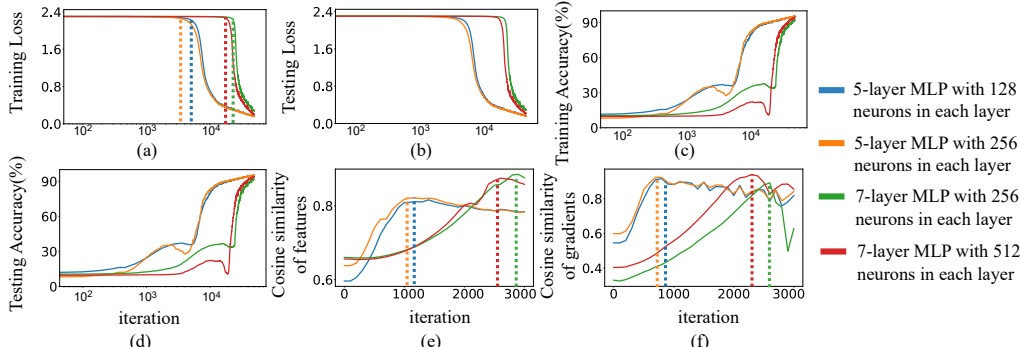

Figure 5: (a) The training loss of four MLPs tranined on the MNIST dataset. (b) The testing loss of four MLPs. (c) Training accuracies of four MLPs. (d) Testing accuracies of four MLPs. (e) Cosine similarity between features of different categories. (f) Cosine similarity between gradients of different samples in a category. The feature and the feature gradient were used in the third linear layer of MLPs.

## B.3    ON THE TINY IMAGENET DATASET

In this subsection, we demonstrated that the two-phase phenomenon was shared by different MLPs on the Tiny ImageNet dataset (Le & Yang, 2015). Specifically, we randomly selected the following 50 categories, *orangutan, parking meter, snorkel, American alligator, oboe, basketball, rocking chair, hopper, neck brace, candy store, broom, seashore, sewing machine, sunglasses, panda, pretzel, pig, volleyball, puma, alp, barbershop, ox, flagpole, lifeboat, teapot, walking stick, brain coral, slug, abacus, comic book, CD player, school bus, banister, bathtub, German shepherd, black stork, computer keyboard, tarantula, sock, Arabian camel, bee, cockroach, cannon, tractor, cardigan, suspension bridge, beer bottle, viaduct, guacamole*, and *iPod* for training. For different MLPs, we adopted the learning rate $\eta = 0.1$, the batch size $bs = 100$, the SGD optimizer, and the ReLU activation function. Besides, we used two data augmentation methods, including random cropping and random horizontal flipping. Note that we took a random cropping with $32{\times}32$ sizes.The training loss, the testing loss, the training accuracy, the testing accuracy, the cosine similarity of features, and the cosine similarity of feature gradients of MLPs trained on the Tiny ImageNet are shown in Figure 6.

## B.4    ON THE CENSUS DATASET

In this subsection, we demonstrated that the two-phase phenomenon was shared by different MLPs on the UCI census income tabular dataset (Census) (Asuncion & Newman, 2007). For different MLPs, we adopted the learning rate $\eta = 0.1$, the batch size $bs = 1000$, the SGD optimizer, and the ReLU activation function. The training loss, the testing loss, the training accuracy, the testing accuracy, the cosine similarity of features, and the cosine similarity of feature gradients of MLPs trained on the census are shown in Figure 7.

## B.5    ON THE COMMERCIAL DATASET

In this subsection, we demonstrated that the two-phase phenomenon was shared by different MLPs on the UCI TV news channel commercial detection dataset (Commercial) (Asuncion & Newman, 2007). For different MLPs, we adopted the learning rate $\eta = 0.1$, the batch size $bs = 1000$, the

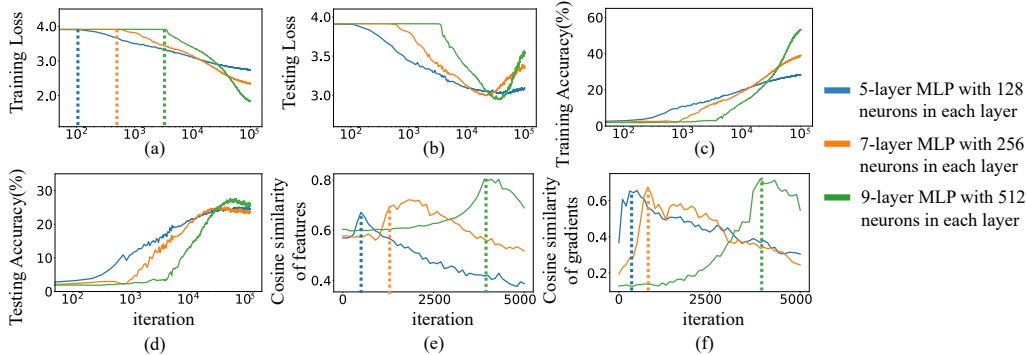

Figure 6: (a) The training loss of three MLPs tranined on the Tiny ImageNet dataset. (b) The testing loss of three MLPs. (c) Training accuracies of three MLPs. (d) Testing accuracies of three MLPs. (e) Cosine similarity between features of different categories. (f) Cosine similarity between gradients of different samples in a category. The features and the feature gradient were used in the second linear layer of MLPs.

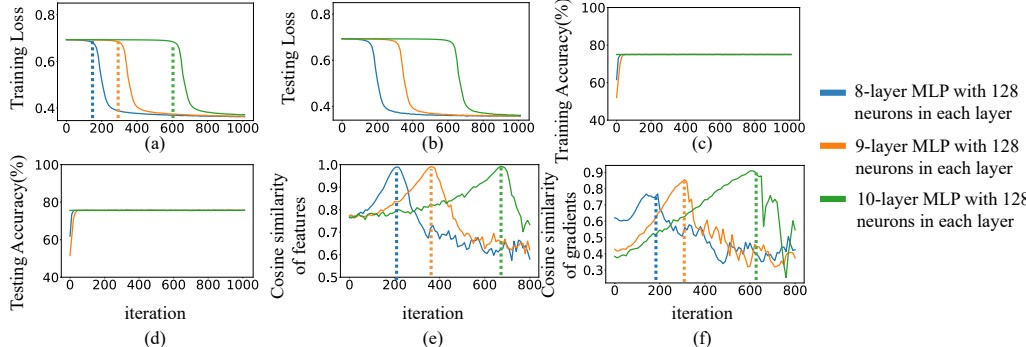

Figure 7: (a) The training loss of three MLPs trained on the Census dataset. (b) The testing loss of three MLPs. (c) Training accuracies of three MLPs. (d) Testing accuracies of three MLPs. (e) Cosine similarity between features of different categories. (f) Cosine similarity between gradients of different samples in a category. The feature and the feature gradient were used in the fifth linear layer of MLPs.

SGD optimizer, and the ReLU activation function. The training loss, the testing loss, the training accuracy, the testing accuracy, the cosine similarity of features, and the cosine similarity of feature gradients of MLPs trained on the census are shown in Figure 8.

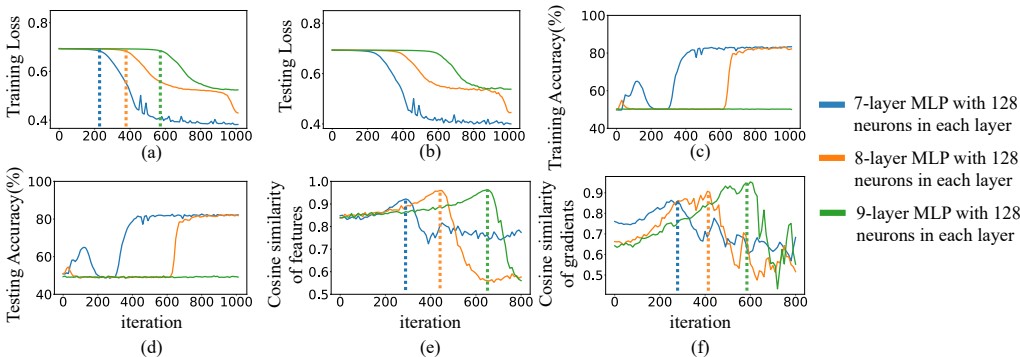

Figure 8: (a) The training loss of three MLPs trained on the Commercial dataset. (b) The testing loss of three MLPs. (c) Training accuracies of three MLPs. (d) Testing accuracies of three MLPs. (e) Cosine similarity between features of different categories. (f) Cosine similarity between gradients of different samples in a category. The feature and the feature gradient were used in the fifth linear layer of MLPs.

### B.6 ON THE COLA DATASET

In this subsection, we demonstrated that the two-phase phenomenon was shared by the revised LSTMs on the CoLA dataset (Warstadt et al., 2019). We used two-layer unidirectional LSTMs concatenated with MLPs. Specifically, we trained two LSTMs with 5-layer MLPs, where each layer of the MLP had 256 and 512 neurons. We adopted the learning rate $\eta = 0.1$, the batch size $bs = 1000$, the SGD optimizer, and the ReLU activation function. The training loss, the testing loss, the training accuracy, the testing accuracy, the cosine similarity of features, and the cosine similarity of feature gradients of LSTMs trained on the CoLA are shown in Figure 9. Since training samples in the CoLA dataset were imbalanced, we constructed a new training set by randomly sampling 2000 training samples from two categories, respectively. DNNs were trained on this new training set.

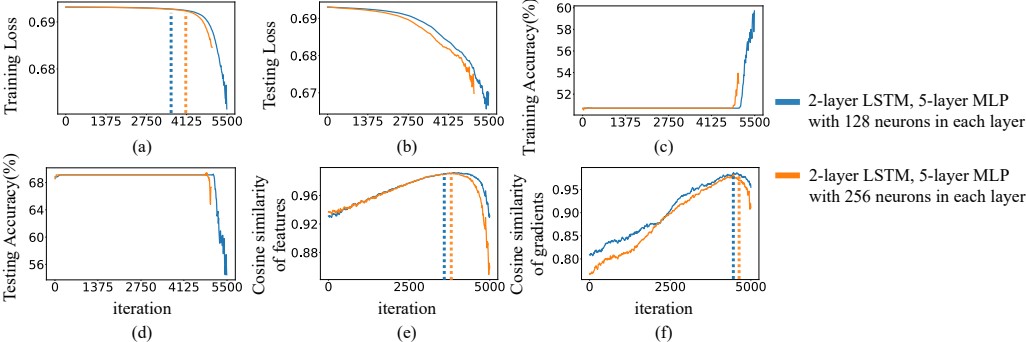

Figure 9: (a) The training loss of two LSTMs trained on the CoLA dataset. (b) The testing loss of two LSTMs. (c) Training accuracies of two LSTMs. (d) Testing accuracies of two LSTMs. (e) Cosine similarity between features of different categories. (f) Cosine similarity between gradients of different samples in a category. The feature and the feature gradient were used in the third linear layer of MLPs.

### B.7 ON THE SST-2 DATASET

In this subsection, we demonstrated that the two-phase phenomenon was shared by the revised LSTMs on the SST-2 dataset (Socher et al., 2013). We used unidirectional LSTMs concatenated with MLPs. Specifically, we trained three LSTMs with 4-layer MLPs, 4-layer MLPs, and 5-layer MLPs, respectively, where each layer of the MLP had 32, 64, 128 neurons. We adopted the learning rate $\eta = 0.1$, the batch size $bs = 500$, the SGD optimizer, and the ReLU activation function. Since the training of LSTMs on the SST-2 with the SGD optimizer is unstable, we randomly selected 15000 training samples from the training set. We trained LSTMs on these 15000 training samples. The training loss, the testing loss, the training accuracy, the testing accuracy, the cosine similarity of features, and the cosine similarity of feature gradients of LSTMs trained on the SST-2 are shown in Figure 10.

### B.8 ON THE AGNEWS DATASET

In this subsection, we demonstrated that the two-phase phenomenon was shared by the revised LSTMs on the AGNEWS dataset. We used two-layer unidirectional LSTMs concatenated with MLPs. Specifically, we trained three LSTMs with 4-layer MLPs, 4-layer MLPs, 5-layer MLPs, respectively, where each layer of the MLP had 32, 64, and 128 neurons, respectively. We adopted the learning rate $\eta = 0.1$, the batch size $bs = 500$, the SGD optimizer, and the ReLU activation function. The training loss, the testing loss, the training accuracy, the testing accuracy, the cosine similarity of features, and the cosine similarity of feature gradients of LSTMs trained on the AGNEWS are shown in Figure 11.

### B.9 DIFFERENT TRAINING BATCH SIZES

In this subsection, we demonstrated that the two-phase phenomenon was shared by MLPs trained on the CIFAR-10 dataset with different training batch sizes. For different MLPs, we adopted the learning rate $\eta = 0.1$, the SGD optimizer, and the ReLU activation function. Besides, we used

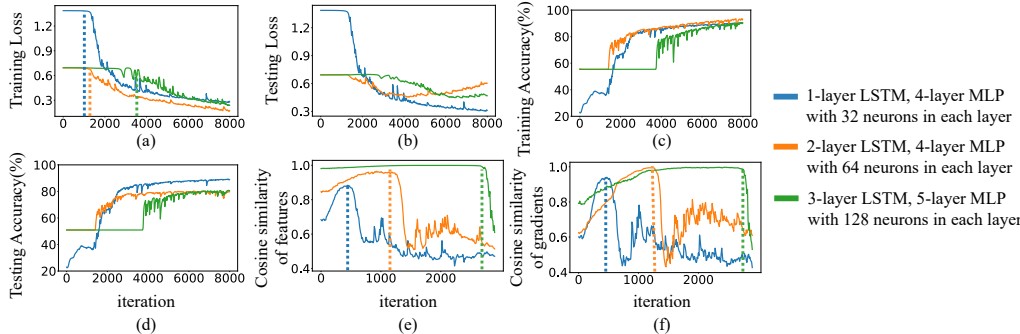

Figure 10: (a) The training loss of three LSTMs trained on the SST-2 dataset. (b) The testing loss of three LSTMs. (c) Training accuracies of three LSTMs. (d) Testing accuracies of three LSTMs. (e) Cosine similarity between features of different categories. (f) Cosine similarity between gradients of different samples in a category. The feature and the feature gradient were used in the second linear layer of MLPs.

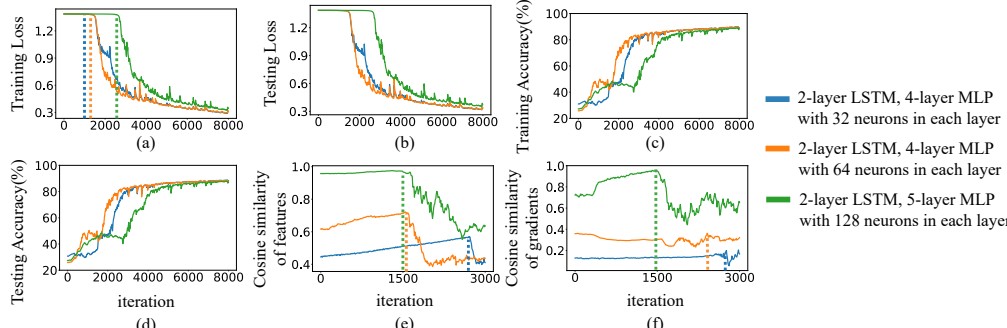

Figure 11: (a) The training loss of three LSTMs trained on the AGNEWS dataset. (b) The testing loss of three LSTMs. (c) Training accuracies of three LSTMs. (d) Testing accuracies of three LSTMs. (e) Cosine similarity between features of different categories. (f) Cosine similarity between gradients of different samples in a category. The feature and the feature gradient were used in the second linear layer of MLPs.

two data augmentation methods, including random cropping and random horizontal flipping. We trained three 7-layer MLPs with 256 neurons in each layer, with $bs = 100, 500, 1000$ respectively. The training loss, the testing loss, the training accuracy, the testing accuracy, the cosine similarity of features, and the cosine similarity of feature gradients of MLPs trained with different batch sizes are shown in Figure 12.

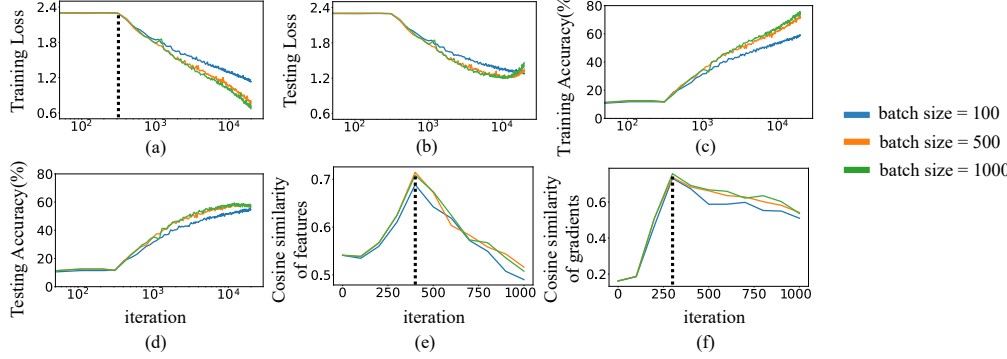

Figure 12: (a) The training loss of three MLPs trained with different batch sizes. (b) The testing loss of three MLPs. (c) Training accuracies of three MLPs. (d) Testing accuracies of three MLPs. (e) Cosine similarity between features of different categories. (f) Cosine similarity between gradients of different samples in a category. The feature and the feature gradient were used in the second linear layer of MLPs.

## B.10 DIFFERENT LEARNING RATES

In this subsection, we demonstrated that the two-phase phenomenon was shared by MLPs trained on the CIFAR-10 dataset with different learning rates. For different MLPs, we adopted the batch size $bs = 100$, the SGD optimizer, and the ReLU activation function. Besides, we used two data augmentation methods, including random cropping and random horizontal flipping. We trained two 7-layer MLPs with 256 neurons in each layer, with learning rates $\eta = 0.1, 0.01$ respectively. The training loss, the testing loss, the training accuracy, the testing accuracy, the cosine similarity of features, and the cosine similarity of feature gradients of MLPs trained with different learning rates are shown in Figure 13.

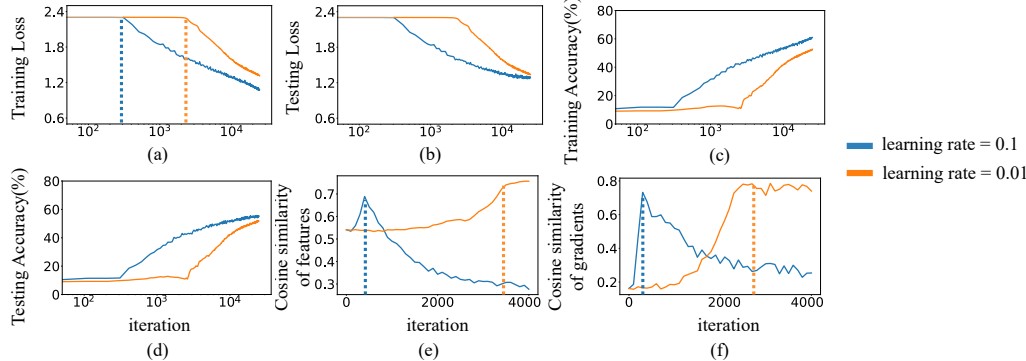

Figure 13: (a) The training loss of two MLPs trained with different learning rates. (b) The testing loss of two MLPs. (c) The training accuracies of two MLPs. (d) The testing accuracies of two MLPs. (e) Cosine similarity between features of different categories. (f) Cosine similarity between gradients of different samples in a category. The feature and the feature gradient were used in the second linear layer of MLPs.

## B.11 DIFFERENT ACTIVATION FUNCTIONS

In this subsection, we demonstrated that the two-phase phenomenon was shared by MLPs with different activation functions. For different MLPs, we adopted the learning rate $\eta = 0.1$, the batch size $bs = 100$, and the SGD optimizer. Besides, we used two data augmentation methods, including random cropping and random horizontal flipping. We trained three 9-layer MLPs with 512 neurons in each layer with the ReLU activation function, the Leaky ReLU (slope=0.1) activation function, and the Leaky ReLU (slope=0.01) activation function, respectively. The training loss, the testing loss, the training accuracy, the testing accuracy, the cosine similarity of features, and the cosine similarity of feature gradients of MLPs trained with different activation functions are shown in Figure 14.

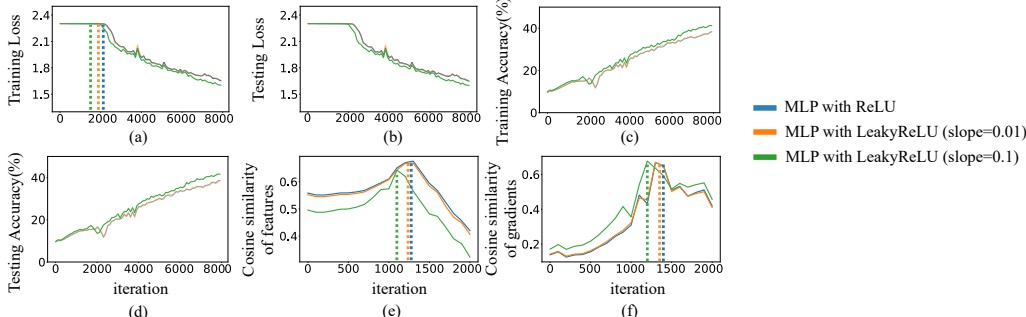

Figure 14: (a) The training loss of three MLPs with different activation functions. (b) The testing loss of three MLPs. (c) Training accuracies of three MLPs. (d) Testing accuracies of three MLPs. (e) Cosine similarity between features of different categories. (f) Cosine similarity between gradients of different samples in a category. The feature and the feature gradient were used in the second linear layer of MLPs.

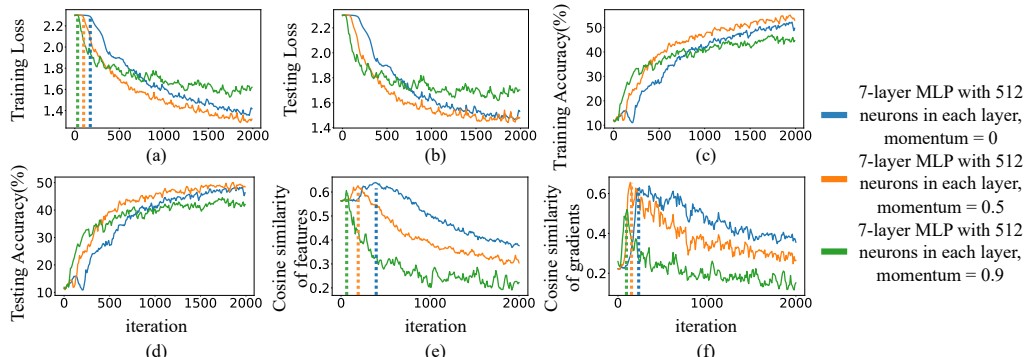

Figure 15: (a) The training loss of three MLPs with different momentums trained on the CIFAR-10 dataset. (b) The testing loss of three MLPs. (c) Training accuracies of three MLPs. (d) Testing accuracies of three MLPs. (e) Cosine similarity between features of different categories. (f) Cosine similarity between gradients of different samples in a category. The feature and the feature gradient were used in the second linear layer of MLPs.

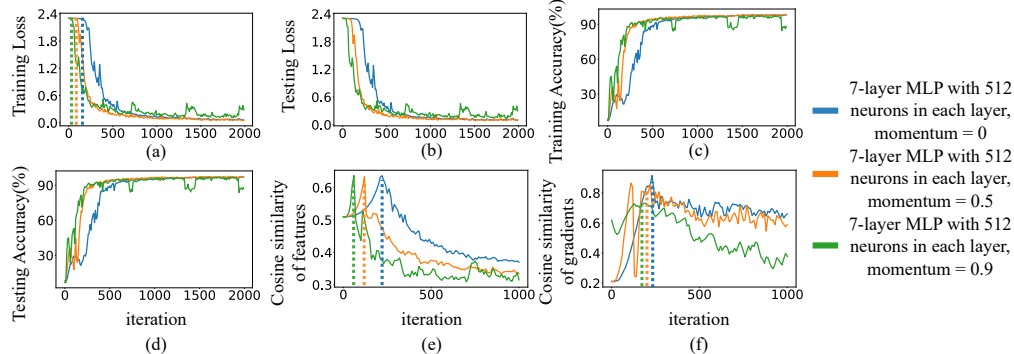

Figure 16: (a) The training loss of three MLPs with different momentums trained on the MNIST datasets. (b) The testing loss of three MLPs. (c) Training accuracies of three MLPs. (d) Testing accuracies of three MLPs. (e) Cosine similarity between features of different categories. (f) Cosine similarity between gradients of different samples in a category. The feature and the feature gradient were used in the second linear layer of MLPs.

## B.12   DIFFERENT MOMENTUMS

In this subsection, we demonstrated that the two-phase phenomenon was shared by MLPs trained on the CIFAR-10, MNIST and Tiny ImageNet dataset with different momentums. For different MLPs, we adopted the learning rate $\eta = 0.1$, the batch size $bs = 100$, and the SGD optimizer. Besides, we used two data augmentation methods, including random cropping and random horizontal flipping. We trained 7-layer MLPs and 9-layer MLPs with 512 neurons in each layer with the ReLU activation function. The training loss, the testing loss, the training accuracy, the testing accuracy, the cosine similarity of features, and the cosine similarity of feature gradients of MLPs trained with different momentum are shown in Figure 15, Figure 16, Figure 17, Figure 18, Figure 19, and Figure 20, respectively.

## B.13   DIFFERENT WEIGHT DECAYS

In this subsection, we demonstrated that the two-phase phenomenon was shared by MLPs trained on the CIFAR-10, MNIST and Tiny ImageNet dataset with different weight decays. For different MLPs, we adopted the learning rate $\eta = 0.1$, the batch size $bs = 100$, and the SGD optimizer. Besides, we used two data augmentation methods, including random cropping and random horizontal flipping. We trained 7-layer MLPs and 9-layer MLPs with 512 neurons in each layer with the ReLU activation function. The training loss, the testing loss, the training accuracy, the testing accuracy, the cosine similarity of features, and the cosine similarity of feature gradients of MLPs trained with different

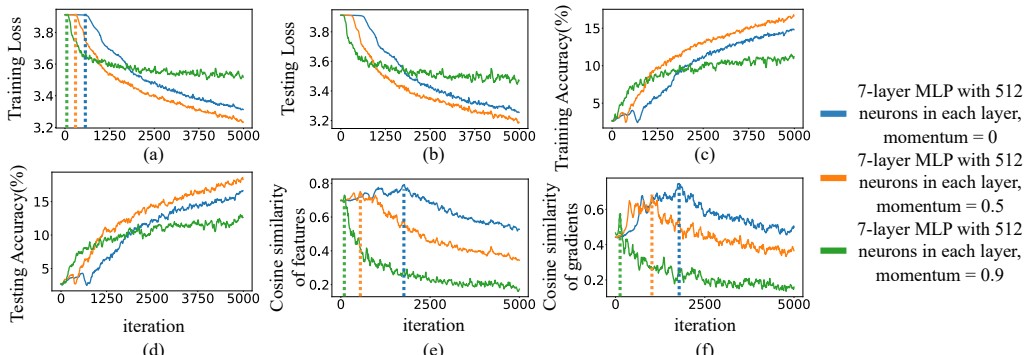

Figure 17: (a) The training loss of three MLPs with different momentums trained on the Tiny ImageNet dataset. (b) The testing loss of three MLPs. (c) Training accuracies of three MLPs. (d) Ttesting accuracies of three MLPs. (e) Cosine similarity between features of different categories. (f) Cosine similarity between gradients of different samples in a category. The feature and the feature gradient were used in the fourth linear layer of MLPs.

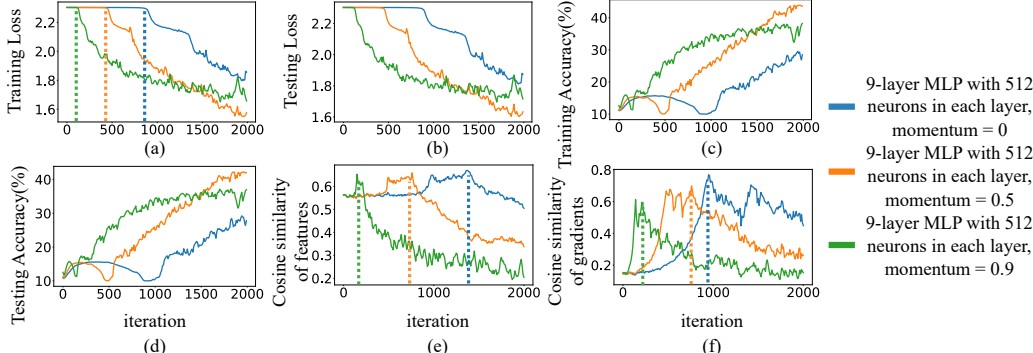

Figure 18: (a) The training loss of three MLPs with different momentums trained on the CIFAR-10 dataset. (b) The testing loss of three MLPs. (c) Training accuracies of three MLPs. (d) Testing accuracies of three MLPs. (e) Cosine similarity between features of different categories. (f) Cosine similarity between gradients of different samples in a category. The feature and the feature gradient were used in the second linear layer of MLPs.

weight decays are shown in Figure 21, Figure 22, Figure 23, Figure 24, Figure 25, and Figure 26, respectively.

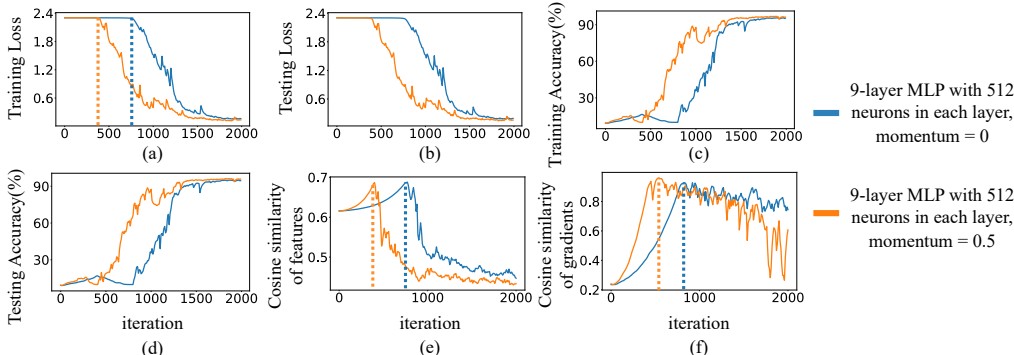

Figure 19: (a) The training loss of two MLPs with different momentums trained on the MNIST dataset. (b) The testing loss of two MLPs. (c) Training accuracies of two MLPs. (d) Testing accuracies of two MLPs. (e) Cosine similarity between features of different categories. (f) Cosine similarity between gradients of different samples in a category. The feature and the feature gradient were used in the second linear layer of MLPs.

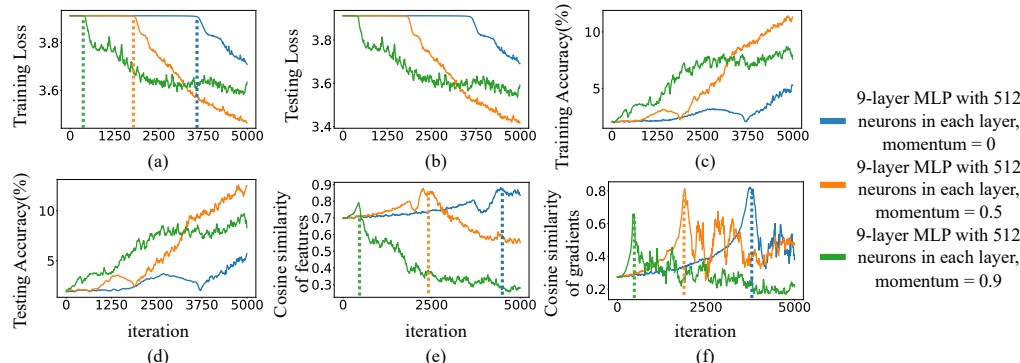

Figure 20: (a) The training loss of three MLPs with different momentums trained on the Tiny ImageNet dataset. (b) The testing loss of three MLPs. (c) Training accuracies of three MLPs. (d) Testing accuracies of three MLPs. (e) Cosine similarity between features of different categories. (f) Cosine similarity between gradients of different samples in a category. The feature and the feature gradient were used in the fourth linear layer of MLPs.

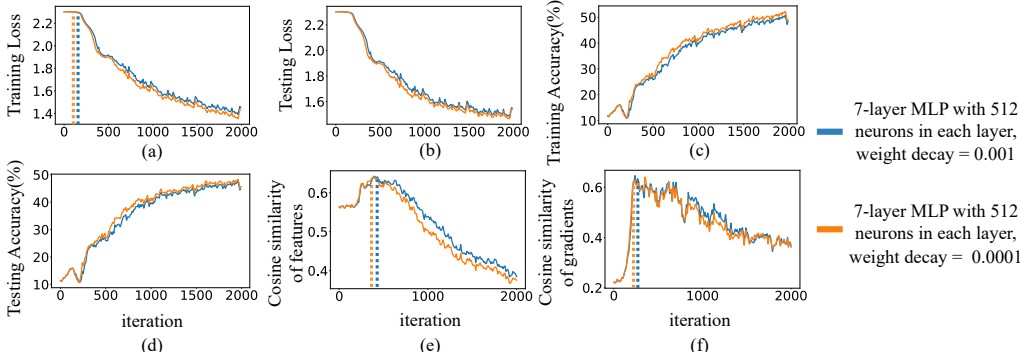

Figure 21: (a) The training loss of two MLPs with different weight decays trained on the CIFAR-10 dataset. (b) The testing loss of two MLPs. (c) Training accuracies of two MLPs. (d) Testing accuracies of two MLPs. (e) Cosine similarity between features of different categories. (f) Cosine similarity between gradients of different samples in a category. The feature and the feature gradient were used in the second linear layer of MLPs.

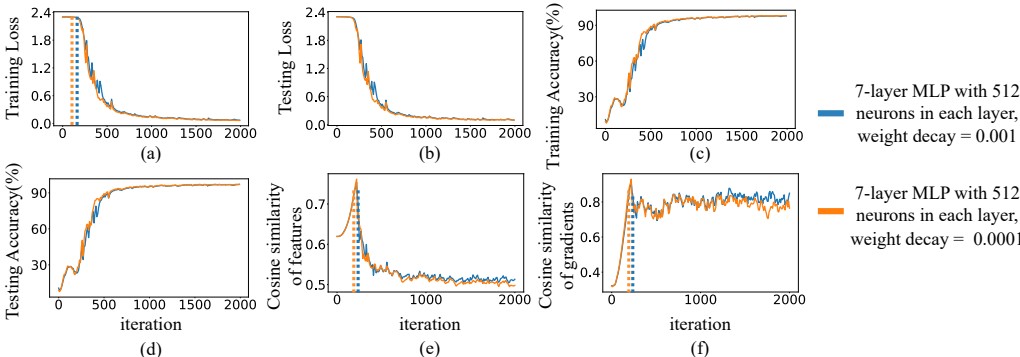

Figure 22: (a) The training loss of two MLPs with different weight decays trained on the MNIST dataset. (b) The testing loss of two MLPs. (c) Training accuracies of two MLPs. (d) Testing accuracies of two MLPs. (e) Cosine similarity between features of different categories. (f) Cosine similarity between gradients of different samples in a category. The feature and the feature gradient were used in the second linear layer of MLPs.

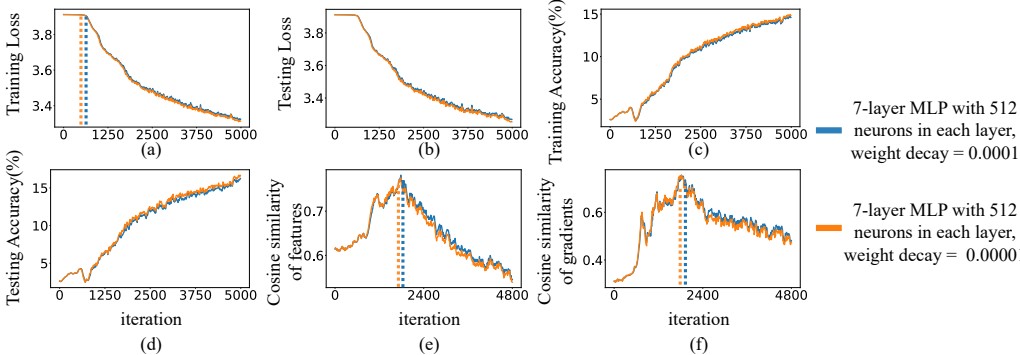

Figure 23: (a) The training loss of two MLPs with different weight decays trained on the Tiny ImageNet dataset. (b) The testing loss of two MLPs. (c) Training accuracies of two MLPs. (d) Testing accuracies of two MLPs. (e) Cosine similarity between features of different categories. (f) Cosine similarity between gradients of different samples in a category. The feature and the feature gradient were used in the third linear layer of MLPs.

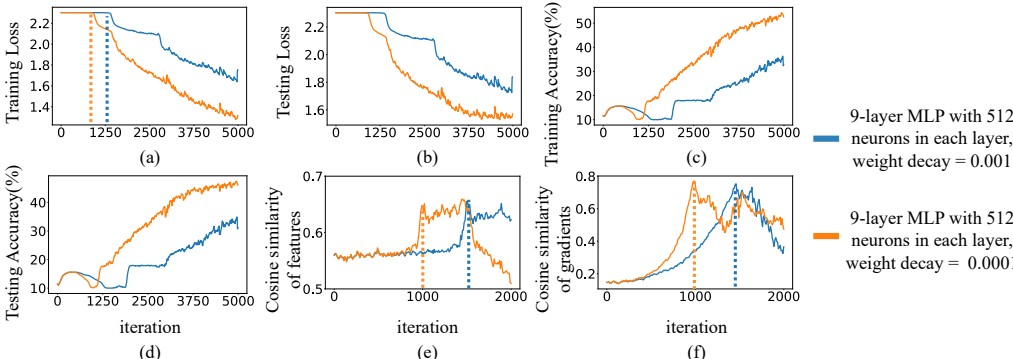

Figure 24: (a) The training loss of two MLPs with different weight decays trained on the CIFAR-10 dataset. (b) The testing loss of two MLPs. (c) Training accuracies of two MLPs. (d) Testing accuracies of two MLPs. (e) Cosine similarity between features of different categories. (f) Cosine similarity between gradients of different samples in a category. The feature and the feature gradient were used in the second linear layer of MLPs.

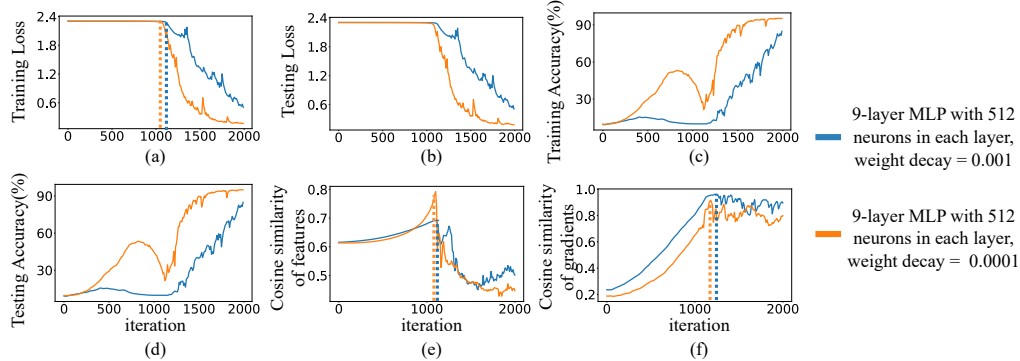

Figure 25: (a) The training loss of two MLPs with different weight decays trained on the MNIST dataset. (b) The testing loss of two MLPs. (c) Training accuracies of two MLPs. (d) Testing accuracies of two MLPs. (e) Cosine similarity between features of different categories. (f) Cosine similarity between gradients of different samples in a category. The feature and the feature gradient were used in the second linear layer of MLPs.

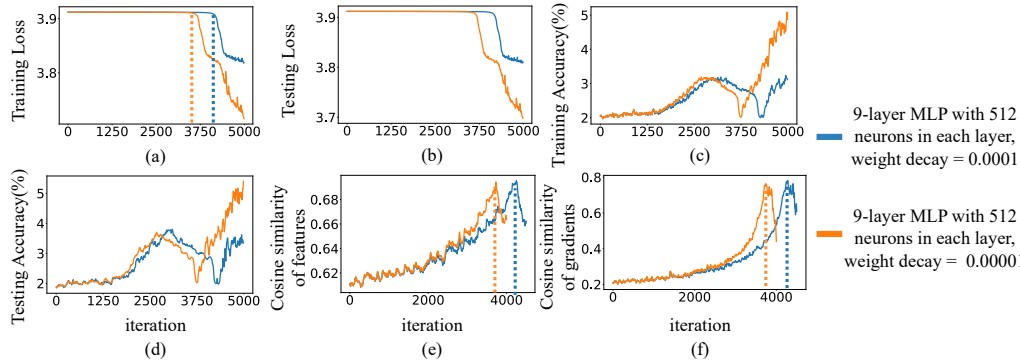

Figure 26: (a) The training loss of two MLPs with different weight decays trained on the Tiny ImageNet dataset. (b) The testing loss of two MLPs. (c) Training accuracies of two MLPs. (d) Testing accuracies of two MLPs. (e) Cosine similarity between features of different categories. (f) Cosine similarity between gradients of different samples in a category. The feature and the feature gradient were used in the third linear layer of MLPs.

### B.14 THE TFC PHENOMENON WITH THE FOCAL LOSS

In this subsection, we demonstrated that the two-phase phenomenon was shared by MLPs learned on the CIFAR-10 dataset with the focal loss. Specifically, for different MLPs, we adopted the learning rate $\eta = 0.1$, the batch size $bs = 100$, and the SGD optimizer. Besides, we used two data augmentation methods, including random cropping and random horizontal flipping. We trained 9-layer MLPs and 7-layer MLPs with 512 neurons in each layer with the ReLU activation function. The training loss, the testing loss, the training accuracy, the testing accuracy, the cosine similarity of features, and the cosine similarity of feature gradients of MLPs trained with different focusing parameters $\gamma$ are shown in Figure 27 and Figure 28. Figure 27 and Figure 28 show that the TFC phenomenon was still observed by different MLPs with the focal loss on the CIFAR-10 dataset.

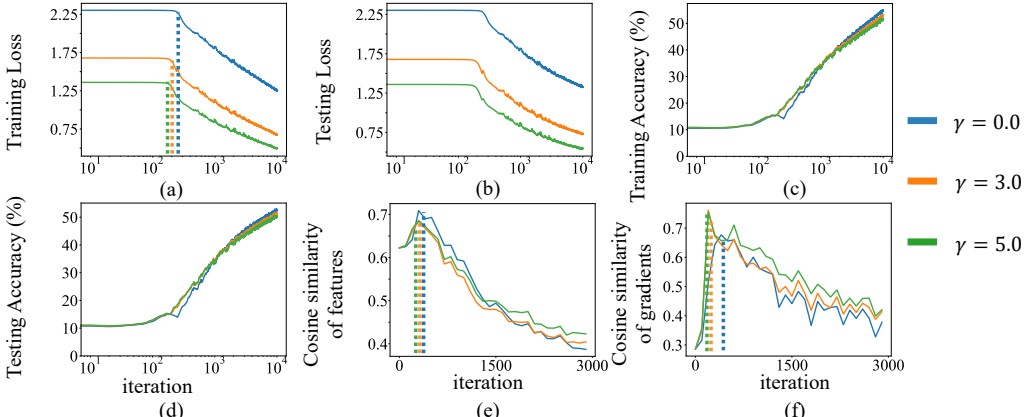

Figure 27: (a) The training loss of three 7-layer MLPs with different focusing parameters $\gamma$ trained on the CIFAR-10 dataset. (b) The testing loss of three MLPs. (c) Training accuracies of three MLPs. (d) Testing accuracies of three MLPs. (e) Cosine similarity between features of different categories. (f) Cosine similarity between gradients of different samples in a category. The feature and the feature gradient were used in the third linear layer of MLPs.

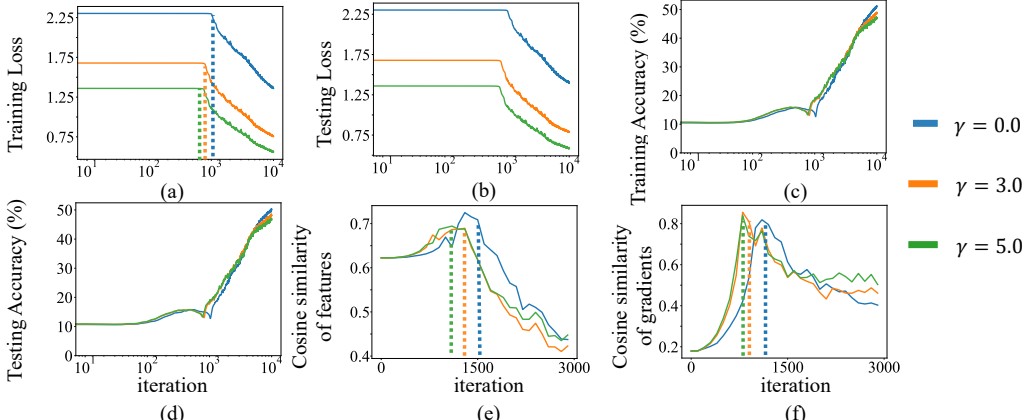

Figure 28: (a) The training loss of three 9-layer MLPs with different focusing parameters $\gamma$ trained on the CIFAR-10 dataset. (b) The testing loss of three MLPs. (c) Training accuracies of three MLPs. (d) Testing accuracies of three MLPs. (e) Cosine similarity between features of different categories. (f) Cosine similarity between gradients of different samples in a category. The feature and the feature gradient were used in the third linear layer of MLPs.

### B.15 DIFFERENT TRAIN/TEST SPLIT FOR DATASETS

In this subsection, we demonstrated that the two-phase phenomenon was shared by MLPs trained on the CIFAR-10 dataset with different train/test splits. There are 50000 samples in the training set and 10000 samples in the testing set on the CIFAR-10 dataset. We combined the training set and the testing set into one dataset and split it with the train/test split ratios of 5:1, 4:2, and 3:3, respectively. Note that the ratio of 5:1 was the official ratio for the CIFAR-10 dataset. For different MLPs, we adopted the learning rate $\eta = 0.1$, the batch size $bs = 100$, and the SGD optimizer. Besides, we used two data augmentation methods, including random cropping and random horizontal flipping. We trained 9-layer MLPs with 512 neurons in each layer with the ReLU activation function on these three different datasets. The training loss, the testing loss, the training accuracy, the testing accuracy, the cosine similarity of features, and the cosine similarity of feature gradients of MLPs trained on different train/test split ratios are shown in Figure 29. Figure 29 shows that the TFC phenomenon was still observed by different train/test split ratios.

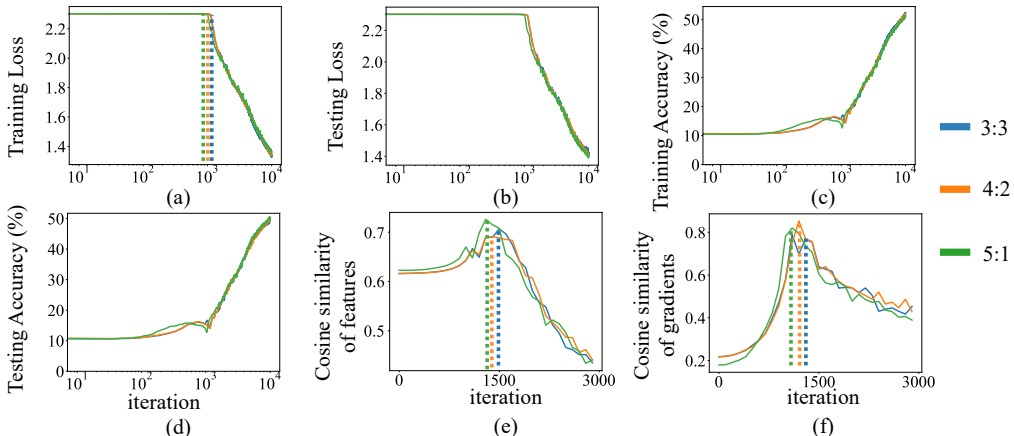

Figure 29: (a) The training loss of three MLPs with different train/test dataset split ratios trained on the CIFAR-10 dataset. (b) The testing loss of three MLPs. (c) Training accuracies of three MLPs. (d) Testing accuracies of three MLPs. (e) Cosine similarity between features of different categories. (f) Cosine similarity between gradients of different samples in a category. The feature and the feature gradient were used in the third linear layer of MLPs.

## C DISCUSSION OF THE PRACTICAL VALUES: THE LEARNING-STICKING PROBLEM

In this section, we aim to discuss the learning-sticking problem in the learning of MLPs. In fact, this problem appears in various DNNs, including MLPs, CNNs, and RNNs, when the task is difficult enough. Explaining and solving the occasional sticking of the training of DNNs are of significant values on different tasks. We consider the learning-sticking problem as the first phase with an infinite length. Moreover, we theoretically explain mechanisms of several heuristic solutions to the learning-sticking problem.

To this end, the learning-sticking problem can be solved based on our study, as shown in Figure 30, Figure 31, Figure 32, Figure 33, Figure 34, and Figure 35. Specifically, we trained a 9-layer MLP on the CIFAR-10 dataset, where each layer of the MLP had 512 neurons and its initial weights were sample from $\mathcal{N}(\mathbf{0}, \Sigma = \gamma_1 \sigma_{\text{var}}^2 I)$. $\sigma_{\text{var}}^2$ was computed following (Glorot & Bengio, 2010) and $\gamma_1 = 0.1$. We trained a VGG-11 model on the CIFAR-10 dataset and its initial weights of fully connected layers were sample from $\mathcal{N}(\mathbf{0}, \Sigma = \gamma_1 \sigma_{\text{var}}^2 I)$ ($\gamma_1 = 0.1$). We trained a VGG-13 model on the CIFAR-10 dataset and its initial weights of fully connected layers were sample from $\mathcal{N}(\mathbf{0}, \Sigma = \gamma_1 \sigma_{\text{var}}^2 I)$ ($\gamma_1 = 0.1$). We trained two ResNet-18 models (without BN layers) on the CIFAR-10 dataset and the Tiny ImageNet dataset, respectively, and initial weights of fully connected layers were sample from $\mathcal{N}(\mathbf{0}, \Sigma = \gamma_1 \sigma_{\text{var}}^2 I)$ ($\gamma_1 = 0.1$).

We observed that these DNNs all suffered from the learning-sticking problem (i.e., the loss minimization of these DNNs get stuck), when their initial weights were sampled from $\mathcal{N}(\mathbf{0}, \Sigma = \gamma_1 \sigma_{\text{var}}^2)$ (orange curves). According to our study, the technique of increasing the variance of initial weights can shorten the first phase, thereby solving the learning-sticking problem. To this end, we trained compared versions of these DNNs, and the only difference from previous DNNs is that the variance of initial weights was increased to $\gamma_2 \sigma_{\text{var}}^2 I$ ($\gamma_2 = 1$). Figure 30, Figure 31, Figure 32, Figure 33, Figure 34, and Figure 35 verify that we could solve the learning-sticking problem by increasing the variance of initialization.

Actually, far beyond solving the learning-sticking problem, the two-phase phenomenon of MLPs is generally considered a counter-intuitive phenomenon. In this paper, our distinctive contribution is to explain the counter-intuitive two-phase phenomenon of MLPs theoretically.

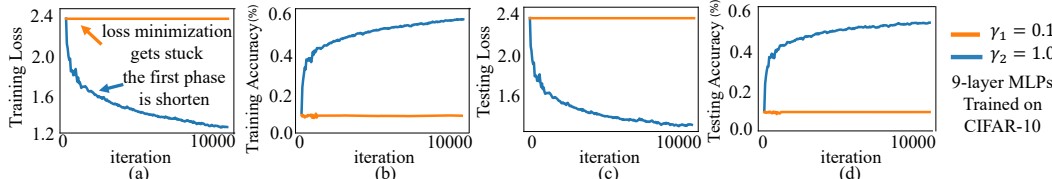

Figure 30: (a) The training loss of two MLPs trained on the CIFAR-10 dataset. When the loss minimization gets stuck (orange curve), we can consider it as the first phase with an infinite length. Therefore, the "learning-sticking" problem can be solved by techniques of shortening the first phase, such as the technique of increasing the variance of initial weights, which is a theoretically certificated solution in our study (blue curve). (b) The training accuracy of two MLPs. (c) The testing loss of two MLPs. (d) The testing accuracy of two MLPs.

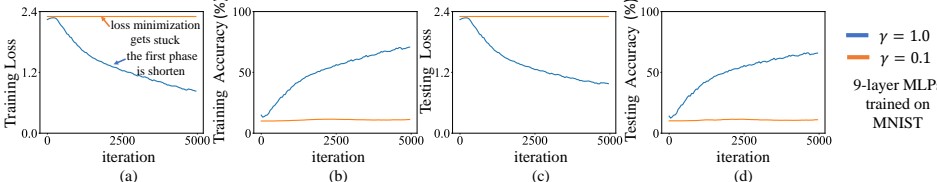

Figure 31: (a) The training loss of two MLPs trained on the MNIST dataset. When the loss minimization gets stuck (orange curve), we can consider it as the first phase with an infinite length. Therefore, the "learning-sticking" problem can be solved by techniques of shortening the first phase, such as the technique of increasing the variance of initial weights, which is a theoretically certificated solution in our study (blue curve). (b) The training accuracy of two MLPs. (c) The testing loss of two MLPs. (d) The testing accuracy of two MLPs.

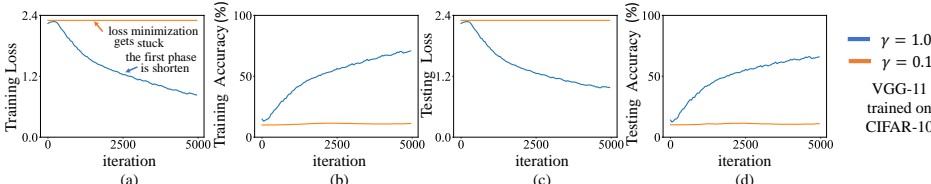

Figure 32: (a) The training loss of two VGG-11 models trained on the CIFAR-10 dataset. When the loss minimization gets stuck (orange curve), we can consider it as the first phase with an infinite length. Therefore, the "learning-sticking" problem can be solved by techniques of shortening the first phase, such as the technique of increasing the variance of initial weights, which is a theoretically certificated solution in our study (blue curve). (b) The training accuracy of two VGG-11 models. (c) The testing loss of two VGG-11 models. (d) The testing accuracy of two VGG-11 models.

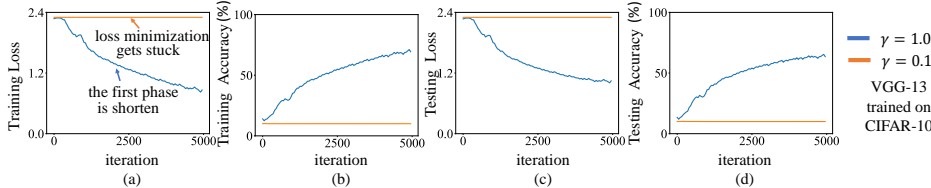

Figure 33: (a) The training loss of two VGG-13 models trained on the CIFAR-10 dataset. When the loss minimization gets stuck (orange curve), we can consider it as the first phase with an infinite length. Therefore, the "learning-sticking" problem can be solved by techniques of shortening the first phase, such as the technique of increasing the variance of initial weights, which is a theoretically certificated solution in our study (blue curve). (b) The training accuracy of two VGG-13 models. (c) The testing loss of two VGG-13 models. (d) The testing accuracy of two VGG-13 models.

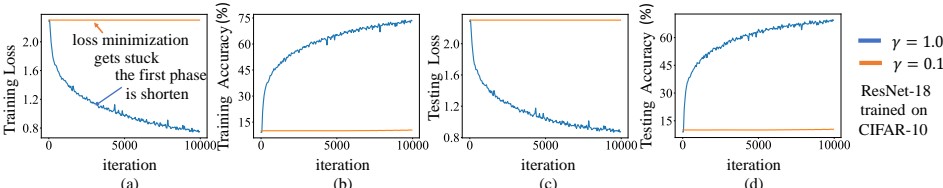

Figure 34: (a) The training loss of two ResNet-18 models trained on the CIFAR-10 dataset. When the loss minimization gets stuck (orange curve), we can consider it as the first phase with an infinite length. Therefore, the "learning-sticking" problem can be solved by techniques of shortening the first phase, such as the technique of increasing the variance of initial weights, which is a theoretically certificated solution in our study (blue curve). (b) The training accuracy of two ResNet-18 models. (c) The testing loss of two ResNet-18 models. (d) The testing accuracy of two ResNet-18 models.

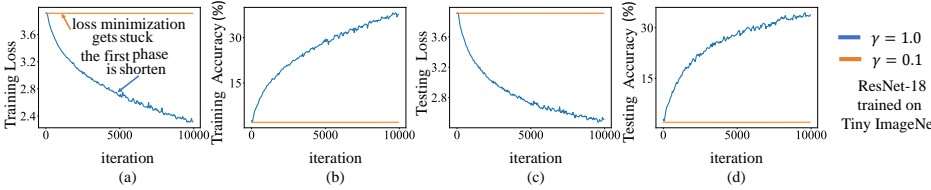

Figure 35: (a) The training loss of two ResNet-18 models trained on the Tiny ImageNet dataset. When the loss minimization gets stuck (orange curve), we can consider it as the first phase with an infinite length. Therefore, the "learning-sticking" problem can be solved by techniques of shortening the first phase, such as the technique of increasing the variance of initial weights, which is a theoretically certificated solution in our study (blue curve). (b) The training accuracy of two ResNet-18 models. (c) The testing loss of two ResNet-18 models. (d) The testing accuracy of two ResNet-18 models.

# D    MORE RESULTS ON OTHER DATASETS

In this section, we provide more results on the MNIST dataset and the Tiny ImageNet dataset. Figure 36 and Table 1 empirically verify the strength of the primary common direction, which are supplementary to Figure 4 and Table 1 in the main paper, respectively. Figure 37 illustrates the change of $o^{(l)} = \cos(\Delta V_t^{(l)}, F_t^{(l-1)}) \cdot \cos(V_t^{(l)}, \Delta F_t^{(l-1)})$ in the first phase, which is supplementary to Figure 6 in the main paper.

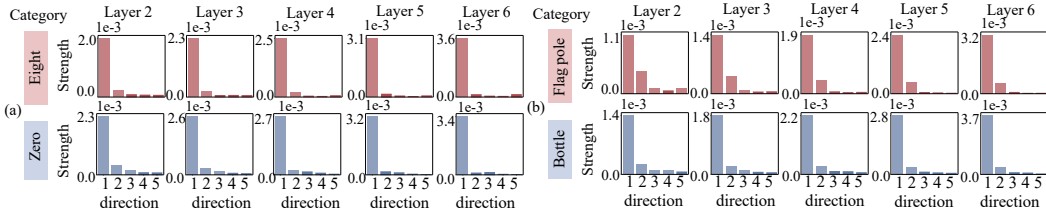

Figure 36: The strength of top-ranked common directions on the (a) MNIST dataset and the (b) Tiny ImageNet dataset. We trained a 9-layer MLP, where each layer of the MLP had 512 neurons. We computed the strength of common directions on the two categories with the highest training accuracies. $s_i = \|C_i \Delta \overline{V}_i^\top\|_F$ measures the strength of weight changes along the $i$-th common direction, where $\Delta \overline{V}_i = \mathbb{E}_t[\Delta V_{i,t}]$. It can be observed that the strength of the primary direction was much greater than the strength of other directions.

Table 1: Strength of components of weight changes along the primary common direction and other directions. We trained a 9-layer MLP on the MNIST dataset. Each layer of the MLP had 512 neurons. It can be observed that the strength of the primary common direction was much greater than those of other directions.

| | Category | Eight | | | | | Zero | | | | |
|---|---|---|---|---|---|---|---|---|---|---|---|
| | $S_{(\times 10^{-3})}$ | Layer 2 | Layer 3 | Layer 4 | Layer 5 | Layer 6 | Layer 2 | Layer 3 | Layer 4 | Layer 5 | Layer 6 |
| MNIST | $S_{\text{primary}}^{(l)}$ | $367.1_{\pm 56.8}$ | $364.5_{\pm 52.8}$ | $381.9_{\pm 56.3}$ | $444.4_{\pm 68.7}$ | $504.0_{\pm 81.3}$ | $441.7_{\pm 86.0}$ | $448.2_{\pm 83.5}$ | $429.0_{\pm 78.1}$ | $493.1_{\pm 87.2}$ | $504.1_{\pm 89.0}$ |
| | $S_1^{(l)}$ | $14.9_{\pm 0.8}$ | $15.9_{\pm 1.4}$ | $15.5_{\pm 1.1}$ | $15.6_{\pm 1.5}$ | $13.5_{\pm 2.0}$ | $24.6_{\pm 3.1}$ | $30.0_{\pm 4.3}$ | $18.4_{\pm 2.6}$ | $17.2_{\pm 2.2}$ | $15.6_{\pm 1.8}$ |
| | $S_2^{(l)}$ | $16.3_{\pm 1.7}$ | $13.1_{\pm 0.9}$ | $16.4_{\pm 0.8}$ | $18.1_{\pm 3.2}$ | $11.7_{\pm 1.6}$ | $16.6_{\pm 1.7}$ | $23.9_{\pm 4.2}$ | $17.9_{\pm 2.4}$ | $14.3_{\pm 1.5}$ | $12.2_{\pm 1.9}$ |
| | $S_3^{(l)}$ | $15.1_{\pm 1.5}$ | $16.3_{\pm 1.7}$ | $13.5_{\pm 0.6}$ | $15.1_{\pm 1.4}$ | $15.0_{\pm 1.1}$ | $29.4_{\pm 5.2}$ | $21.1_{\pm 4.2}$ | $15.5_{\pm 1.8}$ | $21.2_{\pm 3.6}$ | $14.7_{\pm 1.6}$ |

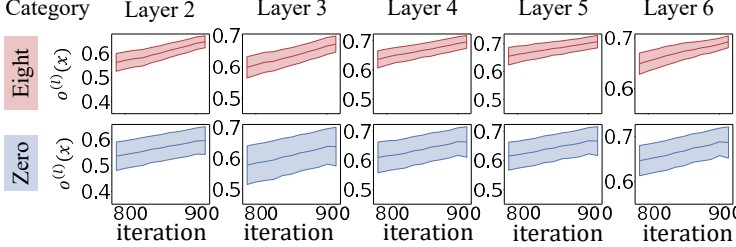

Figure 37: The change of $o^{(l)} = \cos(\Delta V_t^{(l)}, F_t^{(l-1)}) \cdot \cos(V_t^{(l)}, \Delta F_t^{(l-1)})$ in the first phase. We trained a 9-layer MLP on the MNIST dataset. Each layer of the MLP had 512 neurons. The shade represents the standard deviation over different samples.

## E  Proof for the lemma 1

In this section, we present the detailed proof for Lemma 1.

**Lemma 1.** *For the decomposition $\Delta W_t^\top = \Delta V_t C^\top + \Delta \varepsilon_t$, given weight changes over different samples $\Delta W_t^\top$, we can compute the common direction $C$ by minimizing the fitting error $\Delta \epsilon_t$ when we use $\Delta v_{t,i} C^\top$ to approximate $\Delta w_{t,i}^\top$ over different samples across different iterations. I.e., $\min_{C, \Delta V_t|_x} \left( \mathbb{E}_{t \in [T_{start}, T_{end}]} \mathbb{E}_{x \in X} \|\Delta \varepsilon_t|_x\|_F^2 \right)$, s.t. $\Delta \varepsilon_t|_x = \Delta W_t^\top|_x - \Delta V_t|_x C^\top$. Thus, we obtain $\Delta V_t = \frac{\Delta W_t^\top C}{C^\top C}$ and $\Delta \varepsilon_t = \Delta W_t^\top - \Delta W_t^\top \frac{CC^\top}{C^\top C}$, s.t. $\Delta \varepsilon_t C = \mathbf{0}$. Such settings minimize $\|\Delta \varepsilon_t\|_F$.*

*proof.* Let $\Delta \varepsilon_t^\top[j]$ denote the $j$-th column of the matrix $\Delta \varepsilon_t^\top \in \mathbb{R}^{h \times d}$. Given a sample $x$, we can represent $\Delta \varepsilon_t^\top[j]$ by the vector $C$ and a residual term $\Delta \varepsilon_t^\top[j]'$ as follows:

$$\Delta \varepsilon_t^\top[j] = \lambda C + \Delta \varepsilon_t^\top[j]', \tag{1}$$

where $C^\top \Delta \varepsilon_t^\top[j]' = 0$, and $\lambda$ is a scalar.

Then,

$$\begin{aligned}
\left\|\Delta \varepsilon_t^\top[j]\right\|_2^2 &= \left\|\lambda C + \Delta \varepsilon_t^\top[j]'\right\|_2^2 \\
&= (\lambda C + \Delta \varepsilon_t^\top[j]')^\top (\lambda C + \Delta \varepsilon_t^\top[j]') \\
&= \lambda^2 C^\top C + (\Delta \varepsilon_t^\top[j]')^\top \Delta \varepsilon_t^\top[j]' \\
&= \lambda^2 C^\top C + \left\|\Delta \varepsilon_t^\top[j]'\right\|_2^2
\end{aligned} \tag{2}$$

Obviously, $\left\|\Delta \varepsilon_t^\top[j]\right\|_2^2$ is the smallest when $\lambda = 0$. In other words, $\Delta \varepsilon_t^\top[j]$ does not contain the component along the direction $C$ and $C^\top \Delta \varepsilon_t^\top[j] = 0$. Therefore, $\left\|\Delta \varepsilon_t^\top[j]\right\|_2^2$ reaches its minimum if and only if $\Delta \varepsilon_t C = \mathbf{0}$.

When $\left\|\Delta \varepsilon_t^\top[j]\right\|_2^2$ reaches its minimum, $\|\Delta \varepsilon_t\|_F^2$ becomes the smallest. Thus, we have:

$$\begin{aligned}
\Delta W_t &= C \Delta V_t^\top + \Delta \varepsilon_t^\top \\
C^\top \Delta W_t &= C^\top C \Delta V_t^\top + C^T \Delta \varepsilon_t^\top \\
&= C^\top C \Delta V_t^\top + \mathbf{0}
\end{aligned} \tag{3}$$

Then, $\Delta V_t^\top$ can be represented as follows.

$$\Delta V_t^\top = \frac{C^\top \Delta W_t}{C^\top C} \tag{4}$$

Substituting Eq. 4 into $\Delta W_t = C \Delta V_t^\top + \Delta \varepsilon_t^\top$, we have

$$\Delta \varepsilon_t = \Delta W_t^\top - \Delta W_t^\top \frac{CC^\top}{C^\top C} \tag{5}$$

## F  Proof for the lemma 2

In this section, we present the detailed proof for Lemma 2.

**Lemma 2.** *(**We can also decompose the weight $W_t^{(l)}$ into the component along the common direction $C$ and the component $\varepsilon_t$ in other directions.**) Given the weight $W_t^\top$ and the common direction $C$, the decomposition $W_t^\top = V_t C^\top + \varepsilon_t$ can be conducted as $V_t = \frac{W_t^\top C}{C^\top C}$ and $\varepsilon_t = W_t^\top - W_t^\top \frac{CC^\top}{C^\top C}$ s.t. $\varepsilon_t C = \mathbf{0}$. Such settings minimize $\|\varepsilon_t\|_F$. .*

*proof.* Let $\varepsilon_t^\top[j]$ denote the $j$-th column of the matrix $\varepsilon_t^\top \in \mathbb{R}^{h \times d}$. We can represent $\varepsilon_t^\top[j]$ by the vector $C$ and a residual term $\varepsilon_t^\top[j]'$ as follows:

$$\varepsilon_t^\top[j] = \lambda C + \varepsilon_t^\top[j]', \tag{6}$$

where $C^\top \varepsilon_t^\top[j]' = 0$ and $\lambda$ is a scalar.

Then,

$$\begin{aligned}
\left\|\varepsilon_t^\top[j]\right\|_2^2 &= \left\|\lambda C + \varepsilon_t^\top(x)[j]'\right\|_2^2 \\
&= (\lambda C + \varepsilon_t^\top[j]')^\top (\lambda C + \varepsilon_t^\top[j]') \\
&= \lambda^2 C^\top C + (\varepsilon_t^\top[j]')^\top \varepsilon_t^\top[j]' \\
&= \lambda^2 C^\top C + \left\|\varepsilon_t^\top[j]'\right\|_2^2
\end{aligned} \tag{7}$$

Obviously, $\left\|\varepsilon_t^\top[j]\right\|_2^2$ becomes the smallest when $\lambda = 0$. In other words, $\varepsilon_t^\top[j]$ does not contain the component along the direction $C$ and $C^\top \varepsilon_t^\top[j] = 0$. Therefore, $\left\|\varepsilon_t^\top[j]\right\|_2^2$ reaches its minimum if and only if $\varepsilon_t C = \mathbf{0}$.

When $\left\|\varepsilon_t^\top[j]\right\|_2^2$ reaches its minimum, $\|\varepsilon_t\|_F^2$ becomes the smallest. Thus, we have:

$$\begin{aligned}
W_t &= CV_t^\top + \varepsilon_t^\top \\
C^\top W_t &= C^\top CV_t^\top + C^\top \varepsilon_t^\top \\
&= C^\top CV_t^\top + \mathbf{0}
\end{aligned} \tag{8}$$

Then, $V_t^\top$ can be written as follows.

$$V_t^\top = \frac{C^\top W_t}{C^\top C} \tag{9}$$

Substituting Eq. 9 into $W_t = CV_t^\top + \varepsilon_t^\top$, we have

$$\varepsilon_t = W_t^\top - W_t^\top \frac{CC^\top}{C^\top C} \tag{10}$$

## G  DECOMPOSITION OF COMMON DIRECTIONS

Actually, the estimation of the common direction $C$ is similar to the singular value decomposition (SVD), although there are slight differences.

We compute the average weight change $\Delta \overline{W}_t = \mathbb{E}_{x \in X} \Delta W_t|_x$, where $\Delta W_t|_x$ denotes the weight change made by the sample $x$. Then, we decompose $\Delta \overline{W}_t$ into components along five common directions as $\Delta \overline{W}_t = C_1 \Delta \overline{V}_{1,t}^\top + C_2 \Delta \overline{V}_{2,t}^\top + \cdots + C_5 \Delta \overline{V}_{5,t}^\top + \Delta \overline{\varepsilon}_{5,t}^\top$, where $C_1 = C$ is termed the *primary common direction*. $C_1, C_2, C_3, C_4$, and $C_5$ are orthogonal to each other. $C_2, C_3, C_4$ and $C_5$ represent the second, third, forth, and fifth common directions, respectively. $C_i$ represents the $i$-th common direction. $\Delta \overline{V}_{i,t}$ denotes the average weight change along the $i$-th common direction decomposed from $\Delta \overline{W}_t$.

Specifically, we first decompose the average weight change $\Delta \overline{W}_t$ after the $t$-th iteration as $\Delta \overline{W}_t = C \Delta \overline{V}_t^\top + \Delta \overline{\varepsilon}_t^\top$. We remove all components along the common direction $C$ from $\Delta \overline{W}_t$, and obtain $\Delta \overline{W}_{\mathrm{new},t} = \Delta \overline{W}_t - C \Delta \overline{V}_t^\top = \Delta \overline{\varepsilon}_t^\top$. Then, we further decompose $\Delta \overline{W}_{\mathrm{new},t} = C_2 \Delta V_{2,t}^\top + \Delta \varepsilon_{2,t}^\top$. In this way, we can consider $C_2$ as the secondary common direction, while $C_1 = C$ is termed as the primary common direction. Thus, we conduct this process recursively and obtain common directions $\{C_1, C_2, \cdots C_5\}$. Accordingly, $\Delta \overline{W}_t$ is decomposed into $\Delta \overline{W}_t = C_1 \Delta \overline{V}_{1,t}^\top + C_2 \Delta \overline{V}_{2,t}^\top + \cdots + C_5 \Delta V_{5,t}^\top + \Delta \varepsilon_{5,t}^\top$.

# H DECOMPOSITION OF THE WEIGHT CHANGE MADE BY A SAMPLE $x$

## H.1 PROOF FOR THEOREM 1.

In this subsection, we present the detailed proof for Theorem 1.

**Theorem 1.** *The weight change made by a sample can be decomposed into $(h+1)$ terms after the $t$-th iteration as follows.*

$$\Delta W_t^{(l)} = \Delta W_{\text{primary},t}^{(l)} + \sum_{k=1}^{h} \Delta W_{\text{noise},t}^{(l,k)} \xrightarrow{\text{rewritten}} \Gamma_t^{(l)} F_t^{(l-1)^\top} + \kappa_t^{(l)^\top}, \tag{11}$$

*where $\Delta W_{\text{primary},t}^{(l)} = D_t^{(l)} V_t^{(l+1)} C^{(l+1)^\top} C^{(l+1)} \Delta V_t^{(l+1)^\top} F_t^{(l)} F_t^{(l-1)^\top} / \|F_t^{(l)}\|_2^2$ denotes the component along the primary common direction, and $\Delta W_{\text{noise},t}^{(l,k)} = D_t^{(l)} \varepsilon_t^{(l+1,k)} \Delta \varepsilon_t^{(l+1)^\top} F_t^{(l)} F_t^{(l-1)^\top} / \|F_t^{(l)}\|_2^2$ denotes the component along the $k$-th common direction in the noise term. $\varepsilon_t^{(l+1,k)} = \mathbf{\Sigma}_{kk} \mathcal{U}_k \mathcal{V}_k^\top$, where the SVD of $\varepsilon_t^{(l+1)} \in \mathbb{R}^{h \times h'}$ is given as $\varepsilon_t^{(l+1)} = \mathcal{U} \mathbf{\Sigma} \mathcal{V}^\top$ ($h \leq h'$), and $\mathbf{\Sigma}_{kk}$ denotes the $k$-th singular value $\in \mathbb{R}$. $\varepsilon_t^{(l+1)} = \sum_k \varepsilon_t^{(l+1,k)}$. $\mathcal{U}_k$ and $\mathcal{V}_k$ denote the $k$-th column of the matrix $\mathcal{U}$ and $\mathcal{V}$, respectively. Besides, we have $\forall k \in \{1, 2, \ldots, h\}$, $\mathcal{U}_k^\top C^{(l+1)} = 0$. Consequently, we have $\Gamma_t^{(l)} = D_t^{(l)} V_t^{(l+1)} C^{(l+1)^\top} C^{(l+1)} \Delta V_t^{(l+1)^\top} F_t^{(l)} / \|F_t^{(l)}\|_2^2 \in \mathbb{R}^h$, and $\kappa_t^{(l)^\top} = D_t^{(l)} \varepsilon_t^{(l+1)} \Delta \varepsilon_t^{(l+1)^\top} F_t^{(l)} F_t^{(l-1)^\top} / \|F_t^{(l)}\|_2^2 \in \mathbb{R}^{h \times d}$.*

*proof.* We can represent weight matrix as $W_t^{(l)} = C^{(l)} V_t^{(l)^\top} + \varepsilon_t^{(l)^\top}$. In addition, according to the back propagation and chain rule, we have $\Delta W_t^{(l)} = -\eta D_t^{(l)} \dot{F}_t^{(l)} F_t^{(l-1)^\top}$, where $\dot{F}_t^{(l)} = \frac{\partial Loss}{\partial F_t^{(l)}}$, and $\eta$ denotes the learning rate.

According to Lemma 1 and Lemma 2, we have $\Delta \varepsilon_t^{(l+1)} C^{(l+1)} = \mathbf{0}$ and $\varepsilon_t^{(l+1)} C^{(l+1)} = \mathbf{0}$. After the $t$-th iteration, the weight change made by a training sample $x$ can be computed as follows.

$$
\begin{aligned}
\Delta W_t^{(l)} &= -\eta D_t^{(l)} \dot{F}_t^{(l)} F_t^{(l-1)^\top} \\
&= -\eta D_t^{(l)} W_t^{(l+1)^\top} D_t^{(l+1)} \dot{F}_t^{(l+1)} F_t^{(l-1)^\top} \\
&= D_t^{(l)} W_t^{(l+1)^\top} \Delta W_t^{(l+1)} F_t^{(l)} F_t^{(l-1)^\top} / \left\| F_t^{(l)} \right\|_2^2 \\
&= D_t^{(l)} \left[ V_t^{(l+1)} C^{(l+1)^\top} + \varepsilon_t^{(l+1)} \right] \left[ C^{(l+1)} \Delta V_t^{(l+1)^\top} + \Delta \varepsilon_t^{(l+1)^\top} \right] F_t^{(l)} F_t^{(l-1)^\top} / \left\| F_t^{(l)} \right\|_2^2 \\
&= D_t^{(l)} [V_t^{(l+1)} C^{(l+1)^\top} C^{(l+1)} \Delta V_t^{(l+1)^\top} + V_t^{(l+1)} C^{(l+1)^\top} \Delta \varepsilon_t^{(l+1)^\top} \\
&\quad + \varepsilon_t^{(l+1)} C^{(l+1)} \Delta V_t^{(l+1)^\top} + \varepsilon_t^{(l+1)} \Delta \varepsilon_t^{(l+1)^\top}] F_t^{(l)} F_t^{(l-1)^\top} / \left\| F_t^{(l)} \right\|_2^2 \\
&= D_t^{(l)} \left[ V_t^{(l+1)} C^{(l+1)^\top} C^{(l+1)} \Delta V_t^{(l+1)^\top} + \varepsilon_t^{(l+1)} \Delta \varepsilon_t^{(l+1)^\top} \right] F_t^{(l)} F_t^{(l-1)^\top} / \left\| F_t^{(l)} \right\|_2^2 \\
&= D_t^{(l)} V_t^{(l+1)} C^{(l+1)^\top} C^{(l+1)} \Delta V_t^{(l+1)^\top} F_t^{(l)} F_t^{(l-1)^\top} / \left\| F_t^{(l)} \right\|_2^2 \\
&\quad + D_t^{(l)} \varepsilon_t^{(l+1)} \Delta \varepsilon_t^{(l+1)^\top} F_t^{(l)} F_t^{(l-1)^\top} / \left\| F_t^{(l)} \right\|_2^2
\end{aligned}
\tag{12}
$$

$\varepsilon_t^{(l+1,k)} = \Sigma_{kk} \mathcal{U}_k \mathcal{V}_k^\top$, where the singular value decomposition of $\varepsilon_t^{(l+1)}$ is given as $\varepsilon_t^{(l+1)} = \mathcal{U} \mathbf{\Sigma} \mathcal{V}^\top$, and $\mathbf{\Sigma}_{kk}$ denotes the $k$-th singular value. $\mathcal{U}_k$ and $\mathcal{V}_k$ denote the $k$-th column of the matrix $\mathcal{U}$ and $\mathcal{V}$, respectively. We can derive the following equations.

$$\Delta W_t^{(l)} = D_t^{(l)} V_t^{(l+1)} C^{(l+1)^T} C^{(l+1)} \Delta V_t^{(l+1)^\top} F_t^{(l)} F_t^{(l-1)^\top} / \left\| F_t^{(l)} \right\|_2^2$$

$$+ D_t^{(l)} \varepsilon_t^{(l+1)} \Delta \varepsilon_t^{(l+1)^\top} F_t^{(l)} F_t^{(l-1)^\top} / \left\| F_t^{(l)} \right\|_2^2$$

$$= D_t^{(l)} V_t^{(l+1)} C^{(l+1)^T} C^{(l+1)} \Delta V_t^{(l+1)^\top} F_t^{(l)} F_t^{(l-1)^\top} / \left\| F_t^{(l)} \right\|_2^2 \qquad (13)$$

$$+ \sum_{k=1}^{h} D_t^{(l)} \varepsilon_t^{(l+1,k)} \Delta \varepsilon_t^{(l+1)^\top} F_t^{(l)} F_t^{(l-1)^\top} / \left\| F_t^{(l)} \right\|_2^2.$$

$$= \Delta W_{\text{primary},t}^{(l)} + \sum_{k=1}^{h} \Delta W_{t,\text{noise}}^{(l,k)}$$

In addition, if we set $\Gamma_t^{(l)} = D_t^{(l)} V_t^{(l+1)} C^{(l+1)^\top} C^{(l+1)} \Delta V_t^{(l+1)^\top} F_t^{(l)} / \|F_t^{(l)}\|_2^2$, and $\kappa_t^{(l)^\top} = D_t^{(l)} \varepsilon_t^{(l+1)} \Delta \varepsilon_t^{(l+1)^\top} F_t^{(l)} F_t^{(l-1)^\top} / \|F_t^{(l)}\|_2^2$. Then we can re-write the Eq. (13) as follows.

$$\Delta W_t^{(l)} \stackrel{\text{rewritten}}{=\joinrel=\joinrel=} \Gamma_t^{(l)} F_t^{(l-1)^\top} + \kappa_t^{(l)^\top} \qquad (14)$$

## H.2 THE EXPLANATION FOR THE PHENOMENON THAT $S_1^{(l)}$, $S_2^{(l)}$, AND $S_3^{(l)}$ DO NOT DECREASE MONOTONICALLY.

In this subsection, we explain the phenomenon that $S_1^{(l)}$, $S_2^{(l)}$, and $S_3^{(l)}$ does not decrease monotonically in Table 1 in Appendix and Table 1 in the main paper (Page 6). In fact, we first decompose $\varepsilon_t^{(l+1)} = \sum_k \varepsilon_t^{(l+1,k)}$ according to the SVD. Then $\Delta W_{\text{noise},t}^{(l,k)}$ is computed as $\Delta W_{\text{noise},t}^{(l,k)} = D_t^{(l)} \varepsilon_t^{(l+1,k)} \Delta \varepsilon_t^{(l+1)^\top} F_t^{(l)} F_t^{(l-1)^\top} / \left\| F_t^{(l)} \right\|_2^2$. Accordingly, the strength of weight changes along the primary direction is computed as $S_{\text{primary}}^{(l)} = \mathbb{E}_{t \in [T_{\text{start}}, T_{\text{end}}]} \mathbb{E}_{x \in X} \left[ \| \Delta W_{\text{primary},t}^{(l,k)} | x \|_F \right]$. The strength of weight changes along the $k$-th noise direction is computed as $S_k^{(l)} = \mathbb{E}_{t \in [T_{\text{start}}, T_{\text{end}}]} \mathbb{E}_{x \in X} \left[ \| \Delta W_{\text{noise},t}^{(l,k)} | x \|_F \right]$. In this way, $S_1^{(l)}$, $S_2^{(l)}$, and $S_3^{(l)}$ do not decrease monotonically, although $\| \varepsilon_t^{(l+1,1)} \|_F$, $\| \varepsilon_t^{(l+1,2)} \|_F$, and $\| \varepsilon_t^{(l+1,3)} \|_F$ are directly decomposed from $\varepsilon_t^{(l+1)}$ based on the SVD and decrease monotonically.

## I ANALYSIS BASED ON EQ. (3) IN THE MAIN PAPER AND EXPLANATION FOR THE PARALLELISM.

According the Eq. (3) in the main paper, we have

$$\dot{F}_t^{(l-1)} = (C^{(l)^\top} D_t^{(l)} \dot{F}_t^{(l)}) \cdot \boldsymbol{\beta} + \boldsymbol{\epsilon} D_t^{(l)} \dot{F}_t^{(l)} \qquad (15)$$

Thus, if $C^{(l)^\top} D_t^{(l)} \dot{F}_t^{(l)}$ is large enough (*i.e.*, keeping optimizing $W_t^{(l)^\top}$ along the common direction $C^{(l)}$ for a long time), then the feature gradients $\dot{F}_t^{(l-1)}$ of different samples will be roughly parallel to the same vector $\boldsymbol{\beta}$. This is because $C^{(l)^\top} D_t^{(l)} \dot{F}_t^{(l)}$ is a scalar and the term $\boldsymbol{\epsilon} D_t^{(l)} \dot{F}_t^{(l)}$ is small. In other words, the diversity between feature gradients $\dot{F}_t^{(l-1)}$ of different samples decreases. Here, $\boldsymbol{\beta} = [\beta_1, \beta_2, \cdots, \beta_d]$, and $\boldsymbol{\epsilon} = [\epsilon_1, \epsilon_2, \cdots, \epsilon_d]^\top$.

## J DISCUSSION ON THE BACKGROUND ASSUMPTION.

In the above section, we demonstrate that on the ideal state, *i.e.*, $W_t^{(l)^\top}$ has been optimized towards the common direction $C^{(l)}$ for a long time, we can consider that the feature gradients $\dot{F}_t^{(l-1)}$ of different samples will be roughly parallel to the same vector $\boldsymbol{\beta}$. In this way, we can explain that the diversity between feature gradients $\dot{F}_t^{(l-1)}$ of different samples decreases.

In comparison, in the current section, we mainly discuss the trustworthiness of the background assumption in Section 4.2 in the main paper. We aim to discuss that on the assumption that features $F_t^{(l-1)}$ of different samples have been pushed a little bit towards a specific common direction, we can find at least one learning iteration in the first phase where $\Delta F_t^{(l-1)}$ and $F_t^{(l-1)}$ of most samples have similar directions, and $V_t^{(l)}$ and $\Delta V_t^{(l)}$ have similar directions. The assumption that features $F_t^{(l-1)}$ of different samples have been pushed a little bit towards a specific common direction is an intermediate state between the chaotic initial state of the MLP and the ideal state introduced in the above section. In this way, we can assume that $C^{(l)^\top} D_t^{(l)} \dot{F}_t^{(l)}$ is large.

According to Eq. (2) in the main paper and Lemma 2, we have $\dot{F}_t^{(l-1)} = W_t^{(l)^\top} D_t^{(l)} \dot{F}_t^{(l)}$ and $W_t^{(l)^\top} = V_t^{(l)} C^{(l)^\top} + \varepsilon_t^{(l)^\top}$. Thus, we have

$$
\begin{aligned}
\dot{F}_t^{(l-1)} &= W_t^{(l)^\top} D_t^{(l)} \dot{F}_t^{(l)} \\
&= (V_t^{(l)} C^{(l)^\top} + \varepsilon_t^{(l)^\top}) D_t^{(l)} \dot{F}_t^{(l)} \\
&= V_t^{(l)} C^{(l)^\top} D_t^{(l)} \dot{F}_t^{(l)} + \varepsilon_t^{(l)^\top} D_t^{(l)} \dot{F}_t^{(l)}
\end{aligned}
\tag{16}
$$

If the scalar $C^{(l)^\top} D_t^{(l)} \dot{F}_t^{(l)}$ is large, we can roughly consider

$$
\begin{aligned}
\dot{F}_t^{(l-1)} &\approx V_t^{(l)} C^{(l)^\top} D_t^{(l)} \dot{F}_t^{(l)} \\
&= V_t^{(l)} \cdot (C^{(l)^\top} D_t^{(l)} \dot{F}_t^{(l)}) \;/\!/\; V_t^{(l)}
\end{aligned}
\tag{17}
$$

It means that the feature gradient $\dot{F}_t^{(l-1)}$ is roughly parallel to the vector $V_t^{(l)}$. Furthermore, the feature gradient $\dot{F}_t^{(l-1)}$ and the change of feature $\Delta F_t^{(l-1)}$ can be considered negatively parallel to each other, we have

$$
\Delta F_t^{(l-1)} \;/\!/\; \dot{F}_t^{(l-1)} \;/\!/\; V_t^{(l)}
\tag{18}
$$

Similarly, we have $\Delta F_{t+1}^{(l-1)} \;/\!/\; V_{t+1}^{(l)}$. Therefore, we can roughly consider that $V_t^{(l)} \approx k_t \Delta F_t^{(l-1)}$, and $V_{t+1}^{(l)} \approx k_{t+1} \Delta F_{t+1}^{(l-1)}$, where $k_t, k_{t+1} \in \mathbb{R}$ are two scalars. Then, we can derive that

$$
\Delta V_t^{(l)} = V_{t+1}^{(l)} - V_t^{(l)} \approx k_{t+1} \Delta F_{t+1}^{(l-1)} - k_t \Delta F_t^{(l-1)}
\tag{19}
$$

If features $F_t^{(l-1)}$ of different samples have been pushed a little bit towards a specific common direction, then it is easy to find at least one learning iteration that $\Delta F_t^{(l-1)}$ and $F_t^{(l-1)}$ of most samples have similar directions, *i.e.* $\Delta F_t^{(l-1)} \;/\!/\; F_t^{(l-1)}$. Meanwhile, we can find at least one learning iteration in the first phase where the change of feature in $t$-th iteration $\Delta F_t^{(l-1)}$ and $(t+1)$-th iteration $\Delta F_{t+1}^{(l-1)}$ are roughly the same. In other words, $\Delta F_t^{(l-1)} \approx \Delta F_{t+1}^{(l-1)}$. Thus, we have

$$
\Delta V_t^{(l)} \approx (k_{t+1} - k_t) \Delta F_t^{(l-1)} \;/\!/\; \Delta F_t^{(l-1)} \;/\!/\; V_t^{(l)}
\tag{20}
$$

In this way, we can obtain that $V_t^{(l)}$ and $\Delta V_t^{(l)}$ have similar directions.

## K  PROOF FOR LEMMA 3

In this section, we present the detailed proof for Lemma 3.

**Lemma 3.** *Given an input sample $x \in X$ and a common direction $C^{(l)}$ after the $t$-th iteration, if the noise term $\varepsilon_t^{(l)}$ is small enough to satisfy $|\Delta V_t^{(l)^\top} F_t^{(l-1)} V_t^{(l)^\top} V_t^{(l)} C^{(l)^\top} C^{(l)} \Delta V_t^{(l)^\top} F_t^{(l-1)}| \gg |\Delta V_t^{(l)^\top} F_t^{(l-1)} V_t^{(l)^\top} \varepsilon_t^{(l)} \Delta \varepsilon_t^{(l)^\top} F_t^{(l-1)}|$, we can obtain $\cos(\Delta V_t^{(l)}, F_t^{(l-1)}) \cdot \cos(V_t^{(l)}, \Delta F_t^{(l-1)}) \geq 0$, where $\Delta V_t^{(l)} = \frac{\Delta W_t^{(l)^\top} C^{(l)}}{C^{(l)^\top} C^{(l)}}$, and $V_t^{(l)} = \frac{W_t^{(l)^\top} C^{(l)}}{C^{(l)^\top} C^{(l)}}$. $\Delta F_t^{(l-1)}$ denotes the change of features $\Delta F_t^{(l-1)} = F_{t+1}^{(l-1)} - F_t^{(l-1)}$ made by the training sample $x$ after the $t$-th iteration. To this end, we approximately consider the change of features $\Delta F_t^{(l-1)}$ after the $t$-th iteration negatively parallel to feature gradients $\dot{F}_t^{(l-1)}$, although strictly speaking, the change of features is not exactly equal to the feature gradients.*

*proof.* Given a sample $x$, we can prove that $\cos(\Delta V_t^{(l)}, F_t^{(l-1)}) \cdot \cos(V_t^{(l)}, \Delta F_t^{(l-1)}) \geq 0$.

According to chain rule, we have

$$\Delta W_t^{(l)} = -\eta D_t^{(l)} \dot{F}_t^{(l)} F_t^{(l-1)^T} \tag{21}$$

According to Lemma 1 and Lemma 2, we have $C^{(l)^\top} \Delta \varepsilon_t^{(l)^\top} = 0$ and $\varepsilon_t^{(l)} C^{(l)} = 0$. Then, we have

$$\cos(\Delta V_t^{(l)}, F_t^{(l-1)}) \cdot \cos(V_t^{(l)}, \dot{F}_t^{(l-1)}) = \left[ \frac{\Delta V_t^{(l)^\top} F_t^{(l-1)}}{\|\Delta V_t^{(l)}\| \cdot \|F_t^{(l-1)}\|} \right] \cdot \left[ \frac{V_t^{(l)^\top} \dot{F}_t^{(l-1)}}{\|V_t^{(l)}\| \cdot \|\dot{F}_t^{(l-1)}\|} \right] \tag{22}$$

Therefore, we have

$\mathrm{sign}(\cos(\Delta V_t^{(l)}, F_t^{(l-1)}) \cdot \cos(V_t^{(l)}, \dot{F}_t^{(l-1)}))$

$= \mathrm{sign}([\Delta V_t^{(l)^\top} F_t^{(l-1)}] \cdot [V_t^{(l)^\top} \dot{F}_t^{(l-1)}] / (\|\Delta V_t^{(l)}\|_2 \|F_t^{(l-1)}\|_2 \|V_t^{(l)}\|_2 \|\dot{F}_t^{(l-1)}\|_2))$

$= \mathrm{sign}([\Delta V_t^{(l)^\top} F_t^{(l-1)}] \cdot [V_t^{(l)^\top} W_t^{(l)^\top} D_t^{(l)} \dot{F}_t^{(l)}] / (\|\Delta V_t^{(l)}\|_2 \|F_t^{(l-1)}\|_2 \|V_t^{(l)}\|_2 \|\dot{F}_t^{(l-1)}\|_2))$

$= \mathrm{sign}([\Delta V_t^{(l)^\top} F_t^{(l-1)}] \cdot [V_t^{(l)^\top} (V_t^{(l)} C^{(l)^\top} + \varepsilon_t^{(l)}) D_t^{(l)} \dot{F}_t^{(l)}] / (\|\Delta V_t^{(l)}\|_2 \|F_t^{(l-1)}\|_2 \|V_t^{(l)}\|_2 \|\dot{F}_t^{(l-1)}\|_2))$

$= \mathrm{sign}([\Delta V_t^{(l)^\top} F_t^{(l-1)}] \cdot [V_t^{(l)^\top} (V_t^{(l)} C^{(l)^\top} + \varepsilon_t^{(l)}) (\Delta W_t^{(l)} F_t^{(l-1)} / (-\eta \left\| F_t^{(l-1)} \right\|_2^2))]$

$/ (\|\Delta V_t^{(l)}\|_2 \|F_t^{(l-1)}\|_2 \|V_t^{(l)}\|_2 \|\dot{F}_t^{(l-1)}\|_2))$

$= \mathrm{sign}([\Delta V_t^{(l)^\top} F_t^{(l-1)}] \cdot [(V_t^{(l)^\top} V_t^{(l)} C^{(l)^\top} + V_t^{(l)^\top} \varepsilon_t^{(l)}) \Delta W_t^{(l)} F_t^{(l-1)}]$

$/ (-\eta \left\| F_t^{(l-1)} \right\|_2^2 \|\Delta V_t^{(l)}\|_2 \|F_t^{(l-1)}\|_2 \|V_t^{(l)}\|_2 \|\dot{F}_t^{(l-1)}\|_2))$

$= \mathrm{sign}([\Delta V_t^{(l)^\top} F_t^{(l-1)}] \cdot [(V_t^{(l)^\top} V_t^{(l)} C^{(l)^\top} + V_t^{(l)^\top} \varepsilon_t^{(l)}) (C^{(l)} \Delta V_t^{(l)^\top} + \Delta \varepsilon_t^{(l)^\top}) F_t^{(l-1)}]$

$/ (-\eta \left\| F_t^{(l-1)} \right\|_2^2 \|\Delta V_t^{(l)}\|_2 \|F_t^{(l-1)}\|_2 \|V_t^{(l)}\|_2 \|\dot{F}_t^{(l-1)}\|_2))$

$= \mathrm{sign}([\Delta V_t^{(l)^\top} F_t^{(l-1)}] \cdot [(V_t^{(l)^\top} V_t^{(l)} C^{(l)^\top} C^{(l)} \Delta V_t^{(l)^\top} + V_t^{(l)^\top} \varepsilon_t^{(l)} \Delta \varepsilon_t^{(l)^\top}$

$+ V_t^{(l)^\top} V_t^{(l)} C^{(l)^\top} \Delta \varepsilon_t^{(l)^\top} + V_t^{(l)^\top} \varepsilon_t^{(l)} C^{(l)} \Delta V_t^{(l)^\top}) F_t^{(l-1)}] / (-\eta \left\| F_t^{(l-1)} \right\|_2^2 \|\Delta V_t^{(l)}\|_2 \|\dot{F}_t^{(l-1)}\|_2 \|V_t^{(l)}\|_2 \|F_t^{(l-1)}\|_2))$

$= \mathrm{sign}([\Delta V_t^{(l)^\top} F_t^{(l-1)}] \cdot [(V_t^{(l)^\top} V_t^{(l)} C^{(l)^\top} C^{(l)} \Delta V_t^{(l)^\top} + V_t^{(l)^\top} \varepsilon_t^{(l)} \Delta \varepsilon_t^{(l)^\top}) F_t^{(l-1)}]$

$/ (-\eta \left\| F_t^{(l-1)} \right\|_2^2 \|\Delta V_t^{(l)}\|_2 \|F_t^{(l-1)}\|_2 \|V_t^{(l)}\|_2 \|\dot{F}_t^{(l-1)}\|_2))$

$= \mathrm{sign}([\Delta V_t^{(l)^\top} F_t^{(l-1)}] \cdot [V_t^{(l)^\top} V_t^{(l)} C^{(l)^\top} C^{(l)} \Delta V_t^{(l)^\top} F_t^{(l-1)} + V_t^{(l)^\top} \varepsilon_t^{(l)} \Delta \varepsilon_t^{(l)^\top} F_t^{(l-1)}]$

$/ (-\eta \left\| F_t^{(l-1)} \right\|_2^2 \|\Delta V_t^{(l)}\|_2 \|F_t^{(l-1)}\|_2 \|V_t^{(l)}\|_2 \|\dot{F}_t^{(l-1)}\|_2))$

$= \mathrm{sign}([\Delta V_t^{(l)^\top} F_t^{(l-1)} V_t^{(l)^\top} V_t^{(l)} C^{(l)^\top} C^{(l)} \Delta V_t^{(l)^\top} F_t^{(l-1)} + \Delta V_t^{(l)^\top} F_t^{(l-1)} V_t^{(l)^\top} \varepsilon_t^{(l)} \Delta \varepsilon_t^{(l)^\top} F_t^{(l-1)}]$

$/ (-\eta \left\| F_t^{(l-1)} \right\|_2^2 \|\Delta V_t^{(l)}\|_2 \|F_t^{(l-1)}\|_2 \|V_t^{(l)}\|_2 \|\dot{F}_t^{(l-1)}\|_2))$

$$\tag{23}$$

According to our assumption, the noise term $\varepsilon_t^{(l)}$ is small enough to satisfy $|\Delta V_t^{(l)^\top} F_t^{(l-1)} V_t^{(l)^\top} V_t^{(l)} C^{(l)^\top} C^{(l)} \Delta V_t^{(l)^\top} F_t^{(l-1)}| \gg |\Delta V_t^{(l)^\top} F_t^{(l-1)} V_t^{(l)^\top} \varepsilon_t^{(l)} \Delta \varepsilon_t^{(l)^\top} F_t^{(l-1)}|$. This assumption is verified in Figure 38. Then we can ignore the last term and obtain

$\mathrm{sign}([\Delta V_t^{(l)^\top} F_t^{(l-1)} V_t^{(l)^\top} V_t^{(l)} C^{(l)^\top} C^{(l)} \Delta V_t^{(l)^\top} F_t^{(l-1)} + \Delta V_t^{(l)^\top} F_t^{(l-1)} V_t^{(l)^\top} \varepsilon_t^{(l)} \Delta \varepsilon_t^{(l)^\top} F_t^{(l-1)}]$

$/ (-\eta \left\| F_t^{(l-1)} \right\|_2^2 \|\Delta V_t^{(l)}\|_2 \|F_t^{(l-1)}\|_2 \|V_t^{(l)}\|_2 \|\dot{F}_t^{(l-1)}\|_2))$

$\approx \mathrm{sign}([\Delta V_t^{(l)^\top} F_t^{(l-1)} V_t^{(l)^\top} V_t^{(l)} C^{(l)^\top} C^{(l)} \Delta V_t^{(l)^\top} F_t^{(l-1)}]$

$(-\eta \left\| F_t^{(l-1)} \right\|_2^2 \|\Delta V_t^{(l)}\|_2 \|F_t^{(l-1)}\|_2 \|V_t^{(l)}\|_2 \|\dot{F}_t^{(l-1)}\|_2)) \leq 0$

$$\tag{24}$$

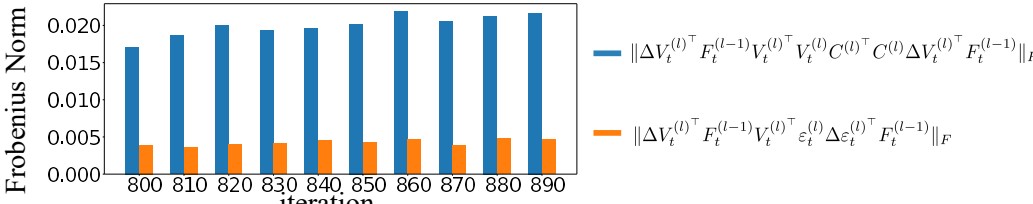

Figure 38: Visualization of the Frobenius norm of the two components $\Delta V_t^{(l)^\top} F_t^{(l-1)} V_t^{(l)^\top} V_t^{(l)} C^{(l)^\top} C^{(l)} \Delta V_t^{(l)^\top} F_t^{(l-1)}$ and $\Delta V_t^{(l)^\top} F_t^{(l-1)} V_t^{(l)^\top} \varepsilon_t^{(l)} \Delta \varepsilon_t^{(l)^\top} F_t^{(l-1)}$. We trained a 9-layer MLP on the MNIST dataset, where each layer had 512 neurons. Iterations were chosen at the end of the first phase.

Thus,

$$\text{sign}(\cos(\Delta V_t^{(l)}, F_t^{(l-1)}) \cdot \cos(V_t^{(l)}, \dot{F}_t^{(l-1)})) \leq 0 \tag{25}$$

In this paper, we approximately consider $\Delta F_t^{(l-1)}$ and $\dot{F}_t^{(l-1)}$ are negatively parallel to each other. Thus, we have $\text{sign}(\cos(\Delta V_t^{(l)}, F_t^{(l-1)}) \cdot \cos(V_t^{(l)}, \Delta F_t^{(l-1)})) = \text{sign}(\cos(\Delta V_t^{(l)}, F_t^{(l-1)}) \cdot (-\cos(V_t^{(l)}, \dot{F}_t^{(l-1)}))) \geq 0$.

## L   PROOF FOR THEOREM 2

In this section, we aim to prove that training samples of the same category have the same effect in the first phase.

**Theorem 2.** *Under the aforementioned background assumption, for any training samples $x, x' \in X_c$ in the category c, if $[C^{(l)^\top} D_t^{(l)}|_x \dot{F}_t^{(l)}|_x] \cdot [C^{(l)^\top} D_t^{(l)}|_{x'} \dot{F}_t^{(l)}|_{x'}] > 0$ (i.e., $F_t^{(l)}|_x$ and $F_t^{(l)}|_{x'}$ have kinds of similarity in very early iterations), then $\cos(\alpha_c \Delta V_t^{(l)}|_x, F_t^{(l-1)}|_x) \geq 0$, and $\cos(\alpha_c V_t^{(l)}, \Delta F_t^{(l-1)}|_x) \geq 0$, where $\alpha_c \in \{-1, +1\}$ is a constant shared by all samples in category c.*

*proof.*   Given a sample $x$ and a sample $x'$ from the same category, we can prove that $\cos(\Delta V_t^{(l)}|_x, F_t^{(l-1)}|_x) \cdot \cos(\Delta V_t^{(l)}|_{x'}, F_t^{(l-1)}|_{x'}) \geq 0$.

$\text{sign}(\cos(\Delta V_t^{(l)}|_x, F_t^{(l-1)}|_x) \cdot \cos(\Delta V_t^{(l)}|_{x'}, F_t^{(l-1)}|_{x'}))$

$= \text{sign}([\Delta V_t^{(l)^\top}|_x F_t^{(l-1)}|_x] \cdot [\Delta V_t^{(l)^\top}|_{x'} F_t^{(l-1)}|_{x'}])$

$= \text{sign}([\frac{C^{(l)^\top} \Delta W_t^{(l)}|_x}{C^{(l)^\top} C^{(l)}} F_t^{(l-1)}|_x] \cdot [\frac{C^{(l)^\top} \Delta W_t^{(l)}|_{x'}}{C^{(l)^\top} C^{(l)}} F_t^{(l-1)}|_{x'}])$

$= \text{sign}([C^{(l)^\top} \Delta W_t^{(l)}|_x F_t^{(l-1)}|_x] \cdot [C^{(l)^\top} \Delta W_t^{(l)}|_{x'} F_t^{(l-1)}|_{x'}])$

$= \text{sign}([C^{(l)^\top}(-\eta D_t^{(l)}|_x \dot{F}_t^{(l)}|_x F_t^{(l-1)^\top}|_x) F_t^{(l-1)}|_x] \cdot [C^{(l)^\top}(-\eta D_t^{(l)}|_{x'} \dot{F}_t^{(l)}|_{x'} F_t^{(l-1)^\top}|_{x'}) F_t^{(l-1)}|_{x'}])$

$= \text{sign}([C^{(l)^\top} D_t^{(l)}|_x \dot{F}_t^{(l)}|_x] \cdot [C^{(l)^\top} D_t^{(l)}|_{x'} \dot{F}_t^{(l)}|_{x'}])$

$\tag{26}$

According to the assumption that $F_t^{(l)}|_x$ and $F_t^{(l)}|_{x'}$ have kinds of similarity, we can consider $[C^{(l)^\top} D_t^{(l)}|_x \dot{F}_t^{(l)}|_x] \cdot [C^{(l)^\top} D_t^{(l)}|_{x'} \dot{F}_t^{(l)}|_{x'}] > 0$. In this way, for the category $c$, there exists a constant $\alpha_c$, which satisfies $\text{sign}(\cos(\alpha_c \Delta V_t^{(l)}|_x, F_t^{(l-1)}|_x) \geq 0$, where $\alpha_c \in \{-1, +1\}$ and training sampl e $x \in X_c$ belongs to the category $c$.

According to Lemma 3, we have $\cos(\Delta V_t^{(l)}|_x, F_t^{(l-1)}|_x) \cdot \cos(V_t^{(l)}, \Delta F_t^{(l-1)}|_x) \geq 0$. Thus, we have $\text{sign}(\cos(\alpha_c \Delta V_t^{(l)}|_x, F_t^{(l-1)}|_x) \cdot \cos(\alpha_c V_t^{(l)}, \Delta F_t^{(l-1)}|_x)) \geq 0$. In addition, the above proof indicates that $\text{sign}(\cos(\alpha_c \Delta V_t^{(l)}|_x, F_t^{(l-1)}|_x) \geq 0$. Therefore, we have $\text{sign}(\cos(\alpha_c V_t^{(l)}|_x, \Delta F_t^{(l-1)}|_x) \geq 0$

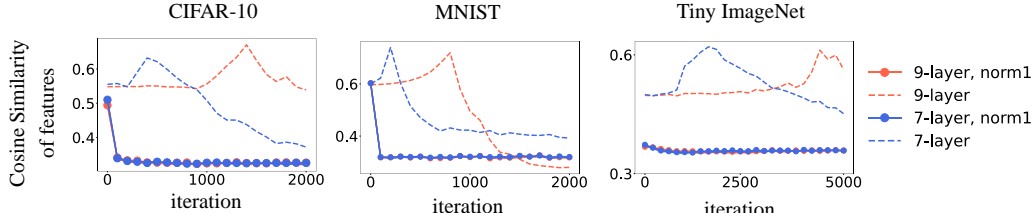

Figure 39: Cosine similarity of features between samples in different categories. We trained 7-layer MLPs and 9-layer MLPs on the CIFAR-10, the MNIST, and the Tiny ImageNet dataset.

# M    DISCUSSION FOR FOUR TYPICAL OPERATIONS

## M.1    CENTERING OPERATIONS FOR NORMALIZATION

The output feature of the $l$-th linear layer *w.r.t.* the input sample $x$ can be described as $[f_1, f_2, \ldots, f_h] = W_t^{(l)} F_t^{(l-1)} \in \mathbb{R}^h$, where $f_i$ denotes the $i$-th dimension of the feature. In this way, the batch normalization operation can be formulated as $BN(f_i) = \gamma_{\text{scale}}[(f_i - \mu_i)/\sigma_i] + \beta_{\text{shift}}$, where $\gamma_{\text{scale}}$ and $\beta_{\text{shift}}$ denote the scaling and the shifting parameters, respectively. In this way, the batch normalization operation subtracts the mean feature $\bar{F}_t^{(l)} = \mathbb{E}_{x \in X}[F_t^{(l)}|_x]$ from features of all samples. Therefore, features of different samples in a same category are no longer similar to each other.

We also propose a simplified normalization operation (*i.e.*, centering operations for normalization) to alleviate the TFC phenemonon in the first phase. The centering operations for normalization is given as $norm_1(f_i) = (f_i - \mu_i)/\sigma_i$, where $\mu_i$ and $\sigma_i$ denote the mean value and the standard deviation of $f_i$ over different samples, respectively. This operation is similar to the batch normalization (Ioffe & Szegedy, 2015), but we do not compute the scaling and shifting parameters in the batch normalization.

In order to verify the centering operations for normalization can alleviate the TFC phenemonon during the training process of the MLP, we trained 7-layer MLPs and 9-layer MLPs with and without the centering operations. Specifically, for the centering normalization operation $norm_1$, we added the centering operations after each linear layer, except the last linear layer. Each linear layer in the MLP had 512 neurons. Figure 39 shows that the feature similarity in MLPs with centering operations kept decreasing, while the feature similarity of the MLP without centering operations kept increasing. This indicated that centering operations for normalization alleviate the TFC phenomenon.

## M.2    MOMENTUM

We can explain that momentum in gradient descent can alleviate this phenomenon. Based on Lemma 3, the "self-enhanced system" of the TFC phenemonon requires singular values of weights along other directions $\varepsilon_t^{(l)}$ to be small enough. However, because the momentum operation strengthens influences of the initialized noisy weights $W_{t=0}^{(l)}$, it strengthens singular values of $\varepsilon_t^{(l)}$, to some extent, thereby alleviating the TFC phenemonon.

Specifically, considering the momentum with the coefficient $m$, the dynamics of weights $W_{t+1}$ can be described as,

$$W_{t+1} = W_t - \eta \frac{\partial Loss}{\partial W_t} - m \frac{\partial Loss}{\partial W_{t-1}}, \tag{27}$$

where $\eta$ denotes the learning rate. Because we only focus on weights in a single layer, without causing ambiguity, we omit the superscript $(l)$ to simplify the notation in this subsection. In this way, we can write the gradient descent as

$$W_{T+1} = W_0 + \eta \sum_t^T \frac{1 - m^{T+1-t}}{1 - m} \frac{\partial Loss}{\partial W_t}. \tag{28}$$

Since $0 < m < 1$, the coefficient $\frac{1-m^{T+1-t}}{1-m}$ decreases when the variable $t$ increases. Thus, a large $m$ represents that influences of $W_0$ on $W_{T+1}$ are significant. Because $\varepsilon_{T+1}$ is decomposed from

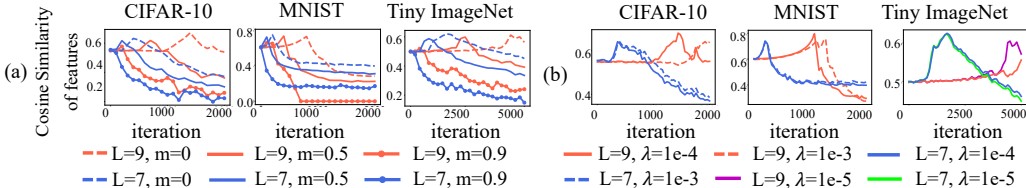

Figure 40: Effects of (a) momentum and (b) $L_2$ regularization. We trained $L$-layer MLPs, where each layer had 512 neurons. A shorter first phase indicates that the TFC phenomenon is more alleviated.

$W_{T+1}$ and singular values of $\varepsilon_{T+1}$ are mainly determined by the noisy $W_0$. Accordingly, singular values of $\varepsilon_{T+1}$ are relatively large, which disturb the "self-enhanced system" and alleviate the TFC phenemonon.

To verify the above analysis, we trained MLPs with $m = 0, 0.5, 0.9$, respectively. Figure 40(a) verifies that a larger value of $m$ usually more alleviates the TFC phenomenon.

### M.3 INITIALIZATION

We explain that the initialization of MLPs also affects the TFC phenemonon. According to Lemma 3, such "self-enhanced system" requires singular values of weights along other directions $\varepsilon_t^{(l)}$ to be small enough. However, because increasing the variance of the initialized weights $W_0^{(l)}$ will increase singular values of $\varepsilon_t^{(l)}$ based on Lemma 2, alleviating the TFC phenemonon. Specifically, we initialize weights with Xavier normal distribution (Glorot & Bengio, 2010), *i.e.* $W_0 \sim \mathcal{N}(\mathbf{0}, \gamma \sigma_{\text{var}}^2 I)$, where $\sigma_{\text{var}} = \sqrt{\frac{2}{fan_{\text{out}} + fan_{\text{in}}}}$. $fan_{\text{in}}$ and $fan_{\text{out}}$ denote the input dimension and the output dimension of the linear layer, respectively. In this way, a large $\gamma$ yields large singular values of initial weights $W_0$. Based on Lemma 2, we also have $\varepsilon_0^{(l)} = W_0^{(l)^\top} - W_0^{(l)^\top} \frac{C^{(l)} C^{(l)^\top}}{C^{(l)^\top} C^{(l)}}$. Large singular values of initial weights $W_0$ lead to large singular values of $\varepsilon_0^{(l)}$. Therefore, a large variance of initialized weights disturbs the "self-enhanced system" and alleviates the TFC phenemonon.

### M.4 $L_2$ REGULARIZATION (RIDGE LOSS)

$L_2$ regularization is equivalent to the weight decay in the case of gradient descent. The total loss is given as $\mathcal{L}(W_t) = \mathcal{L}^{CE}(W_t) + \lambda \|W_t\|_2^2$, where $\mathcal{L}^{CE}(W_t)$ represents the cross entropy loss, and $\lambda \|W_t\|_2^2$ denotes the ridge loss. In this way, we have the following iterates by using gradient descent

$$
\begin{aligned}
W_{t+1} &= W_t - \eta \nabla \mathcal{L}_t(W_t) \\
&= W_t - \eta \nabla \mathcal{L}_t^{CE}(W_t) - 2\eta\lambda W_t \\
&= (1 - 2\eta\lambda) W_t - \eta \nabla \mathcal{L}_t^{CE}(W_t),
\end{aligned}
\tag{29}
$$

According to Lemma 3, such "self-enhanced system" requires singular values of weights along other directions $\varepsilon_t^{(l)}$ to be small enough. Based on Lemma 2, we also have $\varepsilon_t^{(l)} = W_t^{(l)^\top} - W_t^{(l)^\top} \frac{C^{(l)} C^{(l)^\top}}{C^{(l)^\top} C^{(l)}}$. In this way, a larger $\lambda$ yields smaller singular values of $\varepsilon_t^{(l)}$, which disturbs the "self-enhanced system" and strengthens the TFC phenemonon. Figure 40(b) Figure 9(d) in the main paper verify that a larger coefficient $\lambda$ more strengthened the TFC phenemonon.

# N EXPLANATIONS FOR MORE DNNS.

The theoretical analysis of this study can explain which kinds of DNNs are more likely to exhibit the TFC phenomenon in early epochs. In fact, we discovered the two-phase phenomenon and the TFC phenomenon in various DNNs, including MLPs and modern CNNs, *e.g.*, VGG-11 models and VGG-13 models. Specifically, we trained VGG-11 models and VGG-13 models on the CIFAR-10 dataset and the Tiny ImageNet dataset. We adopted the learning rate $\eta = 0.01$, the batch size $bs = 100$, and the SGD optimizer. The training loss, the testing loss, the training accuracy, and the testing accuracy are shown in Figure 41 and Figure 42. Figure 41(e) and Figure 42(e) show that VGGs exhibited TFC phenomenoa in practice.

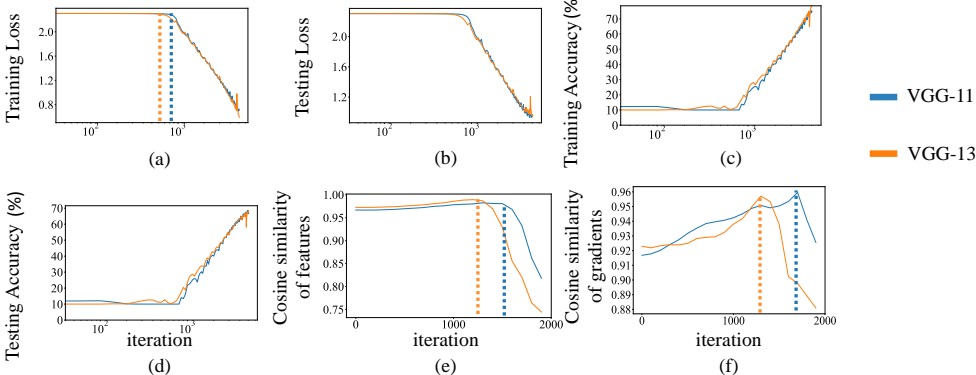

Figure 41: (a) The training loss of a VGG-11 model and a VGG-13 model trained on the CIFAR-10 dataset. (b) The testing loss of two models. (c) Training accuracies of two models. (d) Testing accuracies of two models. (e) Cosine similarity between features of different categories. (f) Cosine similarity between gradients of different samples in a category. The feature and the feature gradient were used in the third linear layer of the MLP in models.

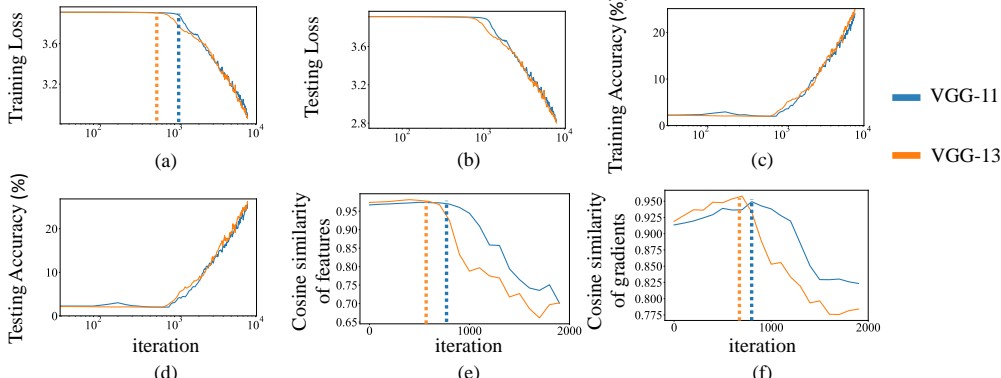

Figure 42: (a) The training loss of a VGG-11 model and a VGG-13 model trained on the Tiny ImageNet dataset. (b) The testing loss of two models. (c) Training accuracies of two models. (d) Testing accuracies of two models. (e) Cosine similarity between features of different categories. (f) Cosine similarity between gradients of different samples in a category. The feature and the feature gradient were used in the third linear layer of the MLP in models.

**Furthermore, we found that our theoretical analysis can be generalized to modern CNNs and transformers.** We conducted experiments on ResNet-18, ResNet-34 (He et al., 2015), and Vision Transformers (ViTs) (Dosovitskiy et al., 2020). Because both ResNets and ViTs were the two most classical network architectures that had been examined for years, it showed that ResNet-18, ResNet-34, and ViT did not exhibit the TFC phenomenon (or the TFC phenomenon only existed in very few iterations within the first epoch), owing to the use of normalization operations in these

DNNs. However, according to our theoretical analysis, if the batch normalization (BN) operations in ResNet-18/34 and the layer normalization (LN) operations in ViTs were removed, then the TFC phenomenon was significantly strengthened.

First, we trained ViTs, ResNet-18, and ResNet-34 models on the CIFAR-10 dataset. The classification heads in both ViTs and ResNet-18/34 were implemented by 4-layer MLP. Specifically, we trained two different ViTs with the patch size $P = 4$, the heads = 18, the dropout rate = 0.1, the embedding dropout rate = 0.1, the learning rate $\eta = 0.1$, the batch size $bs = 100$, and the SGD optimizer. For ResNet-18 and ResNet-34 models, we adopted the learning rate $\eta = 0.01$, the batch size $bs = 100$, and the SGD optimizer. Besides, we used two data augmentation methods, including random cropping and random horizontal flipping. The training loss, the testing loss, the training accuracy, and the testing accuracy are shown in **blue curves** in Figure 43, Figure 44, Figure 45 and Figure 46. These figures verify that ResNets and ViTs' first phases were very short, and the TFC phenomenon only existed in a few iterations, which could be ignored.

Second, in comparison, we further constructed four baseline networks by removing the BN layer from ResNet-18/34 and removing LN layers from ViTs. **Orange curves** in Figure 43, Figure 44, Figure 45 and Figure 46 verify that such new ResNet-18/34 and new ViTs exhibited a significant TFC phenomenon.

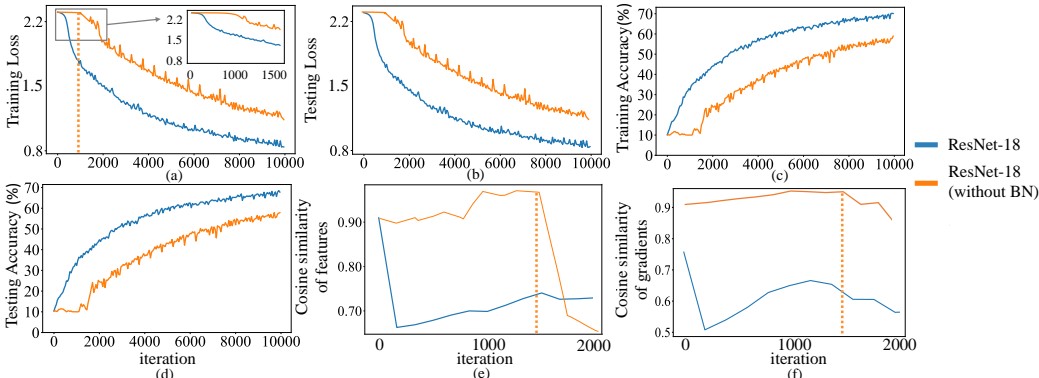

Figure 43: (a) The training loss of a ResNet-18 and a ResNet-18 (without BN) trained on the CIFAR-10 dataset. (b) The testing loss of two models. (c) Training accuracies of two models. (d) Testing accuracies of two models. (e) Cosine similarity between features of different categories. (f) Cosine similarity between gradients of different samples in a category. The feature and the feature gradient were used in the third linear layer of the MLP in models.

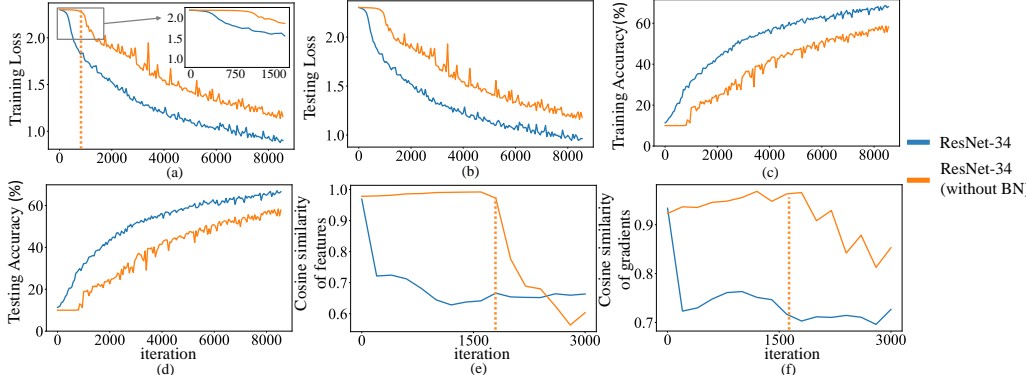

Figure 44: (a) The training loss of a ResNet-34 and a ResNet-34 (without BN) trained on the CIFAR-10 dataset. (b) The testing loss of two models. (c) Training accuracies of two models. (d) Testing accuracies of two models. (e) Cosine similarity between features of different categories. (f) Cosine similarity between gradients of different samples in a category. The feature and the feature gradient were used in the third linear layer of the MLP in models.

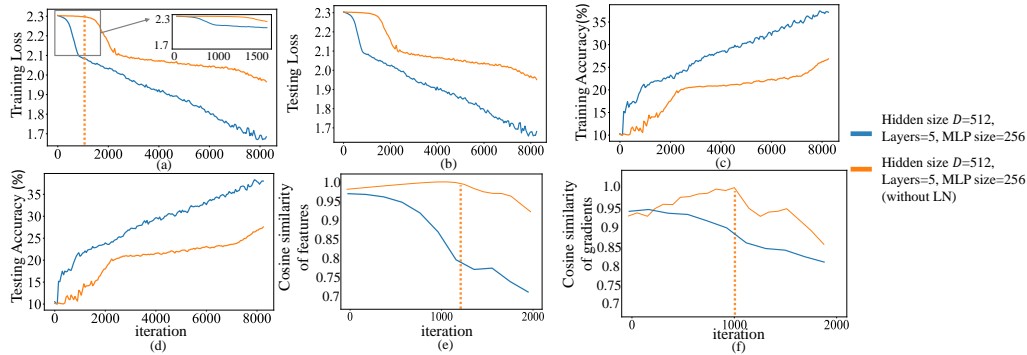

Figure 45: (a) The training loss of a ViT and a ViT (without LN) trained on the CIFAR-10 dataset. (b) The testing loss of two models. (c) Training accuracies of two models. (d) Testing accuracies of two models. (e) Cosine similarity between features of different categories. (f) Cosine similarity between gradients of different samples in a category. The feature and the feature gradient were used in the third linear layer of the MLP in models.

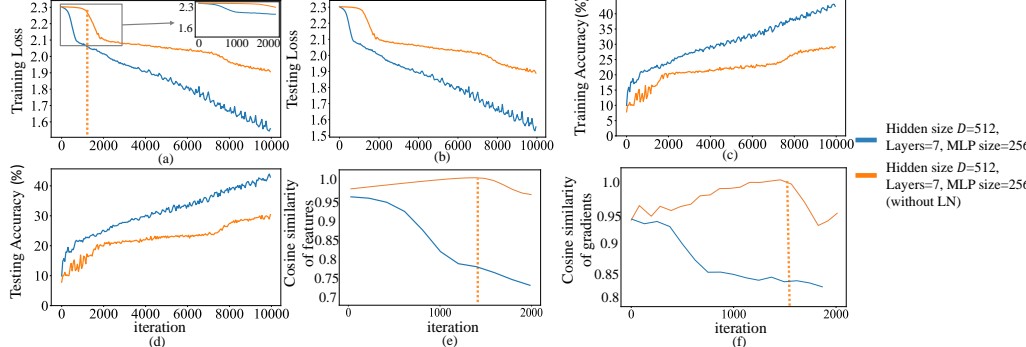

Figure 46: (a) The training loss of a ViT and a ViT (without LN) trained on the CIFAR-10 dataset. (b) The testing loss of two models. (c) Training accuracies of two models. (d) Testing accuracies of two models. (e) Cosine similarity between features of different categories. (f) Cosine similarity between gradients of different samples in a category. The feature and the feature gradient were used in the third linear layer of the MLP in models.

## O   MORE EXPERIMENTAL RESULTS OF ASSUMPTION 1.

**Assumption 1.** *We assume that the MLP encodes features of very few (a single or two) categories in the first phase, instead of simultaneously learning all or most categories in this phase.*

In this section, we aim to verify that Assumption 1 is a common fact in various DNNs, including MLPs, VGGs, and ResNets. To this end, we have conducted new experiments to show that DNNs encoded features of very few (a single or two) categories in early epochs. Specifically, we trained a 9-layer MLP on the CIFAR-10, the MNIST dataset, and the Tiny ImageNet dataset, respectively. Each layer of the MLP had 512 neurons. Besides, We trained a VGG-11 model, a VGG-13 model, and a ResNet-18 on the CIFAR-10 dataset. We evaluated the training accuracy at the end of the first phase. For the Tiny ImageNet dataset, we randomly selected the following 50 categories, *orangutan, parking meter, snorkel, American alligator, oboe, basketball, rocking chair, hopper, neck brace, candy store, broom, seashore, sewing machine, sunglasses, panda, pretzel, pig, volleyball, puma, alp, barbershop, ox, flagpole, lifeboat, teapot, walking stick, brain coral, slug, abacus, comic book, CD player, school bus, banister, bathtub, German shepherd, black stork, computer keyboard, tarantula, sock, Arabian camel, bee, cockroach, cannon, tractor, cardigan, suspension bridge, beer bottle, viaduct, guacamole*, and *iPod* for training. Figure 47, Figure 48, Figure 49, and Figure 50 show that various DNNs encoded features of very few (a single or two) categories in early epochs.

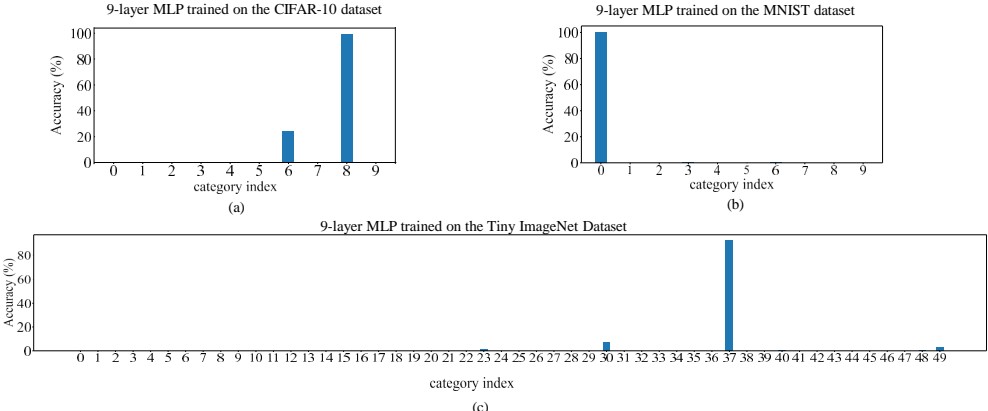

Figure 47: The training accuracies of MLPs on the CIFAR-10 dataset, the MNIST dataset, and the Tiny ImageNet dataset. The accuracies were evaluated at the end of the first phase. MLPs encode features of very few (a single or two) categories in the first phase, instead of simultaneously learning all or most categories in this phase. (a) The training accuracy of a 9-layer MLP trained on the CIFAR-10 dataset. (b) The training accuracy of a 9-layer MLP trained on the MNIST dataset. (c) The training accuracy of a 9-layer MLP trained on the Tiny ImageNet dataset.

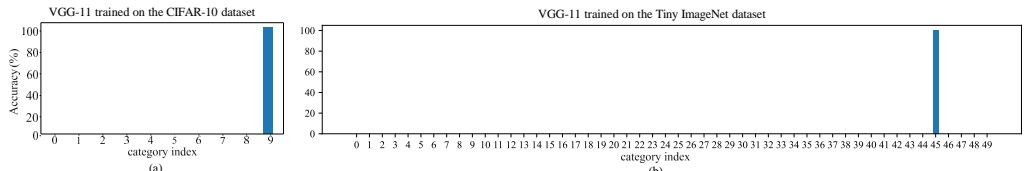

Figure 48: The training accuracies of VGG-11 models on the CIFAR-10 dataset and the Tiny ImageNet dataset. The accuracies were evaluated at the end of the first phase. VGG-11 models encode features of very few (a single or two) categories in the first phase, instead of simultaneously learning all or most categories in this phase. (a) The training accuracy of VGG-11 models trained on the CIFAR-10 dataset. (b) The training accuracy of VGG-11 models trained on the Tiny ImageNet dataset.

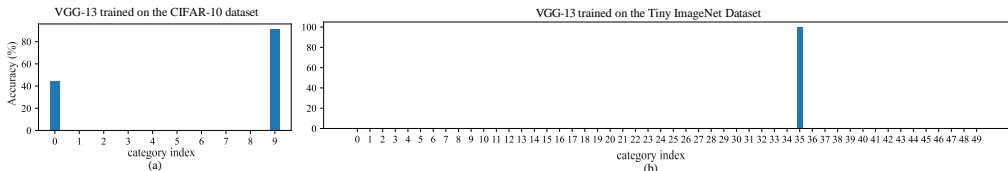

Figure 49: The training accuracies of VGG-13 models on the CIFAR-10 dataset and the Tiny ImageNet dataset. The accuracies were evaluated at the end of the first phase. VGG-13 models encode features of very few (a single or two) categories in the first phase, instead of simultaneously learning all or most categories in this phase. (a) The training accuracy of VGG-13 models trained on the CIFAR-10 dataset. (b) The training accuracy of VGG-13 models trained on the Tiny ImageNet dataset.

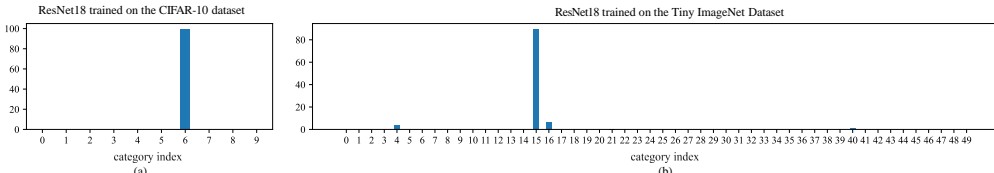

Figure 50: The training accuracies of ResNet-18 models on the CIFAR-10 dataset and the Tiny ImageNet dataset. The accuracies were evaluated at the end of the first phase. ResNet-18 models encode features of very few (a single or two) categories in the first phase, instead of simultaneously learning all or most categories in this phase. (a) The training accuracy of ResNet-18 models trained on the CIFAR-10 dataset. (b) The training accuracy of ResNet-18 models trained on the Tiny ImageNet dataset.

## P    PROPOSE AN IMPROVED TRAINING METHOD

In this section, we use our theory to develop a new normalization method. The new normalization operation was designed considering the following two findings.
• Our theoretical analysis told us that the centering operation in BN could alleviate the TFC phenomenon.
• Previous studies found some shortcomings of the BN operation, *i.e.*, the BN operation usually caused unstable features. Thus, the BN operation was found incompatible with the dropout (Li et al., 2019), hurt the classification accuracy in adversarial training (Galloway et al., 2019), and decreased the quality of images generated by generative models (Salimans et al., 2016).

Therefore, according to our analysis, we only need to update the dynamic normalization parameters (*i.e.*, $\mu_i$ and $\sigma_i$ in the following equation) in the first phase to avoid the learning-sticking problem, instead of applying the dynamic normalization parameters in the entire training process. In this way, we can simultaneously solve the learning-sticking problem and avoid unstable features.

Specifically, we are given the output feature $F = [f_1, f_2, \ldots, f_h] \in \mathbb{R}^h$ of the $l$-th linear layer *w.r.t.* the input sample $x$, where $f_i$ denotes the $i$-th dimension of the feature. The new normalization operation is given as

$$\text{norm}(f_i) = (f_i - \mu_i)/\sigma_i, \tag{30}$$

where $\mu_i$ and $\sigma_i$ denote the mean value and the standard deviation of $f_i$ over different samples, respectively. We only update the mean value $\mu_i$ and the standard deviation $\sigma_i$ in the first phase, as follows.

$$\mu_i = \begin{cases} \mathbb{E}_{x \in \text{batch}}[f_i], a_t > \tau \\ \mu_{i,t-1}, a_t \leq \tau \end{cases}, \quad \sigma_i^2 = \begin{cases} \text{Var}_{x \in \text{batch}}[f_i], a > \tau \\ \sigma_{i,t-1}^2, a \leq \tau \end{cases}, \tag{31}$$

where we keep updating $a_t = 0.99a_{t-1} + 0.01\mathbb{E}_{x,x' \in \text{batch}}[\cos(F|_x, F|_{x'})]$ through all the $t$ previous batches to represent the current cosine similarity between features of different samples. If $a_t$ is greater than a threshold $\tau = 0.3$, then we consider the learning process to be in the first phase and normalize the feature. Otherwise, if $a_t \leq \tau$, then we consider it has already jumped to the second phase, stop updating $\mu_i$ and $\sigma_i^2$, and use constants $\mu_i$ and $\sigma_i^2$ to generate stable features. We set $m = 0.1$ and compute $\mu_{i,t}$ and $\sigma_{i,t}^2$ in the $t$-th batch as follows.

$$\mu_{i,t} = \begin{cases} (1-m)\mu_{i,t-1} + m\mathbb{E}_{x \in \text{batch}}[f_i], a_t > \tau \\ \mu_{i,t-1}, a_t \leq \tau \end{cases}, \quad \sigma_{i,t}^2 = \begin{cases} (1-m)\sigma_{i,t-1}^2 + m\text{Var}_{x \in \text{batch}}[f_i], a > \tau \\ \sigma_{i,t-1}^2, a \leq \tau \end{cases}. \tag{32}$$

To this end, we conducted experiments on two types of MLPs (*i.e.*, 9-layer MLPs and 11-layer MLPs) to compare the proposed method with BN. For each type of MLP, we trained three versions MLPs on the CIFAR-10 dataset. The vanilla MLP had 512 neurons in each layer. We added the proposed norm operation after the first, the third, the fifth, and the seventh linear layers, and constructed the network *MLP-norm*. For a fair comparison, we constructed a baseline MLP, namely *MLP-BN*, by adding the BN operation in the same positions as in *MLP-norm*. In addition, scaling and shifting parameters in the BN operation were closed. Figure 51 shows that both the *MLP-norm* and *MLP-BN* alleviated the learning-sticking problem. However, *MLP-norm* was optimized much faster than *MLP-BN*, because our theoretical analysis told us that it was not necessary to continue updating $\mu_i$ and $\sigma_i^2$, if the learning process did not have a risk of feature collapse, thereby alleviating the optimization problems found in (Li et al., 2019; Galloway et al., 2019).

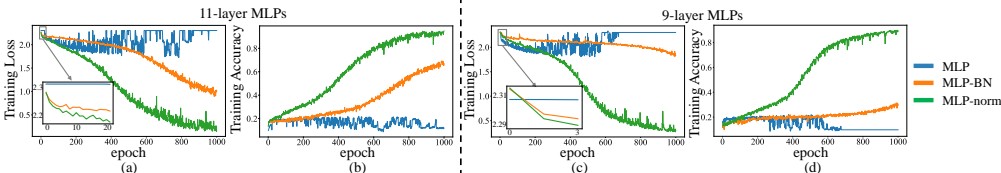

Figure 51: (a) The training loss of three 11-layer MLPs trained on the CIFAR-10 dataset, where each layer had 512 neurons. (b) The training accuracies of three 11-layer MLPs. (c) The training loss of three 9-layer MLPs trained on the CIFAR-10 dataset, where each layer had 512 neurons. (d) The training accuracy of three 9-layer MLPs. Note that the vibration of the blue curve could be explained as the failure of jumping out of the first phase, due to the strong power of the "self-enhancement system."

