# OpenReview forum: "Temporary feature collapse phenomenon in early learning of MLPs"
_ICLR.cc/2023/Conference — Submitted to ICLR 2023_

### Official Review · Reviewer_nGWd · 2022-10-23

**Confidence:** 3
**Correctness:** 3
**Technical Novelty And Significance:** 2
**Empirical Novelty And Significance:** 3
**Recommendation:** 6

**Clarity, Quality, Novelty And Reproducibility:**

The presentation is quite clear. The work has some merits.
The code is provided.


**Strength And Weaknesses:**

#Strength: the work is well presented and well written. It discovers an interesting phenomenon empirically with extensive experiments, and investigates it theoretically: the phenomenon that different neurons in a layer are optimized towards a common direction in the first phase

It provides explanations of why four typical operations can alleviate the feature collapse:

#Weakness: The experiments are mostly done on MLP and VGG network architectures. More experiments should be done on more popular/modern architectures such as ResNet or even large Vision transformer networks.

The analysis is based upon gradient flow and quite strong assumptions.
Although it explains the benefits of four typical operations from the new perspective, it is unclear for providing practical guidance for improving optimization design. The paper can be significantly improved by introducing new methods for improved training.


**Summary Of The Paper:**

This paper studies the two-phase phenomena in training deep neural networks both empirically and theoretically. The work discovers and explains the reason for the feature collapse phenomenon in the first phase, i.e., the diversity of features over different samples keeps decreasing in the first phase, until samples of different categories share almost the same feature, which hurts the optimization of MLPs. It explains such a phenomenon in terms of the learning dynamics of MLPs. Additionally, the work theoretically analyzes the reason why four typical operations can alleviate the feature collapse.

**Summary Of The Review:**

Overall, this work is well-written and well-presented.
It studies an interesting TFC phenomenon in deep learning training, with quite extensive experiments and theoretical justifications.

It is a bit unclear on the universality and practical guidance of TFC.

---

> ### Author Response · Authors · 2022-11-17
> **Response to Reviewer nGWd (Part 1)**
>
> Thank you for your great efforts on the review of this paper. We will try our best to answer all your questions.
> Please let us know if you still have further concerns, or if you are not satisfied with the current responses, so that we can further update the response ASAP.
>
> ---
>
> Q1: Ask for more experiments on modern architectures. "... more experiments should be done on more popular/modern architectures such as ResNet or even large Vision transformer networks."
>
> A: Thanks. We have followed your suggestions to **conduct new experiments** and have found that our theoretical analysis can be generalized to CNNs and transformers in experiments (please see Section N of the supplementary material). We conducted experiments on ResNet-18, ResNet-34,  and Vision Transformers (ViTs). Because both ResNet-18, ResNet-34, and ViT were the two most classical network architectures that had been examined for years, it shows that ResNet-18, ResNet-34, and ViT did not exhibit the TFC phenomenon (or the TFC phenomenon only existed in very few iterations within the first epoch), owing to the use of normalization operations in these DNNs. However, according to our theoretical analysis, if the batch normalization (BN) operations in ResNet-18/34 and the layer normalization (LN) operations in ViTs were removed, then the TFC phenomenon was significantly strengthened.
>
> **First,** we trained ResNet-18, ResNet-34, and ViTs on the CIFAR-10 dataset. Figures 43 and 44 in Section N of the supplementary material verify that ResNet-18, ResNet-34, and ViT's first phases were very short, and the TFC phenomenon only existed in a few iterations, which could be ignored.
>
> **Second,** in comparison, we further constructed two baseline networks by removing the BN layer from ResNet-18/34 and removing LN layers from ViT. Figures 45 and 46 in Section N of the supplementary material verify that such new ResNet-18/34 and new ViT exhibited a significant TFC phenomenon.
>
> ---
> Q2: "The analysis is based upon gradient flow and quite strong assumptions."
>
> A: We mainly use two assumptions in this paper, and for each of these assumptions, we have conducted both extensive experiments and discussions to verify such assumptions were all quite natural in real applications.
>
> Specifically, the first assumption was that DNNs encoded features of very few (a single or two) categories in the first phase, which has been introduced in Assumption 1 on Page 8 of the main paper. To this end, experiments in Figure 7 and Section O in the supplementary material have shown that only a single or two categories exhibited much higher accuracies than the random guessing at the end of the first phase.
>
> Nevertheless, we have followed your suggestions to **conduct additional experiments** to verify the correctness of Assumption 1. To this end, we conducted experiments on various DNNs, including MLPs, VGGs, and ResNets. Figures 47-50 in Section O of the supplementary material show that DNNs encoded features of very few (a single or two) categories in early epochs.
>
> Then, the second assumption was that features of different samples have been pushed a little bit towards a specific common direction. To this end, experiments in Figure 2 and Section B in the supplementary material have shown that the cosine similarity of features of different categories was high. These experimental results indicated that features of different samples have been pushed towards the same direction.
>
> We welcome your feedback if you want even more experiments.

---

> ### Author Response · Authors · 2022-11-17
> **Response to Reviewer nGWd (Part 2)**
>
> Q3: Ask for new methods and more experiments to demonstrate the practical values of the discovery of the TFC phenomenon. "It is a bit unclear on the universality and practical guidance of TFC." "... it is unclear for providing practical guidance for improving optimization design. The paper can be significantly improved by introducing new methods for improved training."
>
> **Answer 1 of the two answers to Q3**: **Using our theory to develop new normalization methods and experimental verification.** Although existing experiments in Section C of the supplementary material have shown the utility of our findings, **we have followed your suggestions to design a new normalization operation based on our theoretical analysis of the TFC phenomenon, which could better solve the learning-sticking problem without causing the feature instability.** The new normalization operation was designed considering the following two findings.
> - Our theoretical analysis told us that the centering operation in BN could alleviate the TFC phenomenon.
> - Previous studies found some shortcomings of the BN operation, *i.e.*, the BN operation usually caused unstable features. Thus, the BN operation was found incompatible with the dropout [c1], hurt the classification accuracy in adversarial training [c2], and decreased the quality of images generated by generative models [c3].
>
> **Therefore, according to our analysis, we only need to update the dynamic normalization parameters** (***i.e.*, $\mu_i$ and $\sigma_i$ in the following equation) in the first phase to avoid the learning-sticking problem, instead of applying the dynamic normalization parameters in the entire training process. In this way, we can simultaneously solve the learning-sticking problem and avoid unstable features.**
>
> Specifically, we are given the output feature $F=[f_1,f_2,\ldots,f_h]$$\in \mathbb{R}^h$ of the $l$-th linear layer *w.r.t.* the input sample $x$, where $f_i$ denotes the $i$-th dimension of the feature. The new normalization operation is given as $$\textit{norm}(f_i)=(f_i-\mu_i)/\sigma_{i},$$where $\mu_i$ and $\sigma_i$ denote the mean value and the standard deviation of $f_i$ over different samples, respectively. Unlike the BN operation, we only update the mean value $\mu_i$ and the standard deviation $\sigma_i$ in the first phase, as follows.
>
> if $a_t>\tau$, we have $\mu_{i}=E_{x \in \text{batch}}[f_i]$ and $\sigma^2_{i}=Var_{x \in \text{batch}}[f_i]$; otherwise, $\mu_{i}=\mu_{i, t-1}$ and  $\sigma^2_{i}=\sigma_{i, t-1}^{2}$. We keep updating $a_{t}=0.99a_{t-1}+0.01E_{x,x' \in \text{batch}}[\cos(F|x,F|x')]$ in the $t$-th batch, to represent the current cosine similarity between features of different samples. If $a_t$ is greater than a threshold $\tau=0.3$, then we consider the learning process is in the first phase and normalize the feature, just like what the BN operation does. Otherwise, if $a_t\leq\tau$, then we consider it has already jumped to the second phase, stop updating $\mu_i$ and $\sigma_i^2,$ and use constant $\mu_i$ and $\sigma_i^2$ to generate stable features. In the $t$-th batch, $\mu_{i,t}$ and $\sigma_{i,t}^2$ are computed as follows. If $a_t>\tau$, we have $\mu_{i,t}=(1-m)\mu_{i,t-1}+mE_{x \in \text{batch}}[f_i]$ and $\sigma^2_{i,t}=(1-m)\sigma^2_{i, t-1}+mVar_{x \in \text{batch}}[f_i]$; otherwise, $\mu_{i, t}=\mu_{i, t-1}$ and  $\sigma^2_{i,t}=\sigma_{i, t-1}^{2}$, where  $m=0.1$.
>
> To this end, we **conducted experiments** on two types of MLPs (*i.e.*, 9-layer MLPs and 11-layer MLPs) to compare the proposed method with BN. For each type of MLP, we trained three versions MLPs on the CIFAR-10 dataset. The vanilla MLP had 512 neurons in each layer. We added the proposed norm operation after the first, the third, the fifth, and the seventh linear layers. layers, and constructed the network *MLP-norm*. For fair comparison, we constructed a baseline MLP, namely *MLP-BN*, by adding the BN operation in the same positions as in *MLP-norm*. Figure 51 in Section P of the supplementary material shows that both the *MLP-BN* and *MLP-norm* alleviated the learning-sticking problem. However, *MLP-norm* was optimized much more faster than MLP-BN, because our theoretical analysis told us that it was not necessary to continue updating $\mu_i$ and $\sigma_i^2$, if the learning process did not have a risk of feature collapse, thereby alleviating the optimization problems found in [c1, c2].
>
> [c1] Xiang Li, Shuo Chen, Xiaolin Hu, and Jian Yang. Understanding the disharmony between dropout and batch normalization by variance shift. In Computer Vision and Pattern Recognition, 2019.
> [c2] Angus Galloway, Anna Golubeva, Thomas Tanay, Medhat Moussa, and Graham W Taylor. Batch normalization is a cause of adversarial vulnerability. arXiv, abs/1905.02161, 2019.
> [c3] Tim Salimans, Ian Goodfellow, Wojciech Zaremba, Vicki Cheung, Alec Radford, and Xi Chen. Improved techniques for training gans. In Neural Information Processing Systems, 2016.

---

> ### Author Response · Authors · 2022-11-17
> **Response to Reviewer nGWd (Part 3)**
>
> Q3: Ask for new methods and more experiments to demonstrate the practical values of the discovery of the TFC phenomenon. "It is a bit unclear on the universality and practical guidance of TFC." "... it is unclear for providing practical guidance for improving optimization design. The paper can be significantly improved by introducing new methods for improved training."
>
> **Answer 2 of the two answers to Q3**: 2.**Further experiments to verify the effectiveness of techniques that are proven by our theoretical analysis.** Besides the above newly proposed methods, our theory also discovers a set of tricks to alleviate the TFC problem in Section 3.3. Here, we **conducted new experiments** to further verify the effectiveness of theoretical analysis in Section 3.3 in the main paper. Specifically, we trained three different types of DNNs on the MNIST, the CIFAR-10, and the Tiny ImageNet datasets for evaluation, including MLPs, VGGs, and ResNets. In experiments, these DNNs all suffered from the learning-sticking problem (*i.e.*, the loss minimization of these DNNs get stuck), when their initial weights were sampled from $N(\pmb{0}, \pmb{\Sigma}=\gamma_1 \sigma_{\textrm{var}}^2I )$. $\sigma_{\textrm{var}}^2$ was computed according to [c4] and $\gamma_1=0.1$. Figures 30-35 in Section C of the supplementary material verify the finding that we could solve the learning-sticking problem by increasing the variance of initial weights to $\gamma_2 \sigma_{\textrm{var}}^2I  (\gamma_2 = 1.0)$.
>
> 3.**Universality of the TFC phenomenon.** We have conducted extensive experiments to illustrate the universality of the TFC phenomenon on various DNNs, including MLPs, CNNs, and LSTMs. Please see Appendix B of the supplementary material.
>
> More crucially, **we have conducted new experiments to further illustrate the TFC phenomenon in the Vision Transformer (ViT).** Specifically,  according to our theoretical analysis, ViT did not exhibit the TFC phenomenon (or the TFC phenomenon only existed in very few iterations within the first epoch), owing to the layer normalization (LN) operations. However, if LN operations in ViTs were removed, then the TFC phenomenon was significantly strengthened, according to the analysis in Section 3.2. Please see Figure 46 in Section N of the supplementary material for experimental verification.
>
> [c4] Xavier Glorot and Yoshua Bengio. Understanding the difficulty of training deep feedforward neural networks. In Proceedings of the thirteenth international conference on artificial intelligence and statistics, pp. 249–256. JMLR Workshop and Conference Proceedings, 2010.

---

### Official Review · Reviewer_4cUL · 2022-10-24

**Confidence:** 3
**Correctness:** 3
**Technical Novelty And Significance:** 3
**Empirical Novelty And Significance:** 3
**Recommendation:** 6

**Clarity, Quality, Novelty And Reproducibility:**

Novelty
- Both the presented temporary feature collapse phenomenon and its explanation seem to be novel a provide a novel insight on the learning dynamics of early training phases.
Related work is cited and analyzed adequately.

Quality
- The claims are well-supported by the broad theoretical analysis and convincing experimental results.
The technical quality seems to be very good. However, I apologize for not being able to understand all the theory and proofs in depth and thus I don't feel capable to evaluate less obvious technical flaws if there are any.

Clarity
- The presentation as a whole is clear and comprehensible, although there are less comprehensible technical proofs.

**Strength And Weaknesses:**

PLus
- There are not many studies that look in depth on the early phase of DNNs training and that examine what exactly happens to the network
internal structure before the training error begins to decrease (and why this period is sometimes unexpectedly long).
The paper tries to fill in this gap.
- The paper shows a counter-intuitive (but possibly common) mechanism that causes a NN-model to get stuck in the early phase of training.
The idea presented is new and surprising to me.
- The provided theoretical (and empirical) analysis in broad and thorough, with convincing results.


Minus
- Although the temporary feature collapse phenomenon is interesting, the impact of the findings on NN applications and future research may be rather limited.
  The authors show that TFC can be easily avoided by commonly used techniques (clever weight initialization, momentum, l2 regularization, batch normalization)
  which implies that TFC phenomenon rarely occurs in current applications.

%%%%%%%%%%%%%%%%%%%%%%%%%%%%%%%%%%%%%%%%%%%%%

After rebuttal:

I have read the authors' comments, the new version of the paper and other reviews. I agree with most of the criticism regarding the original submission (e.g.,the need to test more NN-architectures, the need of real-world applications/impact, strong assumptions, and other theoretical analysis issues). On the other hand, I appreciate how the authors revised the paper to clarify these concerns. Most of the flaws are resolved in the final submission.

I think the work is novel, interesting and solid enough to be accepted despite its remaining flaws (mainly the relatively limited practical impact). However, since my score seems too optimistic compared to other reviewers with similar opinions on the latest version, I lower my score to 6 for consensus (although it is hard for me to decide between 6 and 8).


**Summary Of The Paper:**

The authors examine in depth the learning-sticking problem in early training of DNNs from the perspective of learning dynamics.
They offer a novel explanation for what often happens to network weights and inner activations during the problem - which they call "the temporary feature collapse" (TFC) phenomenon
They provide thorough theoretical (and empirical) analysis and justification of their idea.


**Summary Of The Review:**

A solid contribution. It presents a new phenomenon in early DNNs training and shows its possible explanation. The claims are supported by thorough theoretical and empirical analyses.
However, practical impact of the findings seems to be limited.

---

> ### Author Response · Authors · 2022-11-17
> **Response to Reviewer 4cUL**
>
> Thank you for your great efforts on the review of this paper. We will try our best to answer all your questions.
> Please let us know if you still have further concerns, or if you are not satisfied with the current responses, so that we can further update the response ASAP.
>
> ---
> Q1: About "the impact of the findings on NN applications and future research" and "practical impact of the findings?"
>
> A: Thank you. As discussed in Section 2 of the main paper, the TFC phenomenon has a potential connection with the learning-sticking problem, *i.e.*, when the deep neural networks (DNNs) are very deep, the loss minimization may get stuck, which can be considered as a strong first phase with an infinite length. Thus, alleviating the TFC phenomenon may directly solve the learning-sticking problem. Therefore, breaking the condition for the TFC phenomenon is crucial for the optimization of a DNN and is of significant values in practice.
>
> **New Experiments**. We have **conducted new experiments** to verify that alleviating the TFC phenomenon can solve the learning-sticking problem. Specifically, we trained three different types of DNNs on the MNIST, the CIFAR-10, and the Tiny ImageNet datasets for evaluation, including MLPs, VGGs, and ResNets. In experiments, these DNNs all suffered from the learning-sticking problem (*i.e.*, the loss minimization of these DNNs get stuck), when their initial weights were sampled from $N(\pmb{0}, \pmb{\Sigma}=\gamma_1 \sigma_{\textrm{var}}^2I )$. $\sigma_{\textrm{var}}^2$ was computed according to [c1] and $\gamma_1=0.1$. Figures 30-35 in Section C of the supplementary material verify that we could solve the learning-sticking problem by increasing the variance of initial weights to $\gamma_2 \sigma_{\textrm{var}}^2I  (\gamma_2 = 1.0)$. In contrast to being stuck in the first phase, the theoretical analysis of our study provided guidance for enabling normal training of a DNN without letting the performance be hurt by the learning-sticking problem.
>
> [c1] Xavier Glorot and Yoshua Bengio. Understanding the difficulty of training deep feedforward neural networks. In Proceedings of the thirteenth international conference on artificial intelligence and statistics, pp. 249–256. JMLR Workshop and Conference Proceedings, 2010.

---

> > ### Comment · Reviewer_4cUL · 2022-12-08
> > **Response to the authors**
> >
> > I thank the authors for their responses. I have also read the new version of the paper and other reviews. I agree with most of the concerns from the other reviews (e.g.,the need to test more NN-architectures, the need of real-world applications/impact, strong assumptions, and other theoretical analysis issues). On the other hand, I appreciate how the authors revised the paper to clarify these concerns.
> >
> > I think the work is novel and solid enough to be accepted despite its flaws (e.g., relatively low practical impact). However, since my score seems too optimistic compared to other reviewers with similar opinions, I am willing to lower my score to 6 for consensus.

---

> > > ### Author Response · Authors · 2022-12-08
> > > **Response to further questions of Reviewer 4cUL in the second round (Part 1)**
> > >
> > > Thank you for your comments. Maybe, it seems that you did not notice our responses to Reviewers nGWd, 7KPL, and 4Wyv. We have already conducted extensive experiments to answer these concerns, and other reviewers were satisfied with our responses and raised their scores. We repetitively asked reviewers whether they had new concerns, and these reviewers did not give new concerns. **Please specify the concern and let us know if you still have further concerns, so that we can update the response as soon as possible.**
> > >
> > > Thank you very much for your consideration.
> > >
> > > ---
> > >
> > > Q1: "the need to test more NN-architectures"
> > >
> > > A: Thanks. We **conducted new experiments** and have found that our theoretical analysis can be generalized to CNNs and transformers in experiments (please see Section N of the supplementary material). We conducted experiments on ResNet-18, ResNet-34, and Vision Transformers (ViTs). Because both ResNet-18, ResNet-34, and ViT were the two most classical network architectures that had been examined for years, it shows that ResNet-18, ResNet-34, and ViT did not exhibit the TFC phenomenon (or the TFC phenomenon only existed in very few iterations within the first epoch), owing to the use of normalization operations in these DNNs. However, according to our theoretical analysis, if the batch normalization (BN) operations in ResNet-18/34 and the layer normalization (LN) operations in ViTs were removed, then the TFC phenomenon was significantly strengthened.
> > >
> > > **First,** we trained ResNet-18, ResNet-34, and ViTs on the CIFAR-10 dataset. Figures 43 and 44 in Section N of the supplementary material verify that ResNet-18, ResNet-34, and ViT's first phases were very short, and the TFC phenomenon only existed in a few iterations, which could be ignored.
> > >
> > > **Second,** in comparison, we further constructed two baseline networks by removing the BN layer from ResNet-18/34 and removing LN layers from ViT. Figures 45 and 46 in Section N of the supplementary material verify that such new ResNet-18/34 and new ViT exhibited a significant TFC phenomenon.

---

> > > ### Author Response · Authors · 2022-12-08
> > > **Response to further questions of Reviewer 4cUL in the second round (Part 2)**
> > >
> > > Q2: "the need of real-world applications/impact"
> > >
> > > A: **Using our theory to develop new normalization methods and experimental verification.** Although existing experiments in Section C of the supplementary material have shown the utility of our findings, **we designed a new normalization operation based on our theoretical analysis of the TFC phenomenon, which could better solve the learning-sticking problem without causing the feature instability.** The new normalization operation was designed considering the following two findings.
> > > - Our theoretical analysis told us that the centering operation in BN could alleviate the TFC phenomenon.
> > > - Previous studies found some shortcomings of the BN operation, *i.e.*, the BN operation usually caused unstable features. Thus, the BN operation was found incompatible with the dropout [c1], hurt the classification accuracy in adversarial training [c2], and decreased the quality of images generated by generative models [c3].
> > >
> > > **Therefore, according to our analysis, we only need to update the dynamic normalization parameters (***i.e.***, $\mu_i$ and $\sigma_i$ in the following equation) in the first phase to avoid the learning-sticking problem, instead of applying the dynamic normalization parameters in the entire training process. In this way, we can simultaneously solve the learning-sticking problem and avoid unstable features.**
> > >
> > > Specifically, we are given the output feature $F=[f_1,f_2,\ldots,f_h]$$\in \mathbb{R}^h$ of the $l$-th linear layer *w.r.t.* the input sample $x$, where $f_i$ denotes the $i$-th dimension of the feature. The new normalization operation is given as $$\textit{norm}(f_i)=(f_i-\mu_i)/\sigma_{i},$$where $\mu_i$ and $\sigma_i$ denote the mean value and the standard deviation of $f_i$ over different samples, respectively. Unlike the BN operation, we only update the mean value $\mu_i$ and the standard deviation $\sigma_i$ in the first phase, as follows.
> > >
> > > if $a_t>\tau$, we have $\mu_{i}=E_{x \in \text{batch}}[f_i]$ and $\sigma^2_{i}=Var_{x \in \text{batch}}[f_i]$; otherwise, $\mu_{i}=\mu_{i, t-1}$ and  $\sigma^2_{i}=\sigma_{i, t-1}^{2}$. We keep updating $a_{t}=0.99a_{t-1}+0.01E_{x,x' \in \text{batch}}[\cos(F|x,F|x')]$ in the $t$-th batch, to represent the current cosine similarity between features of different samples. If $a_t$ is greater than a threshold $\tau=0.3$, then we consider the learning process is in the first phase and normalize the feature, just like what the BN operation does. Otherwise, if $a_t\leq\tau$, then we consider it has already jumped to the second phase, stop updating $\mu_i$ and $\sigma_i^2,$ and use constant $\mu_i$ and $\sigma_i^2$ to generate stable features. In the $t$-th batch, $\mu_{i,t}$ and $\sigma_{i,t}^2$ are computed as follows. If $a_t>\tau$, we have $\mu_{i,t}=(1-m)\mu_{i,t-1}+mE_{x \in \text{batch}}[f_i]$ and $\sigma^2_{i,t}=(1-m)\sigma^2_{i, t-1}+mVar_{x \in \text{batch}}[f_i]$; otherwise, $\mu_{i, t}=\mu_{i, t-1}$ and  $\sigma^2_{i,t}=\sigma_{i, t-1}^{2}$, where  $m=0.1$.
> > >
> > > To this end, we **conducted experiments** on two types of MLPs (*i.e.*, 9-layer MLPs and 11-layer MLPs) to compare the proposed method with BN. For each type of MLP, we trained three versions MLPs on the CIFAR-10 dataset. The vanilla MLP had 512 neurons in each layer. We added the proposed norm operation after the first, the third, the fifth, and the seventh linear layers. layers, and constructed the network *MLP-norm*. For fair comparison, we constructed a baseline MLP, namely *MLP-BN*, by adding the BN operation in the same positions as in *MLP-norm*. Figure 51 in Section P of the supplementary material shows that both the *MLP-BN* and *MLP-norm* alleviated the learning-sticking problem. However, *MLP-norm* was optimized much more faster than MLP-BN, because our theoretical analysis told us that it was not necessary to continue updating $\mu_i$ and $\sigma_i^2$, if the learning process did not have a risk of feature collapse, thereby alleviating the optimization problems found in [c1, c2].
> > >
> > > [c1] Xiang Li, Shuo Chen, Xiaolin Hu, and Jian Yang. Understanding the disharmony between dropout and batch normalization by variance shift. In Computer Vision and Pattern Recognition, 2019.
> > > [c2] Angus Galloway, Anna Golubeva, Thomas Tanay, Medhat Moussa, and Graham W Taylor. Batch normalization is a cause of adversarial vulnerability. arXiv, abs/1905.02161, 2019.
> > > [c3] Tim Salimans, Ian Goodfellow, Wojciech Zaremba, Vicki Cheung, Alec Radford, and Xi Chen. Improved techniques for training gans. In Neural Information Processing Systems, 2016.

---

> > > ### Author Response · Authors · 2022-12-08
> > > **Response to further questions of Reviewer 4cUL in the second round (Part 3)**
> > >
> > > Q3: About "strong assumptions."
> > >
> > > A: We mainly use two assumptions in this paper, and for each of these assumptions, we have conducted both extensive experiments and discussions to verify such assumptions were all quite natural in real applications.
> > >
> > > Specifically, the first assumption was that DNNs encoded features of very few (a single or two) categories in the first phase, which has been introduced in Assumption 1 on Page 8 of the main paper. To this end, experiments in Figure 7 and Section O in the supplementary material have shown that only a single or two categories exhibited much higher accuracies than the random guessing at the end of the first phase.
> > >
> > > Nevertheless, we **conducted additional experiments** to verify the correctness of Assumption 1. To this end, we conducted experiments on various DNNs, including MLPs, VGGs, and ResNets. Figures 47-50 in Section O of the supplementary material show that DNNs encoded features of very few (a single or two) categories in early epochs.
> > >
> > > Then, the second assumption was that features of different samples have been pushed a little bit towards a specific common direction. To this end, experiments in Figure 2 and Section B in the supplementary material have shown that the cosine similarity of features of different categories was high. These experimental results indicated that features of different samples have been pushed towards the same direction.

---

### Official Review · Reviewer_7KPL · 2022-10-26

**Confidence:** 3
**Correctness:** 3
**Technical Novelty And Significance:** 3
**Empirical Novelty And Significance:** 2
**Recommendation:** 6

**Clarity, Quality, Novelty And Reproducibility:**

This work has high quality and high clarity. This work has originality in explaining the TFC phenomenon of DNNs. The results should be reproducible because the authors provide codes in the supplementary material.

**Strength And Weaknesses:**

Strength:

(1)  The authors discover the common TFC phenomenon in early learning of the MLP.

(2)  The authors explain this phenomenon from the perspective of learning dynamics.

(3)  The authors explain why four types of operations, including normalization, momentum, initialization, and regularization, can alleviate the TFC phenomenon.

Weaknesses:

This work explains the TFC phenomenon from the perspective of learning dynamics. I have some questions as follows.

(1)  The authors argue that there exist two stages in the training process of DNNs, however, how to determine the percentage of two stages? If the first stage is short, what does this explanation tell us? And can you explain the second stage of the training process?

(2)  It seems that all the explanations are based on DNNs, so is it suitable for CNNs or the recent transformer models? Furthermore, it is also more valuable if the authors point out what it will bring to real-world applications.


**Summary Of The Paper:**

This work finds that the MLP exhibits a fundamental yet counter-intuitive TFC phenomenon in the early stage of the training process. After that, the authors explain this phenomenon from the perspective of learning dynamics and explain why four types of operations can alleviate the TFC phenomenon. Extensive theories and experiments are conducted to support the arguments of this work.

**Summary Of The Review:**

This work finds that in the early stage of the training process, the MLP exhibits a fundamental yet counter-intuitive TFC phenomenon and explains the reason why four typical operations can alleviate the TFC phenomenon. However, from my field, I cannot realize the value it can bring to the real world, and it seems that the theory analyses only focus on DNNs, which might not be suitable for CNNs or transformers. Could the authors give a further explanation because I am only partially in your field? Currently, I tend to vote marginally below the acceptance threshold but I am not an expert in the related field. I will hear more from other reviewers and the authors’ responses.


=== After rebuttal ===

I have read the authors' responses and other reviewers' comments. The responses solve my concerns well. I am willing to raise my score.

---

> ### Author Response · Authors · 2022-11-17
> **Response to Reviewer 7KPL (Part 1)**
>
> Thank you for your great efforts on the review of this paper. We will try our best to answer all your questions.
>
> Please let us know if you still have further concerns, or if you are not satisfied with the current responses, so that we can further update the response ASAP.
>
> ---
> Q1: Ask for clarifying the two-phase of the training process. "how to determine the percentage of two stages? If the first stage is short, what does this explanation tell us? And can you explain the second stage of the training process?"
>
> A: Thanks.
> - Because the two-phase of the training loss is well-known and common[c1, c2, c3, c4, c5, c6, c7, c8], this paper does not aim to define the boundary between the first phase and the second phase. Instead, the goal of this paper is just to explain the reason for the decrease of feature diversity in the first phase. To this end, we think that solving the TFC phenomenon in the first phase can help DNNs escape the first phase and enter the second phase. In this way, alleviating the TFC phenomenon guarantee the normal optimization of DNNs.
> - For the question "Can you explain the second stage of the training process?" The second phase can be explained as the normal training process, in which the training loss begins to decrease fast. Therefore, compared to the normal training process in the second phase, the first phase is abnormal. In this study, we mainly focus on the TFC phenomenon in the first phase, which has distinctive values for understanding DNNs' training processes.
> - For the question "If the first stage is short, what does this explanation tell us?" The significance of this study is to discover a set of conditions of strengthening/alleviating the TFC phenomenon, which potentially corresponds to causing/avoiding the learning-sticking problem. Specifically, when the DNN is very deep, when the task is difficult, when the variance of the initial weights is small, and when the DNN is learned without momentum or BN layers, the TFC phenomenon will probably appear according to our analysis in Section 3.3. To this end, the learning-sticking problem can be considered as a strong first phase with an infinite length. Furthermore, according to our theoretical analysis, we can lengthen/shorten the first phase of the training process by strengthening/alleviating the TFC phenomenon.
> -  We have followed your suggestions to **conduct new experiments** to verify that alleviating the TFC phenomenon can solve the learning-sticking problem. Specifically, we trained three different types of DNNs on the MNIST, the CIFAR-10, and the Tiny ImageNet datasets for evaluation, including MLPs, VGGs, and ResNets. In experiments, these DNNs all suffered from the learning-sticking problem (*i.e.*, the loss minimization of these DNNs get stuck), when their initial weights were sampled from $N(\pmb{0}, \pmb{\Sigma}=\gamma_1 \sigma_{\textrm{var}}^2I )$. $\sigma_{\textrm{var}}^2$ was computed according to [c9] and $\gamma_1=0.1$. Figures 30-35 in Section C of the supplementary material verify that we could solve the learning-sticking problem by increasing the variance of initial weights to $\gamma_2 \sigma_{\textrm{var}}^2I  (\gamma_2 = 1.0)$. In contrast to being stuck in the first phase, the theoretical analysis of our study provided guidance for enabling normal training of a DNN without letting the performance be hurt by the learning-sticking problem.
>
> [c1] Simsekli, Umut et al. “A tail-index analysis of stochastic gradient noise in deep neural networks.” International Conference on Machine Learning. PMLR, 2019.
> [c2] Saxe, Andrew M. et al. “Exact solutions to the nonlinear dynamics of learning in deep linear neural networks.” arXiv preprint arXiv:1312.6120 (2013).
> [c3] Tanaka, Daiki, et al. “Joint optimization framework for learning with noisy labels.” Proceedings of the IEEE conference on computer vision and pattern recognition. 2018.
> [c4] Vogl R K. Deep Learning Methods for Drum Transcription and Drum Pattern Generation/submitted by Richard Vogl[D]. Universitat Linz, 2018.
> [c5] Nguyen, Quynh, et al… “On the loss landscape of a class of deep neural networks with no bad local valleys.” ICLR (2019).
> [c6] Arab, Ali, et al. “A fast and fully-automated deep-learning approach for accurate hemorrhage segmentation and volume quantification in non-contrast whole-head CT.” Scientific Reports 10.1 (2020): 1-12.
> [c7] Jepkoech, Jennifer, et al. “The effect of adaptive learning rate on the accuracy of neural networks.” International Journal of Advanced Computer Science and Applications 12.8 (2021).
> [c8] Stevens, Eli, et al. Deep Learning with PyTorch. Manning Publications, 2020.
> [c9] Xavier Glorot and Yoshua Bengio. Understanding the difficulty of training deep feedforward neural networks. In Proceedings of the thirteenth international conference on artificial intelligence and statistics, pp. 249–256. JMLR Workshop and Conference Proceedings, 2010.

---

> ### Author Response · Authors · 2022-11-17
> **Response to Reviewer 7KPL (Part 3)**
>
> Q3: New experiments to demonstrate the practical value. "Furthermore, it is also more valuable if the authors point out what it will bring to real-world applications."
>
> **Answer 2 of the two answers to Q3**: **Using our theory to develop new normalization methods and experimental verification.** Although existing experiments in Section C of the supplementary material have shown the utility of our findings, **we have followed your suggestions to design a new normalization operation based on our theoretical analysis of the TFC phenomenon, which could better solve the learning-sticking problem without causing the feature instability.** The new normalization operation was designed considering the following two findings.
> - Our theoretical analysis told us that the centering operation in BN could alleviate the TFC phenomenon.
> - Previous studies found some shortcomings of the BN operation, *i.e.*, the BN operation usually caused unstable features. Thus, the BN operation was found incompatible with the dropout [c10], hurt the classification accuracy in adversarial training [c11], and decreased the quality of images generated by generative models [c12].
>
> **Therefore, according to our analysis, we only need to update the dynamic normalization parameters** (*i.e.*, **$\mu_i$ and $\sigma_i$ in the following equation) in the first phase to avoid the learning-sticking problem, instead of applying the dynamic normalization parameters in the entire training process. In this way, we can simultaneously solve the learning-sticking problem and avoid unstable features.**
>
> Specifically, we are given the output feature $F=[f_1,f_2,\ldots,f_h]$$\in \mathbb{R}^h$ of the $l$-th linear layer *w.r.t.* the input sample $x$, where $f_i$ denotes the $i$-th dimension of the feature. The new normalization operation is given as $$\textit{norm}(f_i)=(f_i-\mu_i)/\sigma_{i},$$where $\mu_i$ and $\sigma_i$ denote the mean value and the standard deviation of $f_i$ over different samples, respectively. Unlike the BN operation, we only update the mean value $\mu_i$ and the standard deviation $\sigma_i$ in the first phase, as follows.
>
> if $a_t>\tau$, we have $\mu_{i}=E_{x \in \text{batch}}[f_i]$ and $\sigma^2_{i}=Var_{x \in \text{batch}}[f_i]$; otherwise, $\mu_{i}=\mu_{i, t-1}$ and  $\sigma^2_{i}=\sigma_{i, t-1}^{2}$. We keep updating $a_{t}=0.99a_{t-1}+0.01E_{x,x' \in \text{batch}}[\cos(F|x,F|x')]$ in the $t$-th batch, to represent the current cosine similarity between features of different samples. If $a_t$ is greater than a threshold $\tau=0.3$, then we consider the learning process is in the first phase and normalize the feature, just like what the BN operation does. Otherwise, if $a_t\leq\tau$, then we consider it has already jumped to the second phase, stop updating $\mu_i$ and $\sigma_i^2,$ and use constant $\mu_i$ and $\sigma_i^2$ to generate stable features. In the $t$-th batch, $\mu_{i,t}$ and $\sigma_{i,t}^2$ are computed as follows. If $a_t>\tau$, we have $\mu_{i,t}=(1-m)\mu_{i,t-1}+mE_{x \in \text{batch}}[f_i]$ and $\sigma^2_{i,t}=(1-m)\sigma^2_{i, t-1}+mVar_{x \in \text{batch}}[f_i]$; otherwise, $\mu_{i, t}=\mu_{i, t-1}$ and  $\sigma^2_{i,t}=\sigma_{i, t-1}^{2}$, where  $m=0.1$.
>
> To this end, we **conducted experiments** on two types of MLPs (*i.e.*, 9-layer MLPs and 11-layer MLPs) to compare the proposed method with BN. For each type of MLP, we trained three versions MLPs on the CIFAR-10 dataset. The vanilla MLP had 512 neurons in each layer. We added the proposed norm operation after the first, the third, the fifth, and the seventh linear layers. layers, and constructed the network *MLP-norm*. For fair comparison, we constructed a baseline MLP, namely *MLP-BN*, by adding the BN operation in the same positions as in *MLP-norm*. Figure 51 in Section P of the supplementary material shows that both the *MLP-BN* and *MLP-norm* alleviated the learning-sticking problem. However, *MLP-norm* was optimized much more faster than MLP-BN, because our theoretical analysis told us that it was not necessary to continue updating $\mu_i$ and $\sigma_i^2$, if the learning process did not have a risk of feature collapse, thereby alleviating the optimization problems found in [c10, c11].
>
> [c10] Xiang Li, Shuo Chen, Xiaolin Hu, and Jian Yang. Understanding the disharmony between dropout and batch normalization by variance shift. In Computer Vision and Pattern Recognition, 2019.
> [c11] Angus Galloway, Anna Golubeva, Thomas Tanay, Medhat Moussa, and Graham W Taylor. Batch normalization is a cause of adversarial vulnerability. arXiv, abs/1905.02161, 2019.
> [c12] Tim Salimans, Ian Goodfellow, Wojciech Zaremba, Vicki Cheung, Alec Radford, and Xi Chen. Improved techniques for training gans. In Neural Information Processing Systems, 2016.

---

> > ### Comment · Reviewer_7KPL · 2022-11-29
> > **Response to Authors**
> >
> > Thanks for the authors' responses. The responses solve my concerns well. I am willing to raise my score.

---

### Official Review · Reviewer_4Wyv · 2022-10-30

**Confidence:** 4
**Correctness:** 3
**Technical Novelty And Significance:** 2
**Empirical Novelty And Significance:** 3
**Recommendation:** 5

**Clarity, Quality, Novelty And Reproducibility:**

The description of the feature collapse phenomenon is very clear and easy to follow. The results are also very interesting and could provide insights into understanding the learning dynamics of MLPs. However, this phenomenon is only observed in restricted cases, i.e., the vanilla training without batch normalization or other tricks. On the other hand, the result may provide new insight into the role of batch normalization and other techniques. The analysis is limited and based on strong assumptions.


**Strength And Weaknesses:**

## Strength
-  The feature collapse phenomenon in the first phase is very interesting. It could provide insights into understanding the learning dynamics of MLPs. This phenomenon complements the neural collapse phenomenon [A] observed at the end of the training stage.
-  Partial analysis based on assumptions is provided to understand the feature collapse phenomenon.
## Weakness
- The feature collapse phenomenon happens only in certain cases. For example, this phenomenon disappears when batch normalization is used. This point was not clear until section 4.3. Given that batch normalization is a common practice in training deep neural networks, it should be clearly specified the training settings for observing the feature collapse phenomenon at the beginning (e.g., in both abstracts and introduction).
- Section 4.1 provides two perspectives to analyze the feature collapse phenomenon. But there is no description of why these two perspectives are needed and how they could provide an explanation. For example,  perspective 1 focuses on the change of the weight vectors and observes the existence of the common optimization direction shared by different weight vectors. But recall that the feature collapse phenomenon refers to similar features across different inputs. It is not clear how perspective 1 can provide an explanation for the feature collapse phenomenon.
- The analysis in section 4.2 is based on very strong assumptions. For example, Theorem 2 assumes that the features of different samples from the same category have been pushed in the same direction. This seems already assume the collapse phenomenon.
- It is not clear how the analysis in section 4.2 deals with samples from different classes. Assumption 1 comes out of the blue, and there is no description for it.
- Overall, I found it is not easy to follow section 4.2. The presentation of the analysis could be improved.
- Section 4.3 shows that some techniques (like batch normalization) can alleviate the feature collapse phenomenon, while others (like L2 regularization) can strengthen this phenomenon. So should we alleviate or strengthen this phenomenon? How does this phenomenon affect performance?

[A] Papyan, Han, Donoho. Prevalence of neural collapse during the terminal phase of deep learning training. Proceedings of the National Academy of Sciences, 2020.

____________________ After rebuttal_______________

I appreciate the authors' great efforts in addressing my comments. I have increased my rating, but I still have concerns about the analysis. Per the comment "The key problem of explaining the TFC phenomenon is to prove that the significance of such a common direction is likely to be further enhanced, just like a "self-enhanced system." Without the proof of the "self-enhanced system," the assumed initial common direction does not significantly decrease the diversity of features, and cannot exhibit the TFC phenomenon.", I am still not clear why the common gradient direction is not enough. Also, if it behaves like a "self-enhanced system", then the features will only become more collapsed, but why it disappears in the second phase?

**Summary Of The Paper:**

This paper focuses on learning multi-layer perceptrons (MLPs), particularly the feature collapse phenomenon in the first phase, i.e., the diversity of features over different samples keeps decreasing in the first phase of training until samples of different categories share almost the same feature. The authors explain such a phenomenon in terms of the learning dynamics of MLPs.

**Summary Of The Review:**

Based on the above comments, this paper studies an interesting phenomenon, the feature collapse phenomenon in the first stage of training. While this result is interesting and could provide insight into the training dynamics of MLP, this result is only observed under strict settings, the analysis seems to provide limited guidance on why this happens as it is based on strong assumptions and only considers the samples from the same classes, and it is not clear how this phenomenon affects the performance (since one can either alleviate or strengthen this phenomenon by some common techniques like batch normalization and weight decay).

---

> ### Author Response · Authors · 2022-11-17
> **Response to Reviewer 4Wyv (Part 1)**
>
> Thank you for your great efforts on the review of this paper. We will try our best to answer all your questions.
> Please let us know if you still have further concerns, or if you are not satisfied with the current responses, so that we can further update the response ASAP.
>
> ---
> Q1: "The feature collapse phenomenon in the first phase is very interesting... This phenomenon complements the neural collapse phenomenon [A] observed at the end of the training stage."
>
> A: Thanks a lot for your appreciation. We have cited two papers about neural collapse [c1][c2]  in Section A on Page 13. [c1][c2] mainly focus on the final stage of the training process, when DNNs are well trained, *i.e.*, the training loss is very close to zero. In contrast, our study mainly focuses on **the early epochs of the training process**, when DNNs encode features of very few categories.
>
> [c1] Papyan V, Han X Y, Donoho D L. Prevalence of neural collapse during the terminal phase of deep learning training[J]. Proceedings of the National Academy of Sciences, 2020, 117(40): 24652-24663.
> [c2] Han X Y, Papyan V, Donoho D L. Neural collapse under mse loss: Proximity to and dynamics on the central path[J]. arXiv preprint arXiv:2106.02073, 2021.
>
> ---
> Q2: Ask to specify training settings of the TFC phenomenon in both the abstract and the introduction. "... it should be clearly specified the training settings for observing the feature collapse phenomenon at the beginning (*e.g.*, in both abstracts and introduction)."
>
> A: A good suggestion. We have followed your suggestions to clarify training settings of the TFC phenomenon in both the abstract and the introduction (please see the abstract and the introduction on Page 1). In fact, the TFC phenomenon is common in practice, especially when the DNN is difficult to optimize. For example, according to our analysis, the TFC phenomenon usually appears, when the DNN is very deep, when the task is difficult, when the variance of the initial weights is small, and when the DNN is learned without momentum or BN layers. In fact, we have further analyzed specific training settings that affect the TFC phenomenon in Section 3.3.
>
> Specifically, under the above training settings, the DNN is usually difficult to learn in the early epochs (or iterations) of the training process. The initialized DNNs usually fail to find a clear optimization direction and exhibit the TFC phenomenon in the first phase. In contrast, it seems that some DNNs do not exhibit such a TFC phenomenon. In fact, such DNNs may still have the first phases, *i.e.*, the training loss does not decrease significantly. But these DNNs' first phases are usually too short to be clearly observed. For example, the TFC phenomenon may only appear in a few iterations within the first epoch, instead of never appearing.
>
> More crucially, compared to the normal loss-decreasing process in training, it is more important to study those cases that DNNs are difficult to be optimized. Because when DNNs are difficult to be optimized, people usually tune the DNN blindly or owe the phenomenon to the difficulty of the training task. This may miss the essential reason for a broad class of DNNs’ optimization problems. To this end, the discovery and theoretical analysis of this **fundamental yet counter-intuitive** TFC phenomenon would have a broad impact on explaining the optimization of DNNs.

---

> ### Author Response · Authors · 2022-11-17
> **Response to Reviewer 4Wyv (Part 2)**
>
> Q3: Ask to discuss more about the relationship between two perspectives in Section 4.1. "... there is no description of why these two perspectives are needed and how they could provide an explanation... It is not clear how Perspective 1 can provide an explanation for the feature collapse phenomenon."
>
> A: Thanks a lot. We have followed your suggestions to add three paragraphs on Page 3 and Page 4 to further clarify the relationship between two perspectives.
>
> We propose two perspectives to explain how different weight vectors are changed along a common direction $C^{(l)}$ in the first phase of the training process. It is because we can prove that the significance of the common direction will be further enhanced, just like a "self-enhanced system" by comparing the common directions formulated from two perspectives (please see the fourth paragraph in Section 3.2 on Page 6). Then, the self-enhancement of the significance of the common direction well explains the decreasing diversity of features (please see the third paragraph on Page 8). The above analysis can be summarized in the following two steps.
>
> ***Step1: We aim to analyze the existence of the common optimization direction of weight vectors $w_{t,i}^{(l)}$ from two perspectives.***
>
> Specifically, **Perspective 1 describes the influence of the common direction $C^{(l)}$ of the weight change in $l$-th layer**. $\Delta W_{t}^{{(l)}^\top}=\Delta V_{t}^{(l)}C^{(l)^\top}+\Delta \varepsilon_{t}^{(l)}$  in Equation (4) decomposes the weight change $\Delta W^\top_t$ into the component along a common direction $C^{(l)}$ and a component along other directions $\Delta \varepsilon_{t}^{(l)}$, where $\Delta V_{t}^{(l)} \in \mathbb{R}^d$ denotes the coefficient vector for weight changes of different weight vectors along the common direction $C^{(l)}$. Besides, $\Delta \varepsilon_{t}^{(l)}$ is a relatively small "noise" term, which is orthogonal to $C^{(l)}$, *i.e.* , $\Delta \varepsilon_{t}^{(l)}C^{(l)}=\boldsymbol 0$. The experimental verification of the significance of the common direction  $C^{(l)}$  is shown in Figure 4 on Page 5.
>
> Besides, **Perspective 2 describes the influence of the common direction $C^{(l+1)}$ of the weight change in $(l+1)$-th layer based on feature gradients $\dot F_t^{(l+1)}$**.  $\Delta W_{t}^{(l)}=\Delta W_{\text{\textrm{primary}},t}^{(l)}+ \sum_{k=1}^h\Delta W_{\text{\textrm{noise}}, t}^{(l,k)} = \Gamma_t^{(l)} F_{t}^{(l-1)^{\top}}+ \kappa_t^{(l)^\top}$ in Equation (5) decomposes the weight change $\Delta W^\top_t$ into the component along a common direction $C^{(l+1)}$ and a component along other directions. $\Delta W_{\textrm{primary},t}^{(l)}$ denotes the component along the primary common direction, and $\Delta W_{\text{\rm{noise}}, t}^{(l,k)}$ denotes the component along the $k$-th common direction in the noise term. The experimental verification of the significance of the common direction $C^{(l+1)}$ is shown in Table 1 on Page 5.
>
> ***Step2: We aim to prove that the significance of the common direction will be boosted just like a “self-enhanced system” by comparing two perspectives.***
>
> Specifically, we **compare these two perspectives** and prove that the feature $F_{t}^{(l-1)}$ and the vector $V_{t}^{(l)}$ become more and more similar to each other in the first phase, *i.e.*, proving the self-enhancement of the significance of the common direction (please see section 3.2).
>
> In this way, the self-enhancement of the significance of the common direction well explains the decreasing diversity of features (please see the fourth paragraph on Page 8).

---

> ### Author Response · Authors · 2022-11-17
> **Response to Reviewer 4Wyv (Part 3)**
>
> Q4: Concerns about the relationship between the assumption in Theorem 2 and the final proof of the TFC phenomenon. "Theorem 2 assumes that features of different samples from the same category have been pushed in the same direction. This seems already assume the collapse phenomenon."
>
> A: A good question. In fact, this assumption does not directly reflect the TFC (collapse) phenomenon, but is just the starting point of deriving the TFC phenomenon. Specifically, the assumed common direction is just a very vague phenomenon, which is far from the collapse of features. Such an assumption can be explained intuitively, and we have also conducted **extensive new experiments**  Besides, we have also conducted **extensive new experiments** to verify the trustworthiness of the assumed vague initial common direction. Therefore, our research is to prove the further self-enhancement of such a common direction that finally leads to the collapse of features. We have followed your suggestions to refine the presentation in the main paper. Please see Section 3.2 on Pages 6-8 of the main paper.
> - **This assumption does not essentially lead to a circular argument of our proof.** Based on the analysis in the last paragraph on Page 7 and experiments in the next paragraph, it is quite natural for a DNN first learns very few categories in the very beginning epochs, and this makes a few common directions of weight changes. However, **the saliency of such an initial common-direction phenomenon is still FAR FROM the TFC phenomenon.** The key problem of explaining the TFC phenomenon is to prove that the significance of such a common direction is likely to be further enhanced, just like a "self-enhanced system." Without the proof of the "self-enhanced system," the assumed initial common direction does not significantly decrease the diversity of features, and cannot exhibit the TFC phenomenon. Specifically, such a "self-enhanced system" keeps decreasing the feature diversity in the first phase until all samples of different categories share almost the same feature in the first phase, which leads to the TFC phenomenon.
> - **We have conducted new experiments to verify the trustworthiness of this assumption**. The verification of this assumption can be described in two following aspects.
>
>   **First**, we conducted experiments to verify that DNNs only encode features of very few (a single or two) categories in early epochs of training. Specifically, Figures 47-50 in Section O of the supplementary material show that DNNs (*e.g.*, MLPs, VGG-11, VGG-13, and ResNet-18) usually first learned a few categories, instead of simultaneously learning all categories. Only one or two categories have higher accuracies than random guessing.
>
>
>   **Second**, the above phenomenon that ''DNNs only encode features of very few (a single or two) categories in early epochs'' naturally ensures that different samples from the same category have been pushed in the common optimization direction (please see Figure 2 on Page 3 for more details). Specifically, the learned single or two categories usually have the simplest and common features shared by most training samples. These samples are quite similar to each other and share relatively stable patterns. Therefore, DNNs will learn common patterns in a single or two categories, and the optimization of a single or two categories provides very few optimization directions in the first phase. In this way, different samples from the same category have been pushed in the same direction, which is verified in Figure 2 in the main paper and Section B of the supplementary material. These experimental results indicated that features of different samples from the same category have been pushed in the same direction.

---

> > ### Comment · Reviewer_4Wyv · 2022-11-29
> > **Thank you for the responses**
> >
> > Thank you for the clarification and for addressing my comments. I have increased my rating, but I still have concerns about the analysis. For example, I am still not clear why the common gradient direction is not enough as by common gradient direction, gradient descent pushes all the features to be closer. Also, if it behaves like a "self-enhanced system" as the authors claim, then the features will only become more collapsed as gradient descent proceeds, but why does the feature collapse phenomenon disappear in the second phase?

---

> > > ### Author Response · Authors · 2022-11-30
> > > **Response to further questions of Reviewer 4Wyv in the second round (Part 1)**
> > >
> > >
> > > Thank you very much for your comments, and we will answer all your concerns. **Please let us know if you still have further concerns, so that we can update the response as soon as possible.**
> > >
> > > Q1: ''I am still not clear why the common gradient direction is not enough as by common gradient direction, gradient descent pushes all the features to be closer.''
> > >
> > > A:  A good question. As mentioned in the last paragraph on Page 6, the initial common gradient direction is just a **static phenomenon** observed in **a certain early iteration**. More crucially, the temporary common direction in a certain iteration does not mean that **all neurons will continue to be pushed towards such a direction in future iterations. Therefore, the key task in this research is to prove the future tendency of the self-enhancement of the common direction** (please see Section 3.2 on Page 6), *i.e.,* proving that neurons will continue to be optimized along the same direction.
> > >
> > > Besides, the initial common direction is too vague to cause the TFC phenomenon (please see the last paragraph on Page 6). It is because feature gradients of different samples on an initialized DNN can be considered high-dimensional random vectors, which are roughly orthogonal to each other due to their high dimensionality [c1-c3]. For example, given a 9-layer MLP with initialized parameters, we calculated the cosine similarity of feature gradients in the first linear layer between different samples. The first linear layer had 512 neurons. At the very beginning of the learning process, the cosine similarity of feature gradients between different samples was only 0.08. **Such a small cosine similarity (0.08) of initial feature gradients was far from the TFC phenomenon that corresponded to a high cosine similarity (***e.g.***, cos=0.6 in Figure 2).** Thus, we could not consider such a vague initial common gradient direction as the **direct reason** for the TFC phenomenon. Instead, we need to further prove the self-enhancement of such a common direction.
> > >
> > > In sum, we can summarize our work into the following three items.
> > > (1) Without the proof of the self-enhanced common direction, the vague initial common direction does not significantly increase feature similarity and does not cause the TFC phenomenon.
> > > (2) The vague initial common direction of feature gradients is just the starting point of the self-enhancement of the common direction, rather than its only reason. The vague initial common direction can be widely observed in different DNNs for various tasks, but such a vague initial common direction does not always cause a self-enhancement and lead to the TFC phenomenon, according to Section 3.3. We prove that both the BN operation and the momentum may prevent the initial common direction from being further enhanced towards the TFC phenomenon.
> > > (3) The further self-enhancement of such a common direction significantly boosts the significance of feature collapse, which may even make the learning process get stuck.
> > >
> > > [c1] Gorban A N, Tyukin I Y. Blessing of dimensionality: mathematical foundations of the statistical physics of data[J]. Philosophical Transactions of the Royal Society A: Mathematical, Physical and Engineering Sciences, 2018, 376(2118): 20170237.
> > > [c2] https://www.cs.princeton.edu/courses/archive/fall14/cos521/lecnotes/lec11.pdf
> > > [c3] https://courses.cs.washington.edu/courses/cse521/16sp/521-lecture-6.pdf

---

> > > ### Author Response · Authors · 2022-11-30
> > > **Response to further questions of Reviewer 4Wyv in the second round (Part 2)**
> > >
> > > Q2: "if it behaves like a "self-enhanced system", then the features will only become more collapsed, but why it disappears in the second phase?"
> > >
> > > A: A good question. We have explained this concern in the fifth paragraph on Page 8. Our analysis focuses on the early epochs of training, when only a few training samples of one or two dominating categories can be confidently classified. However, when the optimization of a single or two dominating categories in the first phase soon saturates at the end of the first phase, gradients on the correctly classified samples of the dominating categories vanish. Then, gradients from training samples of other categories weaken the dominating role of a single or two categories in the learning of the MLP. Thus, the ''self-enhanced system'' is destroyed, and the learning of the MLP enters the second phase.
> > >
> > > To this end, we have followed your suggestions to conduct **new experiments** to verify the above analysis. Specifically, we trained a 10-layer MLP on the CIFAR-10 dataset. We used the metric $S=E_{x \in X_c} [||\frac{\partial Loss(x)}{\partial W^{(l)}_{t}}||_F]$ to measure the average gradient strength over samples in the category $c$, where $x \in X_c$ denotes a training sample in the category $c$.  Table 1 shows that the average gradient strength of dominating categories (*e.g.*, the ship is the dominating category with the highest training accuracy) was much higher than other categories in the 2400-th iteration.  The average gradient strength of the dominating category (*i.e.,* ship) decreased, and the average gradient strength of other categories increased at the end of the first phase (from the 2400-th iteration to the 2600-th iteration). In this way, the MLP learned features from more diverse categories at the beginning of the second phase, instead of exclusively learning from the dominating categories. Therefore, according to our theoretical analysis, the learning process of the MLP would escape from the "self-enhanced system." Thus, the training loss of other categories began to decrease.
> > >
> > > **Table 1: Average gradient strength of different categories in different iterations.**
> > > | $S(\times 10^{-2})$ | airplane | automobile|bird|cat|deer|&emsp;&emsp;&emsp; dog|frog|horse|&emsp;&emsp;&emsp;ship|truck|
> > > | :----:| :----: | :-------: | :----:| :----: | :----: |:-------------:| :----: | :----: | :--------------:| :----: |
> > > | Iteration 2400, the end of the first phase | 1.39 | 1.37 | 1.68 | 1.09 | 1.14 | 1.49 | 1.93 | 1.04 |**12.53** (about 7 times greater than average gradient strength of other categories)| 4.89|
> > > | Iteration 2600, the beginning of the second phase | 6.80 | 4.32 | 3.71 | 12.98 | 8.76 | **16.83** (just 2.08 times greater than the average gradient strength of other categories)| 11.22 | 9.46 |8.43| 6.89|

---

> ### Author Response · Authors · 2022-11-17
> **Response to Reviewer 4Wyv (Part 4)**
>
> Q5: About the presentation in Section 4.2. "It is not clear how the analysis in section 4.2 deals with samples from different classes. Assumption 1 comes out of the blue, and there is no description for it."
> Q6: "I found it is not easy to follow section 4.2. The presentation of the analysis could be improved."
>
> A to both Q5 and Q6: Thanks. We have followed your suggestions to revise our presentation in the main paper. Please see Section 3.2 (Section 4.2 in the previous paper) on Pages 6-8 of the main paper.
>
> **First, let us summarize the overall logic of Section 3.2 into the following three steps, where the role of Assumption 1 is explained in Step 2.**
> - **Step1: Explaining the phenomenon that the significance of the common direction is enhanced by all training samples in a certain category**.
>   First of all, we explain the simplest case that only considers samples in **a certain category**. Specifically, Lemma 3 and Theorem 2 indicate that features of training samples **in the same category** $c$ are all pushed towards a common direction $\alpha_c V_{t}^{(l)}$, and training samples in the category $c$ all push $V_{t}^{(l)}$ towards $\alpha E_{x \in X_c} [F_{t}^{(l-1)}|x]$. In this way, $F_{t}^{(l-1)}$ and $\alpha_{c} V_{t}^{(l)}$ become increasingly similar to each other, *i.e.*, the significance of the common direction is enhanced by all training samples in **a certain category**.
> - **Step2: Extending the conclusion of the enhancement of the common direction to a more generic case that considers samples in multiple categories**.
>   Then, we further extend the above conclusion to a more generic case of **all categories**.  Specifically, Theorem 2 and Assumption 1 show that the overall learning effects of all training samples are dominated by very few categories $\hat{c}$. Besides,  features $F_t^{(l-1)}$ of different samples are all pushed towards the vector $\alpha_{\hat c} V_t^{(l)}$. On the other hand, $V_t^{(l)}$ is also pushed towards $\alpha_{\hat c} E_{x \in X_{\hat c}} [F_{t}^{(l-1)}|x]$. In this way, $F_t^{(l-1)}$ and $\alpha_{\hat c} V_t^{(l)}$ become increasingly similar to each other, *i.e.*, the significance of the common direction is very likely to be further enhanced, just like a “self-enhanced system.”
>
> **The role of Assumption 1.** In this step, Assumption 1 indicates that MLPs first learn a single or two categories in the first phase. In this way, the learning of the MLP is dominated by training samples of a single or two categories in very early iterations. Therefore, the overall learning effects of all training samples are dominated by very few categories. Later, we will **conduct new experiments** to verify the trustworthiness of Assumption 1.
> - **Step3: We prove that the self-enhancement of the common direction will decrease the diversity of features,** *i.e.*, **explaining the TFC phenomenon**.
>   Specifically, according to the “self-enhanced system” in **Step 2**, features $F_t^{(l-1)}$ of different samples are consistently pushed towards the same vector  $\alpha_{\hat c} V_t^{(l)}$. It increases the similarity between features of different samples $E_{x,x'\in X} [\cos(F_t^{(l-1)}|x,F_t^{(l-1)}|x')]$ in the first phase. In other words, the “self-enhanced system” decreases the diversity of features and leads to the TFC phenomenon.
>
> **In addition, we have conducted new experiments to verify the trustworthiness of Assumption 1,** *i.e.*, **DNNs encode features of very few (a single or two) categories in early epochs.**
>
> Please see Section O of the supplementary material. Figures 47-50 in Section O of the supplementary material show that DNNs (*e.g.*, MLPs, VGG-11, VGG-13, and ResNet-18) encoded features of very few (a single or two) categories in early epochs. Experimental results show that DNNs usually first learned a few categories, instead of simultaneously learning all categories.

---

> ### Author Response · Authors · 2022-11-17
> **Response to Reviewer 4Wyv (Part 5)**
>
> Q7: "... should we alleviate or strengthen this phenomenon? How does this phenomenon affect performance?"
>
> **Answer 1 of the two answers to Q7**: Thank you. As discussed in Section 2 of the main paper, the TFC phenomenon has a potential connection with the learning-sticking problem, *i.e.*, when the deep neural networks (DNNs) are very deep, the loss minimization may get stuck, which can be considered as a strong first phase with an infinite length. Thus, **alleviating the TFC phenomenon may directly solve the learning-sticking problem.** Therefore, breaking the condition for the TFC phenomenon is crucial for the optimization of a DNN and is of significant values in practice.
>
> **New Experiments**. We have **conducted new experiments** to verify that alleviating the TFC phenomenon can solve the learning-sticking problem. Specifically, we trained three different types of DNNs on the MNIST, the CIFAR-10, and the Tiny ImageNet datasets for evaluation, including MLPs, VGGs, and ResNets. In experiments, these DNNs all suffered from the learning-sticking problem (*i.e.*, the loss minimization of these DNNs get stuck), when their initial weights were sampled from $N(\pmb{0}, \pmb{\Sigma}=\gamma_1 \sigma_{\textrm{var}}^2I )$. $\sigma_{\textrm{var}}^2$ was computed according to [c3] and $\gamma_1=0.1$. Based on our study, we increased the variance of initialization to alleviate the TFC phenomenon. Figures 30-35 in Section C of the supplementary material verify that we could solve the learning-sticking problem by increasing the variance of initial weights to $\gamma_2 \sigma_{\textrm{var}}^2I  (\gamma_2 = 1.0)$. These experiments showed that alleviating the TFC phenomenon avoided the learning-sticking problem. In contrast to being stuck in the first phase, the theoretical analysis of our study provided guidance for enabling normal training of a DNN without letting the performance be hurt by the learning-sticking problem.
>
> [c3] Xavier Glorot and Yoshua Bengio. Understanding the difficulty of training deep feedforward neural networks. In Proceedings of the thirteenth international conference on artificial intelligence and statistics, pp. 249–256. JMLR Workshop and Conference Proceedings, 2010.

---

> ### Author Response · Authors · 2022-11-17
> **Response to Reviewer 4Wyv (Part 6)**
>
> Q7: "... should we alleviate or strengthen this phenomenon? How does this phenomenon affect performance?"
>
> **Answer 2 of the two answers to Q7**: **Using our theory to develop new normalization methods and experimental verification.** Although existing experiments in Section C of the supplementary material have shown the utility of our findings, **we have followed your suggestions to design a new normalization operation based on our theoretical analysis of the TFC phenomenon, which could better solve the learning-sticking problem without causing the feature instability.** The new normalization operation was designed considering the following two findings.
> - Our theoretical analysis told us that the centering operation in BN could alleviate the TFC phenomenon.
> - Previous studies found some shortcomings of the BN operation, *i.e.*, the BN operation usually caused unstable features. Thus, the BN operation was found incompatible with the dropout [c4], hurt the classification accuracy in adversarial training [c5], and decreased the quality of images generated by generative models [c6].
>
> **Therefore, according to our analysis, we only need to update the dynamic normalization parameters** (*i.e.*, **$\mu_i$ and $\sigma_i$ in the following equation) in the first phase to avoid the learning-sticking problem, instead of applying the dynamic normalization parameters in the entire training process. In this way, we can simultaneously solve the learning-sticking problem and avoid unstable features.**
>
> Specifically, we are given the output feature $F=[f_1,f_2,\ldots,f_h]$$\in \mathbb{R}^h$ of the $l$-th linear layer *w.r.t.* the input sample $x$, where $f_i$ denotes the $i$-th dimension of the feature. The new normalization operation is given as $$\textit{norm}(f_i)=(f_i-\mu_i)/\sigma_{i},$$where $\mu_i$ and $\sigma_i$ denote the mean value and the standard deviation of $f_i$ over different samples, respectively. Unlike the BN operation, we only update the mean value $\mu_i$ and the standard deviation $\sigma_i$ in the first phase, as follows.
>
> if $a_t>\tau$, we have $\mu_{i}=E_{x \in \text{batch}}[f_i]$ and $\sigma^2_{i}=Var_{x \in \text{batch}}[f_i]$; otherwise, $\mu_{i}=\mu_{i, t-1}$ and  $\sigma^2_{i}=\sigma_{i, t-1}^{2}$. We keep updating $a_{t}=0.99a_{t-1}+0.01E_{x,x' \in \text{batch}}[\cos(F|x,F|x')]$ in the $t$-th batch, to represent the current cosine similarity between features of different samples. If $a_t$ is greater than a threshold $\tau=0.3$, then we consider the learning process is in the first phase and normalize the feature, just like what the BN operation does. Otherwise, if $a_t\leq\tau$, then we consider it has already jumped to the second phase, stop updating $\mu_i$ and $\sigma_i^2,$ and use constant $\mu_i$ and $\sigma_i^2$ to generate stable features. In the $t$-th batch, $\mu_{i,t}$ and $\sigma_{i,t}^2$ are computed as follows. If $a_t>\tau$, we have $\mu_{i,t}=(1-m)\mu_{i,t-1}+mE_{x \in \text{batch}}[f_i]$ and $\sigma^2_{i,t}=(1-m)\sigma^2_{i, t-1}+mVar_{x \in \text{batch}}[f_i]$; otherwise, $\mu_{i, t}=\mu_{i, t-1}$ and  $\sigma^2_{i,t}=\sigma_{i, t-1}^{2}$, where  $m=0.1$.
>
> To this end, we **conducted experiments** on two types of MLPs (*i.e.*, 9-layer MLPs and 11-layer MLPs) to compare the proposed method with BN. For each type of MLP, we trained three versions MLPs on the CIFAR-10 dataset. The vanilla MLP had 512 neurons in each layer. We added the proposed norm operation after the first, the third, the fifth, and the seventh linear layers. layers, and constructed the network *MLP-norm*. For fair comparison, we constructed a baseline MLP, namely *MLP-BN*, by adding the BN operation in the same positions as in *MLP-norm*. Figure 51 in Section P of the supplementary material shows that both the *MLP-BN* and *MLP-norm* alleviated the learning-sticking problem. However, *MLP-norm* was optimized much more faster than MLP-BN, because our theoretical analysis told us that it was not necessary to continue updating $\mu_i$ and $\sigma_i^2$, if the learning process did not have a risk of feature collapse, thereby alleviating the optimization problems found in [c4, c5].
>
> [c4] Xiang Li, Shuo Chen, Xiaolin Hu, and Jian Yang. Understanding the disharmony between dropout and batch normalization by variance shift. In Computer Vision and Pattern Recognition, 2019.
> [c5] Angus Galloway, Anna Golubeva, Thomas Tanay, Medhat Moussa, and Graham W Taylor. Batch normalization is a cause of adversarial vulnerability. arXiv, abs/1905.02161, 2019.
> [c6] Tim Salimans, Ian Goodfellow, Wojciech Zaremba, Vicki Cheung, Alec Radford, and Xi Chen. Improved techniques for training gans. In Neural Information Processing Systems, 2016.

---

### Author Response · Authors · 2022-11-24
**Looking forward to further discussion**

Dear Reviewers,

Thanks for your valuable comments to help the improvement of our paper.

We would appreciate it if you could let us know if our responses have addressed your concerns and whether you still have any other concerns.

We would be happy to do any follow-up discussion or address any additional comments.

Best regards,

Authors

---

### Author Response · Authors · 2022-12-09
**Looking forward to further discussion**

Thank you very much for your responses. We are glad that all reviewers have noticed our responses and extensive new experiments towards the concerns, and Reviewer (4cUL) changed the rating to "... it is hard for me to decide between 6 and 8." Can we consider it 7?

**Please let us know if you still have further concerns, so that we are glad to update the response as soon as possible.**

---

### Decision · Program_Chairs · 2023-01-20

**Decision:**

Reject

**Justification For Why Not Higher Score:**

Using non-standard initialization variance and removing batch normalzation

**Justification For Why Not Lower Score:**

N / A

**Metareview: Summary, Strengths And Weaknesses:**

This paper demonstrate and explains a feature collapse phenomenon during an early phase of training in which the diversity of the feature representations decreases and the loss value does not change significantly. Authors then study the dynamics of training and discuss how different architectural or hyper-parameter choices could alleviate the feature collapse. There paper is well-written and easy to follow.

Understanding phase transitions and existing phenomena in deep learning is crucial to the community so in that light, this paper is a very interesting attempt. Moreover, the comprehensive experiments and theoretical results are interesting. However, the current paper has several shortcomings:

- Non-standard setting: In order to see the feature collapse phenomenon, authors had to change 2 things:  a) Use a non-standard initialization that has very low variance. b) Remove Batch Normalization. This makes the work much less interesting because the phenomenon is not related to the setting anyone wants to use for training neural networks.

- Lack of discussion about the role of norm of weights/features and the loss function: Given that the feature collapse phenomenon would disappear when initial weights are normalized appropriately, this phenomenon must be related to norm of weights/features as well as scale sensitivity of the cross-entropy loss function. To have a complete picture, it is important for authors to discuss the role of loss and norm of the weights and report the change in the norm of weight/features as well.

- Missing related work: The dynamics of learned feature across different layers of deep networks and the role of appropriate initialization in that has been studied before. See for example [1,2]. Moreover, the dynamics of representation and gradient alignment has been studied before. See [3] and its related work. The relationship between prior work and this paper should be clearly discussed.


This paper is borderline. Given the above, my final decision is to reject the paper but I want to encourage authors to resubmit after improving the paper in light of the reviews. While I understand that this decision might be disappointing, I think improving the work and resubmitting it eventually helps both authors and the community.

[1] Xiao et. al. Dynamical Isometry and a Mean Field Theory of CNNs: How to Train 10,000-Layer Vanilla Convolutional Neural Networks, ICLR 2018.

[2] Brock et. al. High-performance large-scale image recognition without normalization, ICML 2021.

[3] Mehta et. al. Extreme Memorization via Scale of Initialization, ICLR 2021.


**Summary Of Ac-Reviewer Meeting:**

The main contributions of the paper and its pros/cons were discussed.

4cUL: Feature collapse phenomenon is interesting and theory is interesting to me as well. However, I am not an expert in this area.

4Wyv: The main shortcoming is that they have to remove batch normalization in order to observe feature collapse.